# CNS-wide repopulation by hematopoietic-derived microglia-like cells corrects progranulin deficiency in mice

Pasqualina Colella [1] ✉, Ruhi Sayana[1], Maria Valentina Suarez-Nieto [1], Jolanda Sarno [2,3], Kwamina Nyame [4,5,6], Jian Xiong [4,5,6], Luisa Natalia Pimentel Vera[1], Jessica Arozqueta Basurto[1], Marco Corbo[7], Anay Limaye[1,7], Kara L. Davis [2], Monther Abu-Remaileh[4,5,6] & Natalia Gomez-Ospina [1] ✉

Hematopoietic stem cell transplantation can deliver therapeutic proteins to the central nervous system (CNS) through transplant-derived microglia-like cells. However, current conditioning approaches result in low and slow engraftment of transplanted cells in the CNS. Here we optimized a brain conditioning regimen that leads to rapid, robust, and persistent microglia replacement without adverse effects on neurobehavior or hematopoiesis. This regimen combines busulfan myeloablation and six days of Colony-stimulating factor 1 receptor inhibitor PLX3397. Single-cell analyses revealed unappreciated heterogeneity of microglia-like cells with most cells expressing genes characteristic of homeostatic microglia, brain-border-associated macrophages, and unique markers. Cytokine analysis in the CNS showed transient inductions of myeloproliferative and chemoattractant cytokines that help repopulate the microglia niche. Bone marrow transplant of progranulin-deficient mice conditioned with busulfan and PLX3397 restored progranulin in the brain and eyes and normalized brain lipofuscin storage, proteostasis, and lipid metabolism. This study advances our understanding of CNS repopulation by hematopoietic-derived cells and demonstrates its therapeutic potential for treating progranulin-dependent neurodegeneration.

Hematopoietic stem cell transplantation (HSCT) is the recommended treatment for several genetic disorders with severe and rapid neurodegeneration, including several lysosomal storage disorders (LSDs) and peroxisomal disorders (PSDs)[1,2]. HSCT is also a promising investigational therapy for other neurological diseases, including Friedreich's ataxia[3], Pelizaeus–Merzbacher disease[4], and several neuronopathic LSDs[5]. Despite its proven benefit, HSCT's use for neurological indications has been limited to a few fatal diseases because its risk-benefit assessment is usually unfavorable[1,6]. One of the caveats of HSCT is the common use of allogeneic cells (allo-HSCT) which exposes the recipient to immunological complications while often providing insufficient therapeutic correction[1]. To overcome this risk, autologous transplants of gene-modified HSPCs are being developed[7–9]. These approaches are demonstrating significant benefits in clinical studies

[1]Department of Pediatrics, Stanford University School of Medicine, Stanford, CA 94305, USA. [2]Hematology, Oncology, Stem Cell Transplant, and Regenerative Medicine, Department of Pediatrics, Stanford University, Stanford, CA 94305, USA. [3]Tettamanti Center, Fondazione IRCCS San Gerardo dei Tintori, 20900 Monza, Italy. [4]Department of Chemical Engineering, Stanford University, Stanford, CA 94305, USA. [5]Department of Genetics, Stanford University, Stanford, CA 94305, USA. [6]The Institute for Chemistry, Engineering and Medicine for Human Health (Sarafan ChEM-H), Stanford University, Stanford, CA 94305, USA. [7]MedGenome, Inc, 348 Hatch Dr, Foster City, CA 94404, USA. ✉e-mail: pcolella@stanford.edu; gomezosp@stanford.edu

leading to the approval of two autologous HSCT-based treatments for severe leukodystrophies (Metachromatic leukodystrophy and X-linked adrenoleukodystrophy)[7–9]. However, despite advancements in establishing sources of autologous cells, HSCT-based therapies still encounter obstacles related to their limited efficacy in the central nervous system (CNS) and delayed therapeutic onset.

How HSCT halts neurodegeneration is not completely understood, but the effect is partly due to bone marrow-derived cells that migrate to CNS, where they become long-lived resident myeloid cells, often called microglia-like cells (MGLCs)[10–13]. The mechanisms involved in the recruitment of hematopoietic-derived cells to the CNS are not yet entirely characterized, but myeloablative conditioning of the recipient is required for this process[10,14]. Myeloablation is commonly achieved in the clinic with busulfan (BU), a CNS-penetrant DNA alkylating drug[7–9,15,16]. Even at the highest tolerated myeloablative dose, the combination of BU and HSCT, results in low, variable, and slow-paced engraftment of MGLCs in the CNS[17,18]. This modest and slow engraftment of MGLCs in the brain significantly limits the therapeutic efficacy of HSCT, further skewing the risk/benefit assessment unfavorably and preventing its broader applicability, particularly in diseases with rapid neurological progression[19–22]. To improve the success of allogeneic and autologous HSCT for neurological indications, it is crucial to repopulate the CNS quickly and efficiently with MGLCs. This could enhance HSCT's efficacy for the diseases for which it is currently used and offer a promising option for many other conditions that could theoretically benefit from an HSCT approach.

Several studies have examined the role of microglia depletion in overcoming the limited engraftment of bone marrow (BM)-derived MGLCs in the CNS[18,23,24]. The Colony-stimulating factor 1 receptor (CSF1R) is crucial for the survival of microglia (MG) and macrophages (MF) in rodents and humans[25–27]. Pivotal studies showed that the genetic depletion of CSF1R strongly favors the engraftment of BM-derived cells in the brain without any form of conditioning[23,28–30]. However, pharmacological depletion of MG and MF via inhibition of CSF1R (CSF1Ri) does not[23,31]. Several regimens that combine CSF1Ri with myeloablative total body irradiation or busulfan have been reported, resulting in the near-complete replacement of microglia with BM-derived cells in mice[18,23,24,32,33]. Accordingly, combining myeloablation with CSF1Ri could represent a promising approach for pre-transplant conditioning in neurometabolic indications. However, available regimens need significant optimization for clinical use, and it is crucial to establish a regimen that includes the most appropriate reagents and dosage scheme for translational potential.

CSF1Ri is typically achieved by administering two main inhibitors through the chow, with prolonged courses lasting several weeks to months[18,23,24,33]. PLX5622 has been reported to affect hematopoiesis[34,35], while PLX3397, approved by the FDA for the life-long treatment of individuals affected by tenosynovial giant cell tumor (TGCT), is predicted to have non-specific activity on other receptors[35,36]. Achieving efficacy with the fewest possible administrations is also crucial in optimizing a CSF1Ri regimen, as treatment-related toxicity depends on the dose and duration of treatment[35,36]. Considering the potential toxicities associated with CSF1Ri, a comprehensive evaluation of its impact on hematopoiesis and neurobehavior is essential to support its safety. Additionally, a better understanding of the kinetics of brain repopulation, the signaling molecules involved, and the characteristics of the cells repopulating the brain would provide insights to optimize microglia replacement by MGLCs and predict unwanted toxicities. Herein, we present an optimized and maximally effective conditioning regimen consisting of a short oral course of PLX3397, resulting in fast, robust, and long-term repopulation of the CNS by BM-derived MGLCs. This regimen does not negatively impact the hematopoiesis or neurobehavior in recipient mice. Our studies also determined the kinetics of MGLC repopulation, the signaling events in the brain in response to conditioning and repopulation, and the previously unrecognized heterogeneity of brain-engrafted MGLCs.

To assess the therapeutic potential of our conditioning regimen for a neurological disorder, we applied it to a mouse model of pro-granulin (GRN) deficiency. In humans, insufficient GRN expression causes neurodegenerative diseases with an allele dose-dependent pattern[37]. Bi-allelic loss of function (lof) mutations cause Neuronal Ceroid Lipofuscinosis type 11 (CLN11, OMIM 614706), a rare LSD characterized by childhood-onset cognitive decline, retinitis pigmentosa, and early death[38]. Mono-allelic *GRN* lof mutations represent 10–15% of all cases of Frontotemporal dementia (GRN-FTD, OMIM 607485), an adult-onset disease presenting progressive changes in behavior, personality, and language, ultimately leading to early death[39–43]. CLN11/FTD represents a critical unmet need and is potentially amenable to protein, gene, and cell therapy approaches. However, the effectiveness of these approaches is challenged by the difficulty in delivering the large GRN protein across the blood-brain barrier (BBB), achieving widespread GRN distribution in the CNS[44–47], and by potential complications of supraphysiological or ectopic GRN expression[46,48].

GRN is a highly secreted, ubiquitous lysosomal protein expressed in the CNS in neurons, microglia, and other glial cells[49,50]. Importantly, neighboring cells can take up secreted extracellular GRN via receptor-mediated endocytosis[51,52]. Accordingly, HSCT-based microglia replacement could be an efficacious treatment for GRN deficiency, as secreted GRN from MGLCs can cross-correct other cells while concomitantly replacing diseased microglia[53–56]. Furthermore, by maintaining the expression within a cell type that typically expresses high GRN protein levels, an HSCT-based approach could be safer as it minimizes potential complications of ectopic expression. Herein, we report that transplantation of wild-type HSPCs restores therapeutic GRN levels in the CNS of $Grn^{-/-}$ mice and corrects lipofuscin accumulation, defects of proteostasis, and lipid metabolism in the brain when combined with an optimized, clinically translatable conditioning regimen-based on busulfan and PLX3397.

## Results

### Robust engraftment of bone marrow-derived microglia-like cells in the CNS with busulfan myeloablation and short PLX3397 treatment

Current protocols for replacing microglia with bone marrow-derived cells in mice differ in the type of CSF1R inhibitor used, its formulation, the time of initiation, which can range from 14 days to several months, and the duration of administration, which typically lasts 2 to 4 weeks[18,23,24,33,57–60]. We focused on the CSF1R inhibitor PLX3397 (Pexidartinib, PLX) since it is FDA-approved for treating TGCT and has demonstrated safety even with long-term administration[35,36]. Mouse conditioning with PLX3397 complexed chow results in dose and time-dependent microglia depletion[31,61]. However, inconsistencies among batches of PLX3397-complexed diets[62] and variability in microglia replacement have been reported[33]. To establish a highly efficient and reproducible protocol with high translational potential, we optimized the route and duration of drug administration. We administered PLX3397 (PLX) to adult C57BL/6 mice by oral gavage at 100 mg/kg/day, a dose that was chosen based on the effectiveness of the 580-600 ppm complexed chow[61] and the daily mouse food intake of 4–5 g/day[63]. We observed maximal depletion of CD45 + CD11b+ microglia cells by flow cytometry after a 6-day regimen of PLX administered by oral gavage (95 ± 2% depletion vs untreated mice, Supplementary Fig. 1a–c; and Supplementary Fig. 2a shows the gating scheme). We tested this 6-day PLX regimen in adult C57BL/6 mice transplanted with bone marrow (BM) from coisogenic mice expressing green fluorescent protein (CAG-GFP). Consistent with previous reports, CSF1Ri alone did not result in donor cell engraftment in the brain or bone marrow following either

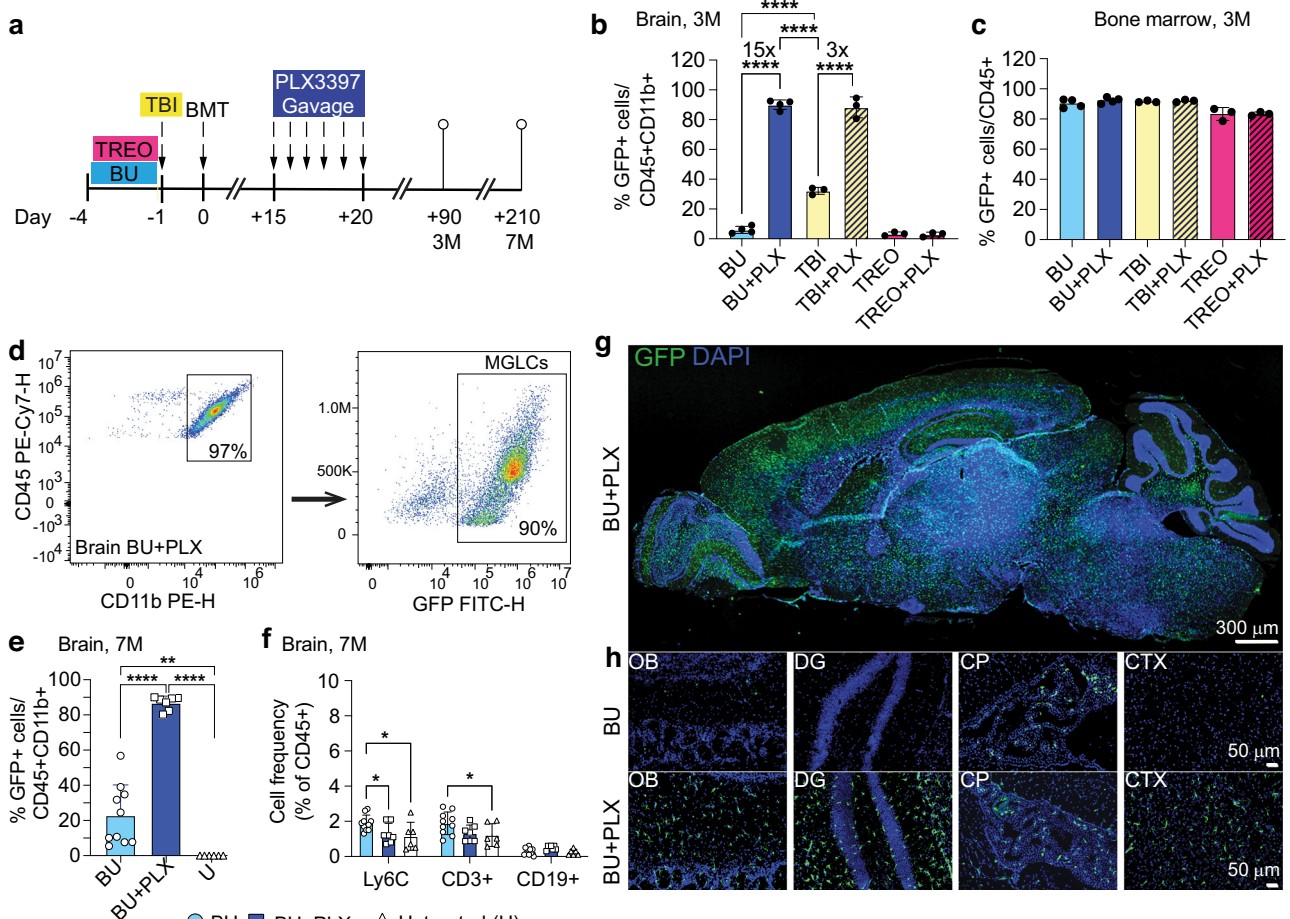

**Fig. 1 | A six-day course of PLX3397 enhances the replacement of microglia by bone marrow-derived MGLCs. a** Experimental timeline: conditioning of adult C57BL/6 mice with total body irradiation (TBI), busulfan (BU) or treosulfan (TREO) combined or not with PLX3397 (PLX) by oral gavage 15 days post bone marrow transplant (BMT). Adult homozygous C57BL/6-CAG-GFP mice were used as bone marrow donors. The time points of the analysis are 3 months (3 M) or 7 months (7 M) post-BMT. **b–f,** Flow cytometry analyses. **b** Fraction of transplant-derived GFP+ microglia-like cells (MGLCs) measured in the brain 3 M post-BMT. **c** Fraction of transplant-derived GFP+ cells measured in the bone marrow (BM) 3 M post-BMT. **a, b** BU n = 4, BU + PLX n = 4, TBI n = 3, TBI + PLX n = 3, TREO n = 3, TREO + PLX n = 3. **d** Representative flow plots of CD45 + CD11b+ cells (left panel, gated on total CD45+ cells) and transplant-derived GFP+ MGLCs (right panel) measured in the brain 7 M post-BMT. **e** Fraction of transplant-derived GFP+ MGLCs in the brain 7 M post-BMT; U: Untreated mice. **f** Fractions of Ly6C + , CD3 + , and CD19+ cells in the brain of mice 7 M post-BMT, and in untreated mice. **e-f** BU n = 10, BU + PLX n = 7, U n = 6. **g** Sagittal section of the brain from a mouse treated with BU + PLX + BMT and analyzed 7 M post-BMT (image representative of n = 7 mice). **h** Representative images of transplant-derived GFP+ cells repopulating the olfactory bulb (OB), dentate gyrus (DG), choroid plexus (CP), and cortex (CTX) of mice conditioned with BU alone (top) or BU + PLX (bottom) and analyzed 7 M post-BMT; images representative of n = 4 mice/group. **g–h** Scale bars are depicted. **b, c, e–f** Data are Mean ± SD. Source data are provided as a Source Data file. Statistical analysis: *p < 0.05, **p < 0.01, ***p < 0.001, ****p < 0.0001, the exact p-values of all comparisons are reported in the Source Data file; **b, e, f** One-way ANOVA with Tukey post-hoc; **c** Kruskal–Wallis test with Dunn's. GFP: green fluorescent protein; DAPI: 4′,6-diamidino-2-phenylindole.

intravenous bone marrow transplant (BMT) or intracerebroventricular delivery (ICV) of lineage-negative BM cells[18,23,31] (Supplementary Fig. 1d–f).

To examine how this short PLX regimen performs in combination with different myeloablative protocols, we compared brain and hematopoietic engraftment in wild-type C57BL/6 mice conditioned with either total body irradiation (TBI, 10 Gy), busulfan (BU, 100 mg/kg), or treosulfan (TREO, 5.5 g/kg), a non-CNS penetrant BU analog[10] (Fig. 1a). Despite comparable bone marrow chimerism, PLX administration resulted in near complete repopulation of the microglia niche when combined with either BU (BU 6 ± 2.4% vs BU + PLX 90 ± 3.2%, 15-fold increase) or TBI (TBI 32 ± 2.4% vs TBI + PLX 88 ± 7%, 3-fold increase) but it did not increase brain engraftment in mice conditioned with TREO (Fig. 1b–c, Supplementary Fig. 2b). The lack of effect with TREO supports the idea that irreversible genotoxic damage of CNS-resident microglia progenitor cells is needed for robust replacement by hematopoietic-derived myeloid cells following BMT.

Our data indicated that using BU as the myeloablative agent, which is the standard drug for conditioning patients with neurometabolic disorders, along with a 6-day course of PLX at 100 mg/kg/day (BU + PLX), robustly increased the replacement of microglia with hematopoietic-derived MGLCs. To reduce the potential effects of CSF1Ri on hematopoietic reconstitution, we tested the effectiveness of administering PLX before transplantation. Concomitant administration of PLX (day −6 to −1) with BU (day −4 to −1) resulted in a 5-fold increase in GFP + MGLC chimerism in the brain compared to BU alone (15 ± 8%) but was significantly less than when PLX was administered after transplant (Supplementary Fig. 1g-i). Side-by-side comparison of PLX administration pre- and post-transplant in recipient *Cx3cr1*-GFP^+/− mice (expressing GFP in microglia and macrophages) transplanted with BM from CAG-RFP^+/− mice (ubiquitously expressing red fluorescent protein, RFP) re-demonstrated that post-transplant administration of PLX resulted in more efficient depletion of host GFP+ microglia and higher

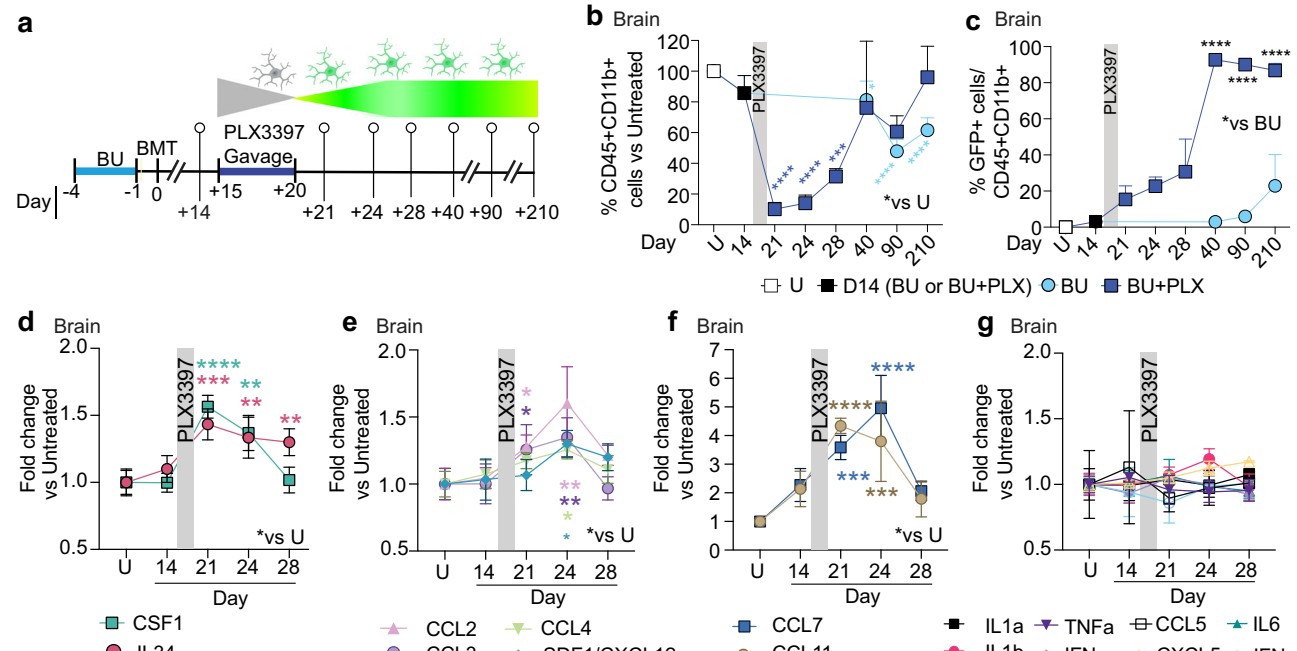

**Fig. 2 | A six-day course of PLX3397 accelerates MGLCs repopulation and induces specific cytokines in the brain. a** Experimental timeline showing sampling time points after BU or BU + PLX and CAG-GFP BMT in C57BL/6 mice. **b–c** Flow cytometry analyses. **b** Fraction of CD45 + CD11b+ cells/total CD45+ cells measured in the brain of mice at the indicated time points (**a**). The quantification and statistics are relative to untreated mice (*p-value vs U). Asterisks: p-value BU vs U (light blue), p-value BU + PLX vs U (dark blue). **c** Fraction of GFP+ microglia-like cells (MGLCs) in the brain at the indicated time points (**a**). The statistics compare BU vs BU + PLX. **b–c** U n = 3; BU day 14 n = 3, day 40 n = 4; BU + PLX n = 3 from day 21 to day 40. **c** The data points for days 90 and 270 are plotted from the experiments described in Fig. 1b and 1e, respectively: BU day 90 n = 4, day 270 n = 10; BU + PLX day 90 n = 4, day 270 n = 7. **d–g** Serial cytokine analysis performed on whole brain

lysates by 48-plex Luminex or ELISA assays (IL34 and SDF-1/CXCL12) at the depicted time points (**a**), n = 3 mice per time point. Cytokine quantifications and statistics are relative to untreated mice (*p-value vs U). The asterisk's color matches the cytokine legend. **b–g** Gray bar: PLX3397 administration window. **b–g** Data are Mean ± SD. Source data are provided as a Source Data file. Statistical analysis: *p < 0.05, **p < 0.01, ***p < 0.001, ****p < 0.0001, the exact p-values of all comparisons are reported in the Source Data file; **b** One-way ANOVA vs Untreated with Dunnett post-hoc. **c** Two-way ANOVA with Sidak post-hoc (BU vs. BU + PLX day 40, day 90, day 210); **d–g** Two-way ANOVA vs Untreated with Dunnett post-hoc. Figure 2a artwork was created with BioRender.com released under a Creative Commons Attribution-NonCommercial-NoDerivs 4.0 International license.

replacement by BM-derived RFP+ MGLCs in the brain, despite similar BM chimerism (Supplementary Fig. 1j–l).

Having confirmed that the best protocol combined BU-myeloablation with six days of post-transplant PLX administered via gavage, we focused on this regimen for subsequent analyses. We first examined the long-term persistence of transplant-derived cells by quantifying their engraftment in the CNS and hematopoietic organs seven months post-BMT. At this time, there remained stable engraftment of GFP + CD45 + CD11b+ MGLCs in BU + PLX-treated mice, which was significantly higher than in BU alone (87 ± 4% vs 23 ± 17%, Fig. 1d–e). Notably, the microglia niche was replaced specifically by BM-derived myeloid cells (CD45 + CD11b+) as we did not find increased frequencies of Ly6C+ cells, CD19 + B cells, and CD3 + T cells in BU + PLX-treated brains compared to untreated mice (Fig. 1f). This suggests that brain repopulation by BM-derived CD45 + CD11b+ cells is a regulated process and not an indiscriminate infiltration of immune cells as observed in several neurodegenerative and autoimmune diseases[64,65]. Histological analysis showed a widespread and homogeneous distribution of BM-derived GFP+ cells throughout the brain, spinal cord, and retina in mice treated with BU + PLX compared to those treated with BU alone (Fig. 1g–h and Supplementary Fig. 3).

### Fast replacement of microglia by BM-derived MGLCs coincides with brain-specific cytokine induction

A known limitation of HSCT for neurological indications is its delayed therapeutic effect[19–22,66]. Therefore, achieving fast repopulation of the CNS by hematopoietic cells would significantly improve HSCT's efficacy for neurological diseases with rapid progression. To examine the

kinetics of microglia depletion and brain repopulation, we looked at freshly isolated microglia preparations at 1, 4, 8, 20, 70 and 190 days after PLX withdrawal (corresponding to 21, 24, 28, 40, 90 and 210 post-BMT, respectively, Fig. 2a). Flow cytometry analyses showed acute and marked microglia depletion in the BU + PLX group (~90% at day 21 post-BMT, Fig. 2b). BU alone depleted the MG niche partially and at a much slower rate (~19% at day 40 and 50% at day 90 and 210, Fig. 2b). PLX withdrawal was followed by fast niche repopulation by GFP + CD45 + CD11b+ cells that peaked after 20 days (Day 40 post-transplant, Fig. 2b–c). MGLC engraftment was substantially slower in the BU-only condition, reaching 2.9% at 40 days post-BMT while the combination regimen achieved 93% (Day 40, Fig. 2c). Microglia replacement was stable in BU + PLX and was higher than that achieved with BU (86.8 ± 4% vs 23 ± 17%, day 210, Fig. 2c). Because BM is not always a clinical source of HSPCs and the timing of repopulation might differ between BM and purified HSPCs, we also examined the kinetics of brain repopulation using Lin- KIT + SCA-1+ (LKS) HSPCs. Time course analysis showed fast and lasting microglia repopulation using primitive LKS HSPCs with similar kinetics as BM following myeloablation with either BU or TBI (Supplementary Fig. 4).

Previous studies on cytokine secretion in the conditioned brain have focused on pro-inflammatory cytokines such as TNF-a, IL1-b, IL1-a, and known myeloid chemokines such as CCL2 (also known as mono-cyte chemoattractant protein 1, MCP-1), CCL5, and CXCL10[11,17,67]. These studies consistently show that irradiation stimulates more pro-inflammatory cytokines than busulfan. To elucidate mechanisms of recruitment and repopulation in the combined regimen, we measured a panel of 50 cytokines in the brain of BU + PLX-treated mice and

compared them to untreated mice (U). Cytokine quantification at 1, 4, and 8 days after PLX withdrawal showed brain-specific and transient increases in CSF1R ligands CSF1 and IL34 (Fig. 2d), presumably reflecting the depletion of CSF1R-expressing cells and signals that promote niche repopulation. The chemokines CCL2, CCL3 (also known as macrophage inflammatory protein 1-alpha, MIP-1a), CCL4 (MIP-1b), CCL7 (MCP-3), and CCL11 were also transiently elevated, which is consistent with their role in promoting the mobilization of myeloid cells (Fig. 2e–f). Interestingly, SDF-1 (also known as CXCL12), a potent chemoattractant for hematopoietic cells[68], was also increased in the brain (Fig. 2e). Most cytokines, except IL34, BAFF, and CXCL10, returned to baseline by day 8 (Fig. 2d–f and Supplementary Fig. 5a). The induction of CCL11, CCL7, and CXCL10 in the brain was not paralleled by increases in other pro-inflammatory cytokines normally co-induced during immune-derived inflammatory processes[69,70] (Fig. 2g). Notably, the cytokine elevations following PLX withdrawal were brain-specific and were not detected in plasma (Supplementary Fig. 5b–f). Overall, the pattern of cytokine induction differs from that observed in immune-derived inflammatory processes and suggests that myeloid proliferative and chemoattractant signals act locally to repopulate the depleted microglial niche.

## Hematopoietic reconstitution and neurobehavior following BU-myeloablation and short PLX3397 regimen

The impact of CSF1Ri on hematopoiesis is currently a topic of debate[34,71,72]. Some studies have observed significant changes in the frequency of hematopoietic lineages over time[34], while others have noted minor ones[73]. To better understand the impact of our 6-day PLX3397 regimen on hematopoietic reconstitution after transplantation, we compared the frequencies of hematopoietic progeny using lineage markers [CD11b (myeloid cells/macrophages), Ly6C (myeloid cells/monocytes/neutrophils), CD3 (T cells), and CD19 (B cells)] in mice that received BMT after conditioning with either BU or BU + PLX. As expected by the expression of CSF1R in CD11b+ and Ly6C+ subsets of myeloid cells, 24 hours after PLX withdrawal, we observed an acute reduction in the respective fraction of these circulating myeloid cells[74–76] with a compensatory increase in circulating B cells (Fig. 3a, Supplementary Fig. 6a). Among these circulating cells most were GFP + (Fig. 3b). There were small but statistically significant differences between BU and BU-PLX conditioned mice in the fraction of Ly6C + : 86% vs 74%, CSF1R + : 93% vs 84%, and CD19 + : 84 vs 89%, respectively (Fig. 3b). By distinguishing lymphocytes, monocytes, and granulocytes in peripheral blood by their relative size and granularity[77,78], we confirmed that all Ly6C+ cells were affected (Supplementary Fig. 6b–d). Seven months post-transplantation, the frequencies of hematopoietic lineages and the donor-derived chimerism in peripheral blood (PB), bone marrow (BM), and spleen (SP), were not different between BU + PLX and BU mice, except for a 13% decrease in Ly6C+ cells in BM (Fig. 3c–h). Time course analysis of hematopoietic reconstitution after transplant in BU + PLX conditioned mice showed full reconstitution of all lineages, including CD41+ platelets and Ly6G+ granulocytes by day 40 post-transplantation (Supplementary Fig. 6e–f).

We also examined the reconstitution of tissue macrophages in the heart, liver, lung, and peritoneum. Long-term engraftment of donor-derived GFP+ macrophages (MF) in these tissues had similar efficiencies in BU + PLX- and BU-treated mice (Fig. 3i, j), Kinetic analysis of the depletion and repopulation of peritoneal MF in BU + PLX-treated mice showed complete MF depletion after PLX (day 21) and complete repopulation three weeks after (day 40, Fig. 3k and Supplementary Fig. 7a). Although no differences were found in the long-term chimerism in tissue MF, PLX accelerated MF replacement whether administered pre- or post-transplant (Supplementary Fig. 7b–c).

To further document the potential toxicities of the developed BU + PLX regimen and the behavioral consequences of near-complete

microglia replacement with MGLCs, we performed serial observations of activity, survival, and a battery of neurobehavioral tests. Consistently with TGCT individuals on PLX3397[35], mice that received PLX3397 had well-demarcated patches of hair discoloration (Supplementary Fig. 7d). No differences were found between untreated, BU- and BU + PLX-conditioned and transplanted mice in survival or serial assessments of spontaneous locomotion (activity chamber), exploratory behavior (Y-maze), and spatial and recognition memory (novel place-novel object recognition) (Fig. 3l–p). Overall, we found that BU myeloablation combined with a short and controlled dosing of PLX3397, followed by transplantation, does not significantly impact mouse motor and cognitive function.

## MGLCs are heterogeneous and express microglia-specific genes

While transcriptional signatures that separate MGLCs and microglia have been described[18,79,80], the single-cell heterogeneity of MGLCs engrafted in the brain following conditioning with BU + PLX3397 has not been characterized. To examine this, we performed single-cell RNA sequencing (scRNA-seq) of FACS-sorted CD45 + CD11b+ cells isolated from mice that underwent transplantation with GFP+ HSPCs and BU + PLX conditioning, and from naive mice. We compared four samples: 1) GFP + CD45 + CD11b+ (MGLCs), 2) GFP- CD45 + CD11b+ (host conditioned MG or host MG), 3) GFP + CD45 + CD11b+ from BM (BM-CD11b + ), and 4) GFP + CD45 + CD11b+ MG from age-matched untreated donor mice (naive MG, Fig. 4a, and Supplementary Table 1 lists the number of cells analyzed). To compare tissue- and ontogeny-specific signatures and heterogeneity, all four populations were first integrated and analyzed as a single dataset. The analysis showed that MGLCs mostly overlapped with host MG and naive MG while BM-CD11b+ cells clustered separately (Fig. 4b). Differential gene expression analyses showed that the CD11b+ cells within the four samples could be represented in 19 subpopulations (clusters; Fig. 4c–d and Supplementary Fig. 8a–b). MGLCs separated into six main clusters: 0 (8.5% of cells), 1 (15.3%), 2 (7.2%), 3 (10.4%), 4 (14.1%), and 5 (29.6%). Smaller subpopulations represented by clusters 7, 12, 13, and 14 comprised between 1 and 4% of MGLCs. Notably, clusters 0 to 4 were shared with host MG and naive MG, comprising 87% and 91% of cells in these samples, respectively. Cluster 5 was exclusively present in MGLCs (Fig. 4d and Supplementary Fig. 8a–b). BM-CD11b+ cells separated primarily into six subpopulations (clusters 6–11) and did not significantly share cell subsets with the brain (Fig. 4d and Supplementary Fig. 8a–b).

To understand and define microglia and MGLC transcriptional states, we examined the expression of signature genes reported across multiple studies[28,81–87]. BM-derived cells that engrafted in the brain activated a transcriptional signature that was distinct from BM-CD11b+ (Fig. 4e). MGLCs expressed many classical homeostatic microglia genes like *Gpr34, Hexb, Olfml3, P2ry12/13, Siglech* and, *Tmem119* though the mean expression per cell and the fraction of positive cells were generally lower than in host and naive MG (Fig. 4e–f). Similarly to host and naive MG, MGLCs also expressed other microglia markers shared among cells in the myeloid/macrophage lineages like *Aif1, Csf1r, Trem2, Mertk* and *Cx3cr1*, in addition to activation genes (e.g., *C1qa, Tnf, Il1a* and *Cd68*, Fig. 4e). Ontogeny-specified microglia genes like *Sall1, Sall3*, and *Sparc* were expressed in most embryonic-derived host and naive MG (>90% *Sall1*+ cells) but only in a small fraction of MGLCs (3.6 ± 0.3% *Sall1* + MGLCs, Fig. 4e), a finding consistent with previous studies[28,86]. However, we found that MGLCs expressed *Irf8* and *Spi1/Pu.1*, transcription factors that specify microglia identity during embryonic development[86,88]. *Irf8* was specifically expressed in the brain (Fig. 4e), confirming its role as a master regulator of microglia and brain macrophages[86]. At the time of the analysis (9 months after transplant), MGLCs did not express genes involved in cycling or proliferation (2 ± 0.1% *Mki67* + MGLCs, Fig. 4e), suggesting cell expansion occurs early after PLX withdrawal. The gene expression signature that

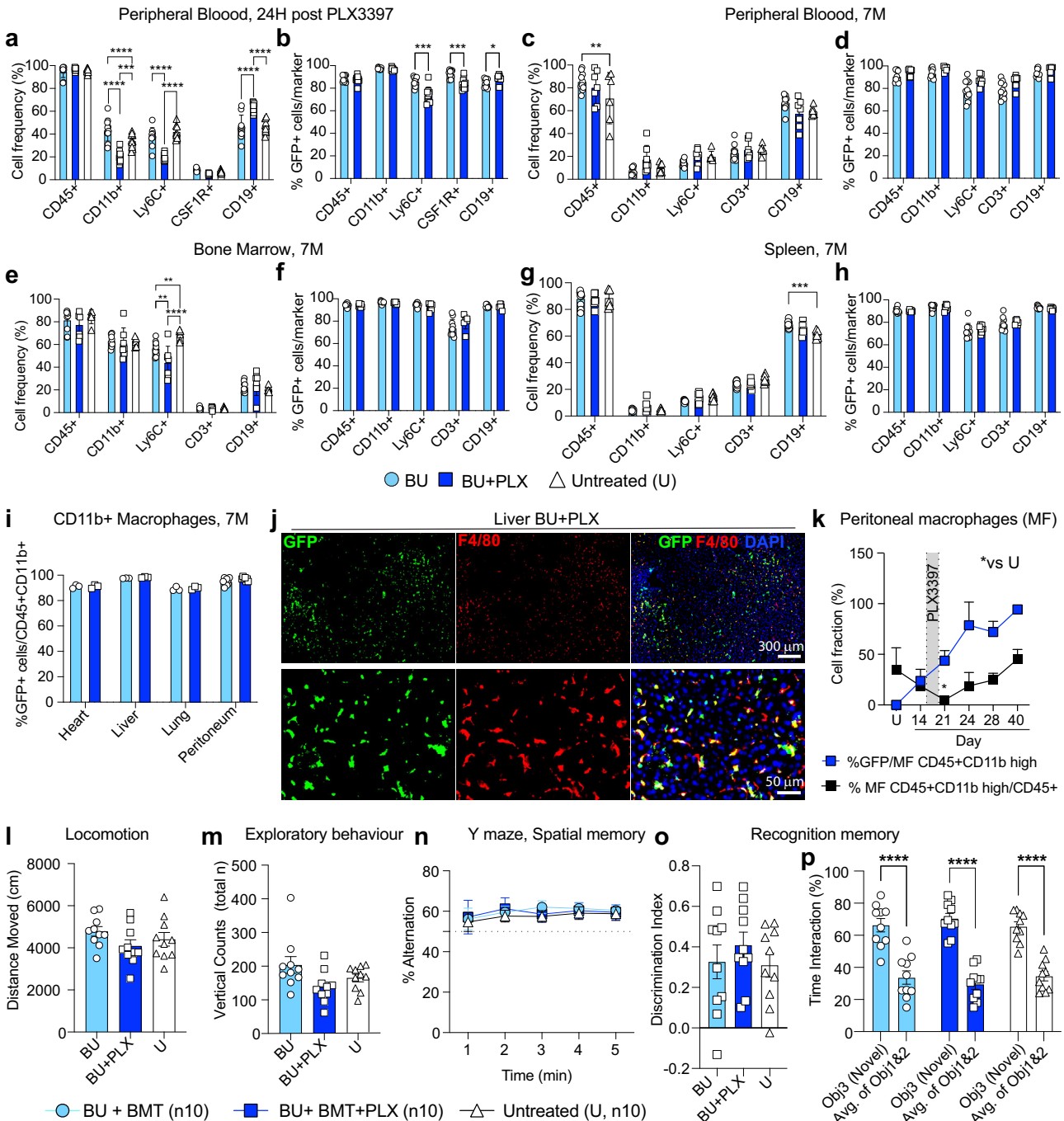

Fig. 3 | **Normal hematopoietic reconstitution and neurobehavior following busulfan myeloablation and a six-day PLX3397 regimen. a–b** Analysis of peripheral blood (PB) 24 hours after PLX withdrawal [equivalent to 21 days after bone marrow transplant (BM)]; BU n = 10, BU + PLX n = 10; Untreated n = 10. Analysis of PB **c**, **d**, bone marrow (**e–f**), and spleen (**g–h**) 7 months (7 M) after BMT; BU n = 10, BU + PLX n = 7, Untreated n = 6. **i** Donor macrophage chimerism in peripheral organs and peritoneum 7 M after BMT. Peritoneum BU n = 10, BU + PLX n = 7; Heart, liver, and lung n = 3 mice/cohort. **j** Representative images of GFP + F4/80+ macrophages repopulating the liver after BU + PLX (7 M post-BMT). Images representative of n = 3 mice. Scale bars are depicted. **k** Kinetics of host macrophage depletion and transplant-derived macrophage repopulation in the peritoneum measured by flow cytometry. The experimental timeline is depicted in Fig. 2a, n = 3 mice/cohort; *p-value vs untreated mice (U). **l–p** Behavioral analyses performed between 6 and 7 months post-BMT; BU n = 10, BU + PLX n = 10, Untreated n = 10. **l** Spontaneous

locomotion (activity chamber). **m** Exploratory behavior (activity chamber) reported by total vertical counts (periphery + center). **n** Spatial memory (Y-maze). **o–p** Recognition memory (Novel object recognition) reported by discrimination index and interaction time with Novel object (Obj 3) and previously encountered objects 1 and 2 (Avg. of Obj 1 & 2). **a–k** Data are Mean ± SD. **l–p** Data are Mean ± SE. **a–k, l–p** Source data are provided as a Source Data file. Statistical analysis: *$p < 0.05$, **$p < 0.01$, ***$p < 0.001$, ****$p < 0.0001$, the exact p-values of all comparisons are reported in the Source Data file; **a, c, e, g, l, n** Two-way ANOVA with Tukey post-hoc correction; **b, d, f, h, i** Two-tailed multiple unpaired t-test with Holm-Sidak post-hoc; **k** One-way ANOVA vs Untreated with Dunnett post-hoc; **m, o** Kruskal–Wallis with Dunn's post-hoc; **p** Two-way ANOVA with Tukey post-hoc (Obj3 vs treatment group; Avg. of Obj 1&2. vs treatment group) and two-way ANOVA with Sidak post-hoc (Obj3 vs Avg. of Obj. 1&2). GFP: green fluorescent protein; DAPI: 4′,6-diamidino-2-phenylindole.

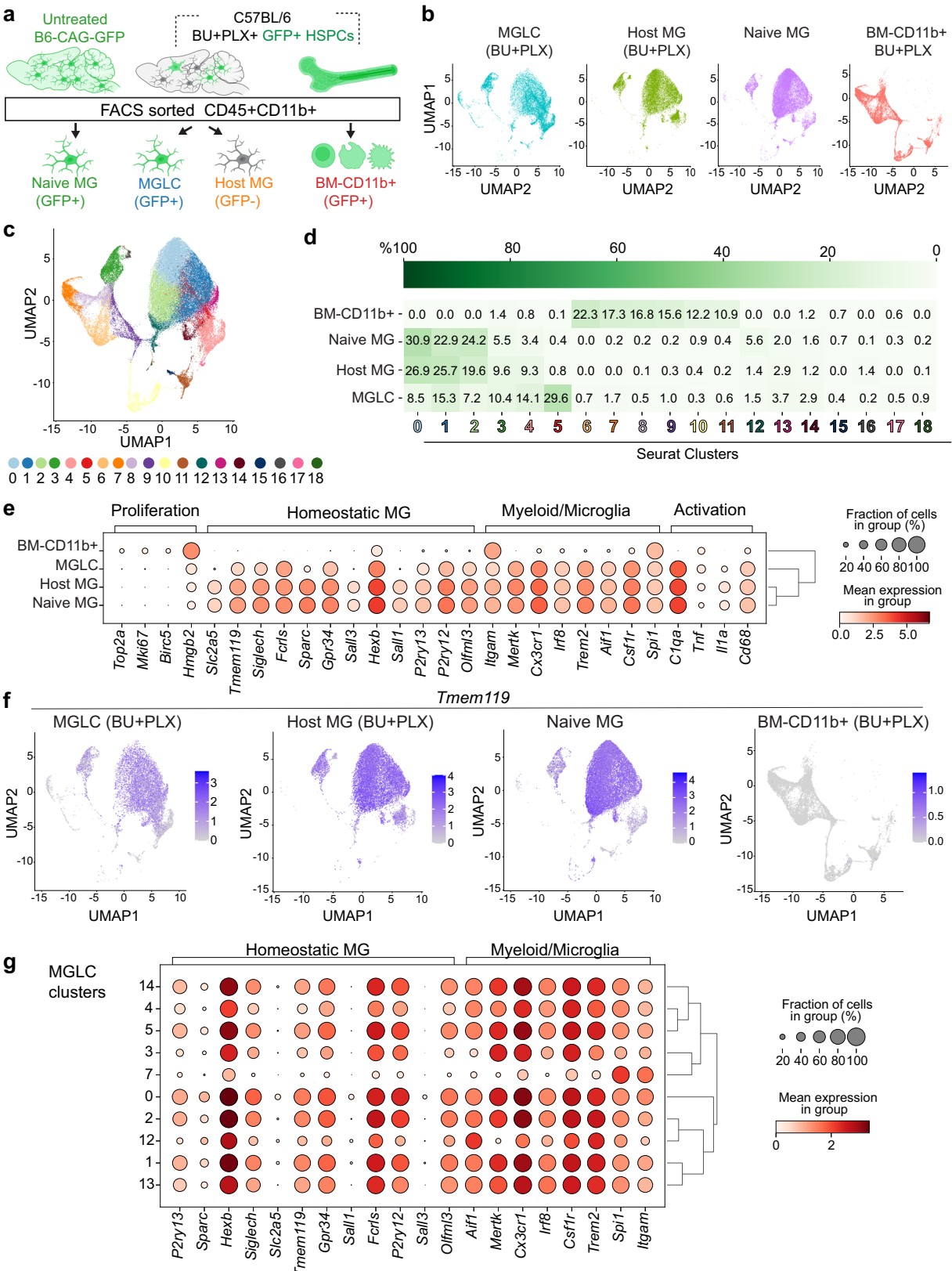

comprises the microglia sensome[82] was also expressed in MGLCs (Supplementary Fig. 8c).

Analysis of microglia gene expression in the identified clusters revealed transcriptional heterogeneity within the MGLCs (Fig. 4g and Supplementary Fig. 9a). The expression of several homeostatic microglia and myeloid/microglia genes was higher in MGLC clusters 0,

1, 2, and 5 than clusters 3, 4, 7 and 12 (Fig. 4g). Interestingly, cluster 5 (29.5% of MGLCs), which is unique to MGLCs, also expressed homeostatic microglia genes in a high fraction of cells (84 ± 2% of *Tmem119*+ cells, Fig. 4g). To identify the distinguishing characteristics of MGLC cluster 5, we performed differential gene expression analysis compared to MGLC clusters 0-4. The analysis showed 226 DEGs, most

**Fig. 4 | Single-cell transcriptional analyses of CD45 + CD11b+ cells from the brain and bone marrow show the heterogeneity of MGLCs and the activation of microglia genes in the brain. a** Experimental design. Single-cell RNA sequencing (scRNA-seq) was performed on FACS-sorted CD45 + CD11b+ cells isolated from mice that underwent busulfan plus PLX3397 (BU + PLX) conditioning and hematopoietic stem and progenitor cell (HSPC) transplantation as reported in Supplementary Fig. 4. Nine months after transplant (day 270), we compared four samples: 1) GFP + CD45 + CD11b+ (MGLCs, n = 3 mice), 2) GFP- CD45 + CD11b+ (host conditioned MG or host MG, n = 3 mice), 3) GFP + CD45 + CD11b+ from bone marrow (BM-CD11b + , n = 3 mice), and 4) GFP + CD45 + CD11b+ MG from untreated age- and sex-matched donor mice (naive MG, n = 3 mice). **b** Uniform Manifold Approximation and Projection (UMAP) showing the clustering of MGLC, host MG, naive MG, and BM-CD11b + . **c**, **d** Seurat cluster analyses of MGLCs, host-MG, naive MG, and BM-

CD11b+ cells. Each cluster is indicated by a color and a number. **d** Heatmap showing each sample's mean cell fraction per cluster (n = 3 mice/sample). **e** Dot plot showing the differential gene expression of microglia signature genes in each sample (n = 3 mice/sample). The dot size indicates the percentage of cells expressing the gene in each sample/cluster, while the color scale represents the mean gene expression calculated as the mean log-normalized UMI counts for each gene of interest. Dendrograms at the right show the clustering of the samples based on the expression profiles of the depicted genes. **f** UMAP showing the expression of the microglia-specific *Tmem119* gene in each sample (n = 3 mice/sample). **g** Dot plot showing the differential expression of microglia signature genes in the MGLC clusters (n = 3 mice/sample). Source data are provided as a Source Data file. Figure 4a artwork was created with BioRender.com released under a Creative Commons Attribution-NonCommercial-NoDerivs 4.0 International license.

downregulated and linked to immune pathways associated with different antigen responses (Supplementary Fig. 9b–c). Cluster 7 (1.7% of MGLCs) was the most divergent showing the lowest fraction of cells expressing *Irf8*, homeostatic microglia genes (e.g., *Siglech* and *Tmem119*), phagocytosis markers (*Mertk*), and several myeloid/microglia markers (e.g., *Trem2 and Csf1r*, Fig. 4g). Multi-gene cell annotation categorized 72 ± 4% of host, 84 ± 0.6% of naive, and 31 ± 4.2% of MGLCs as homeostatic. MGLCs had the largest proportion of cells classified as anti-inflammatory (51 ± 2 .7% vs 7.5 ± 1% in naive MG and 17 ± 4% in host MG, Supplementary Fig. 9d–e). Interestingly, cluster 5 was classified as anti-inflammatory (Supplementary Fig. 9d–e).

## MGLCs upregulate genes characteristic of brain border-associated macrophages (BAM)

The brain CD11b+ population also includes CNS-associated macrophages (CAMs) which comprise brain-border-associated macrophages (BAM) and perivascular macrophages[86,89–91]. Like microglia, BAMs derive from the yolk sac (except a subpopulation of choroid plexus BAM), rely on *Irf8* and *Spi1/Pu.1*, and are long-lived[86,89]. Compared to host and naive MG, MGLCs express many BAM genes at various levels, including MHCII genes (*H2-Aa/H2-Ab1/H2-Eb1*), *Mrc1* (or *Cd206*), *Cd74*, *Axl*, *Tgfbi*, *Ms4a6c*, *Dab2*, *Adgre1 (or F4/80)* and *Ccr1* (Fig. 5a–b). However, several other characteristic BAM genes like *Clec10a*, *Clec4n*, *Forl2*, *Lyve1*, and *Cd163* were not upregulated in MGLCs, suggesting a hybrid microglia/BAM transcriptional identity. Except for *Tgfbi* and *Ccr1*, BAM genes were not enriched in BM-CD11b+ cells, suggesting brain-specific upregulation (Fig. 5a–b). Based on the expression of MHCII genes, cluster 4 can be classified as MHC^high, clusters 12 and 13 as MHC^intermediate, while the remaining clusters, including the most abundant MGLC cluster 5, as MHC^low (Fig. 5b–c).

The transcriptional signature of naive BAM subpopulations isolated from the dura (D-BAM), subdural meninges (SD-BAM), and choroid plexus (CP-BAM) has been characterized by Van Hove et al. with scRNAseq analyses[86]. Most BAM-enriched genes reported by ref. 86 were expressed at higher levels and in a larger fraction of MGLCs compared to naive MG in bulk populations (Fig. 5d). Apart from *ApoE* and *Lyz2*, which were highly expressed in all MGLC subpopulations, BAM-enriched genes were found to be expressed at varied levels and combinations in MGLCs (Supplementary Fig. 10a). To evaluate any similarities between the described BAM and MGLC subpopulations, we compared their gene expression. The results showed that although several CP-BAM genes (such as *Hspa1a* and *Hspa1b*), and to a lesser extent, D-BAM genes were enriched in MGLCs, we did not identify a signature of a specific BAM subtype (Fig. 5e and Supplementary Fig. 10b–d). Notably, CP-BAMs include a small population of *Sall1*-expressing cells residing on the apical surface of the CP epithelium (CP^epi-BAMs)[86]. MGLC cluster 0, which contains the highest fraction of *Sall1*+ cells did not exhibit a CP^epi-BAM signature (Supplementary Fig. 10d). The expression of BAM master transcription factors in a higher fraction of MGLCs than naive MG (e.g., *Runx3* 30 ± 2% vs 2.7 ± 2% of cells, respectively, two-tailed

unpaired t-test *p*-value < 0.001) together with the reduced expression of *Sall1* in most MGLCs likely contributes to their hybrid transcriptional signature (Supplementary Fig. 10e–f).

Next, we evaluated signature proteins expressed in BAMs, microglia, hematopoietic, and activated/proinflammatory immune cells in the brain and BM cells using high-dimensional CyTOF mass cytometry. The full antibody panel is listed in Supplementary Table 2. To better discriminate the phenotype of MGCLs engrafting the brain after CSF1Ri, we compared brain myeloid cells (CD45 + CD11b+) from mice treated with either BU alone, BU + PLX, and naive MG. Intracellular staining with an anti-GFP antibody was used to distinguish host cells from transplant-derived GFP+ cells. The engraftment of GFP+ MGLCs measured by CyTOF matched that measured by standard flow cytometry (83 ± 10% BU + PLX vs 34 ± 14% BU alone, Supplementary Fig. 11a). Analyses of immune cell markers by CyTOF redemonstrated the lack of infiltration of proinflammatory cells in the brain of transplanted mice (Supplementary Fig. 11b). Compared to naive MG, the fraction of MHCII + MGLC (GFP + ) was highest in the MGLCs in mice conditioned with BU alone (naive 2.5 ± 0.5% vs BU 25 ± 6.5% vs BU + PLX 6 ± 0.7%, Fig. 5f–g). This difference may be attributed to the localization of most MGLCs engrafted in mice conditioned with BU alone to the choroid plexus (Fig. 1h), where MHCII^high macrophages are replenished by hematopoiesis-derived cells with a fast turnover[86]. The F4/80 surface marker (encoded by the BAM *Adgre1* gene) was expressed in a higher fraction of MGLCs from both BU + PLX and BU-treated mice compared to naive MG (16-20% vs 1.1 ± 0.6%, respectively, Fig. 5f–g). The myeloid/microglia marker CX3CR1 was expressed on the surface of most MGLCs (78 ± 13% BU, 86 ± 8.5% BU + PLX) similarly to naive MG (95 ± 1.6%, Fig. 5g–h) confirming the induction of *Cx3cr1* observed by scRNA-seq in MGLCs (Fig. 4e, g).

## MGLCs do not exhibit a disease-associated microglia signature

A microglial transcriptional state defined as disease-associated microglia (DAM) has been identified in the brains of an Alzheimer's Disease model[92], and core genes of this signature have been reported to be upregulated in other brain diseases[93,94]. However, several genes in the DAM signature, such as *ApoE*, *Lyz2*, *Axl*, and *Ifi27l2a*, are physiologically expressed in brain macrophages. These genes can also be upregulated in microglia in response to physiological stimuli, suggesting that some DAM genes reflect normal cellular functions[81,86,95]. We found a few DAM genes upregulated in MGLCs compared to naive MG (Fig. 6a). *ApoE* and *Lyz2*, were highly expressed in all MGLC subpopulations (Supplementary Fig. 12a). However, MGLCs did not express many other well-known DAM genes such as *Itgax* (CD11c), *Spp1* and *Clec7a*[92,93], among others (Fig. 6a). Based on studies showing physiological upregulation of core DAM genes in embryonic microglia[81], early postnatal microglia[81,95], and in adult BAMs in healthy mice[86], as well as studies reporting disease-associated signatures including disease-inflammatory macrophages (DIM)[96] and old, proinflammatory microglia[81], we curated a list of genes that we termed

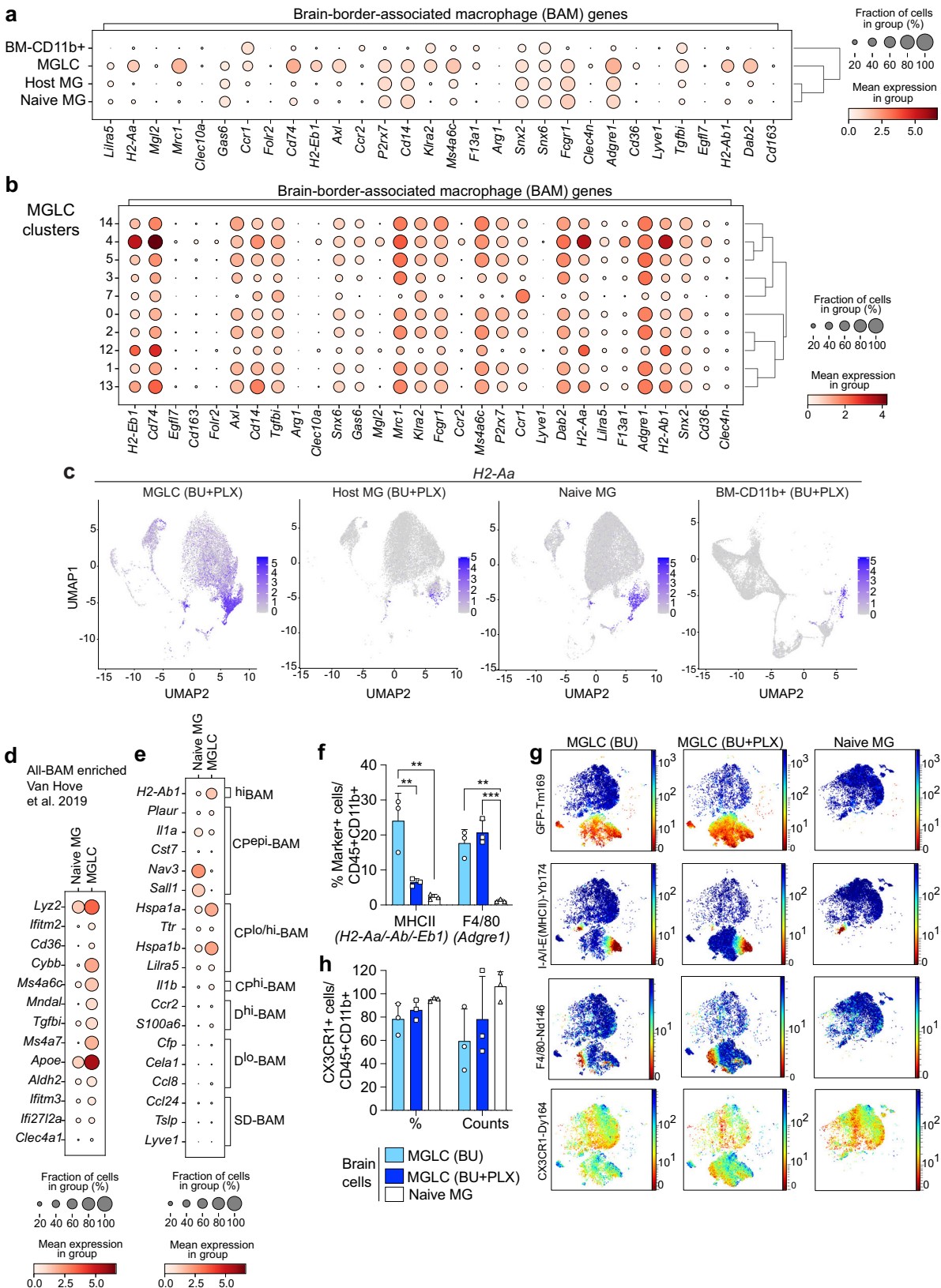

D-D-BAM (Development, Disease and BAM, Supplementary Fig. 12b–c) and evaluated their expression in MGLCs. Compared to naive MG, MGLCs consistently upregulated only genes that are physiologically induced in BAMs and embryonic microglia (*ApoE, Lyz2, Ms4a7, Clec12a, Ifitm3,* and *Igf1*[86], Supplementary Fig. 12b–c). Therefore, the BM-derived MGLCs engrafted in the brain have a hybrid transcriptional identity co-expressing homeostatic adult microglia genes, BAM, and embryonic microglia genes.

*ApoE* upregulation was also observed in host MG (2-fold increase vs naive MG, two-tailed unpaired t-test *p*-value < 0.001, Fig. 6a). To evaluate the state of the residual host MG, we performed differential gene expression analyses compared to naive MG. The analysis showed 1033

**Fig. 5 | MGLCs upregulate genes characteristic of brain-border-associated macrophages (BAM). a–e** Single-cell RNA sequencing (scRNA-seq) analyses of samples depicted in Fig. 4a (day 270 after transplant, n = 3 mice/sample). **a** Dot plot showing the expression of genes characteristic of brain-border-associated macrophages (BAMs) in each sample. Dot size: percentage of cells expressing the gene, color scale: mean gene expression calculated as the mean log-normalized UMI counts. Dendrograms show sample clustering. **b** Dot plot showing the expression of BAM genes in MGLCs. **c** Uniform Manifold Approximation and Projection (UMAP) showing the expression of the *H2-Aa* gene in each sample. Scale: mean gene expression (mean log-normalized UMI counts). **d** Dot plot showing the expression of BAM-enriched signature genes, reported by Van Hove et al. (GSE128855)[86], in the MGLCs and naive MG samples (GSE261246, this work). **e** Dot plot showing expression of selected top differentially expressed genes (DEGs) among BAM subpopulations in the MGLC and naive MG samples. Dura BAM: D-BAM; subdural meninges BAM: SD-BAM; choroid plexus BAM: CP-BAM. The top ten

DEGs in D-BAM, SD-BAM, and choroid CP-BAM were computed using the scRNA-seq data generated by Van Hove et al. (Macrophage aggregate, GSE128855)[86]. **f–h** Expression of BAM and myeloid markers evaluated by CyTOF mass cytometry in cells isolated seven months after bone marrow transplant (study depicted in Fig. 1 and Fig. 3). Groups: mice conditioned with either busulfan [MGLC (BU), n = 3] or BU + PLX3397 [MGLC (BU + PLX), n = 3] and untreated mice (naive MG, n = 3). **f** Percentage of CD45 + CD11b+ cells positive for the major histocompatibility complex (MHCII + ) and F4/80 markers (genes in brackets). **g** Optimized Stochastic Neighbor Embedding (Opt-SNE) plots showing the expression of GFP, MHCII, F4/80, and CX3CR1 in MGLCs and naive MG. **h** Fraction of CD45 + CD11b+ cells expressing CX3CR1 and CX3CR1 mean staining intensity/cell (counts). **f, h** Data are reported as Mean ± SD. Source data are provided as a Source Data file. Statistical analyses: **$p < 0.01$, ***$p < 0.001$, the exact p-values of all comparisons are reported in the Source Data file; one-way ANOVA with Tukey post-hoc for each marker.

DEGs (Fig. 6b). Pathway analysis on the upregulated and downregulated DEGs showed enrichment of pathways related to cellular energetic metabolism (e.g., oxidative phosphorylation), chemotherapy-induced reactive oxygen species, DNA damage, and neurodegeneration (Fig. 6b–c and Supplementary Fig. 12d–f). Differential gene expression analysis between all samples revealed the gene encoding for Growth hormone (*Gh*) as the top DEG in conditioned host MG (Fig. 6d), likely reflecting a busulfan-induced senescent phenotype[97].

A comprehensive analysis of cytokine genes showed no significant upregulation of other pro-inflammatory and senescent-associated cytokines in host MG (Fig. 6e and Supplementary Fig. 13a–b). MGLCs uniquely expressed the BAM cytokine gene *Pf4* (aka *CXCL4*, 64 ± 2% of MGLCs vs 5.6 ± 0.5% of naive MG, two-tailed unpaired t-test p-value < 0.0001) and had a higher fraction of cells expressing the anti-inflammatory *Cxcl16* gene[98] (64 ± 6% of MGLCs vs 30 ± 3% of naive MG, two-tailed unpaired t-test p-value < 0.001; Fig. 6e and Supplementary Fig. 13a–b). While we cannot exclude the possibility that the enzymatic brain dissociation process activated some cytokine genes, we observed a similar expression of immediate-early genes in all samples (IEGs, Supplementary Fig. 13c)[83,95], supporting the validity of this comparison. Analysis of cytokine and chemokine receptors revealed *Ccr1* and *Cxcr4* as two additional receptors that distinguish most MGLCs from host and naive MG upon BU + PLX conditioning (Fig. 6e and Supplementary Fig. 13d–e).

Genes primarily expressed in monocyte and dendritic cells[86,95] were not expressed in MGLCs (Supplementary Fig. 14a–b). Compared to naive MG, only *Itgal* was expressed in a higher fraction of MGLCs (17 ± 5% vs 2.3 ± 1% respectively; two-tailed unpaired t-test p-value = 0.006) and was primarily found in cluster 7 (77 ± 3.4% of cells), the most divergent MGLC subpopulation (Supplementary Fig. 14a–b). Analyses of markers upregulated in brain monocyte-derived cells (MdCs)[99], such as CD64[99] and CD86[99], using high-dimensional CyTOF mass cytometry, showed that their expression is highest in the BM-CD11b+ cells (Fig. 6f). A higher percentage of MGLCs expressed CD64 compared to naive MG (16% vs 3.7%, respectively, Fig. 6f), while no significant differences were found in the expression of CD86 (Fig. 6f). We also examined CD169 (aka siglec1) and MAC-2/LGALS3 (aka galectin 3) as they are expressed in BAM and MdCs in the adult brain[99] but not in healthy microglia[81,92–94,100]. While the fraction of MAC-2 + CD45 + CD11b+ cells in the brain did not differ, CD169 stained most MGLCs and BM-CD11b+ cells but not naive MG, thereby constituting another MGLC-specific surface marker (Fig. 6f–g). Analysis of genes expressed in HSPCs were mostly absent in MGLCs, host, and naive MG except for *Cd48* and *Cd34* (Supplementary Fig. 14c–d).

Taken together, MGLCs engrafted in the brain of BU + PLX-conditioned mice exhibit a mixed transcriptional signature, expressing genes characteristic of homeostatic microglia, BAMs, and embryonic microglia. MGLC subpopulations can be distinguished based on the expression levels of specific gene sets. Additionally, we have identified

several markers, CD169, CCR1, CXCR4, and CXCL4 (*Pf4*) that can differentiate MGLCs from host and naive MG.

## Busulfan plus PLX3397 is more effective than busulfan alone in reconstituting brain progranulin and correcting lipofuscinosis in a mouse model of CLN11/FTD

BMT/HSCT with a conditioning regimen that combines BU and CSF1Ri could be a promising treatment for progranulin (GRN) deficiency. This is because microglia and MGLCs express and secrete high levels of GRN, which can cross-correct other CNS cells (Fig. 7a–b). Moreover, the symptoms of severe GRN deficiency (CLN11) affect the CNS and retina, where high replacement of dysfunctional GRN-deficient microglia can be achieved. However, in a previous study, mice conditioned with TBI and BMT showed only partial efficacy, likely due to the low recruitment of hematopoietic cells to the brain[101]. To evaluate the effectiveness of the optimized BU + PLX conditioning in comparison with BU alone, we transplanted 2-month-old *Grn*$^{-/-}$ mice[102] with wild-type BM from CAG-GFP mice (WT BMT, Fig. 7c). Conditioned *Grn*$^{-/-}$ mice transplanted with BM from *Grn*$^{-/-}$ mice (KO) were used as sham controls. Four months after transplant, the donor chimerism in BM and PB was similarly high after BU and BU + PLX conditioning (>80%, Fig. 7d) but differed slightly in the spleen (70% vs 85%, respectively, Fig. 7d). The myeloid (CD11b + ) and B-cell compartments (CD19 + ) were mostly donor-derived while T (CD3 + ) and Ly6C+ cells had a higher contribution of host-derived cells (Fig. 7e–g). Compared to BU alone, PLX improved T cell chimerism in the BM and spleen while it decreased Ly6C+ chimerism in PB (Fig. 7e–g). Following BU or BU + PLX treatment and WT BMT, serum GRN levels ranged from 16% to 51% of wild-type levels, as measured by ELISA and Western blot, and there was no significant difference between the two conditions (Fig. 7h–j).

Consistent with our observations in wild-type mice, brain chimerism was significantly higher in *Grn*$^{-/-}$ mice conditioned with BU + PLX than in those conditioned with BU alone (64 ± 13% vs 7 ± 3%, respectively, Fig. 7k). Notably, despite similar amounts of GRN in the circulation, GRN was only detected in the brain of *Grn*$^{-/-}$ mice conditioned with BU + PLX by both ELISA and Western blot (Fig. 7l–m and Supplementary Fig. 15a). Brain GRN was reconstituted between 23% to 33% of wild-type levels on average when using BU + PLX and WT BMT (Fig. 7l–m).

Studies on the brain histopathology of GRN-deficient mice have shown a late-onset appearance of several disease markers, such as microgliosis and astrogliosis (≥9–12 months of age)[44,56,102–105]. However, lipofuscin storage in the CA3 region of the hippocampus, a well-known disease biomarker, has been detected in both young and aged GRN-deficient mice[105–108]. We confirmed a significant increase in CA3 lipofuscin deposits in *Grn*$^{-/-}$ compared to WT mice at six months of age (Fig. 7n–o). Notably, lipofuscin storage was cleared in *Grn*$^{-/-}$ treated with BU + PLX and WT BMT, while no significant decrease was observed in *Grn*$^{-/-}$ mice after BU conditioning and WT BMT (Fig. 7n–o).

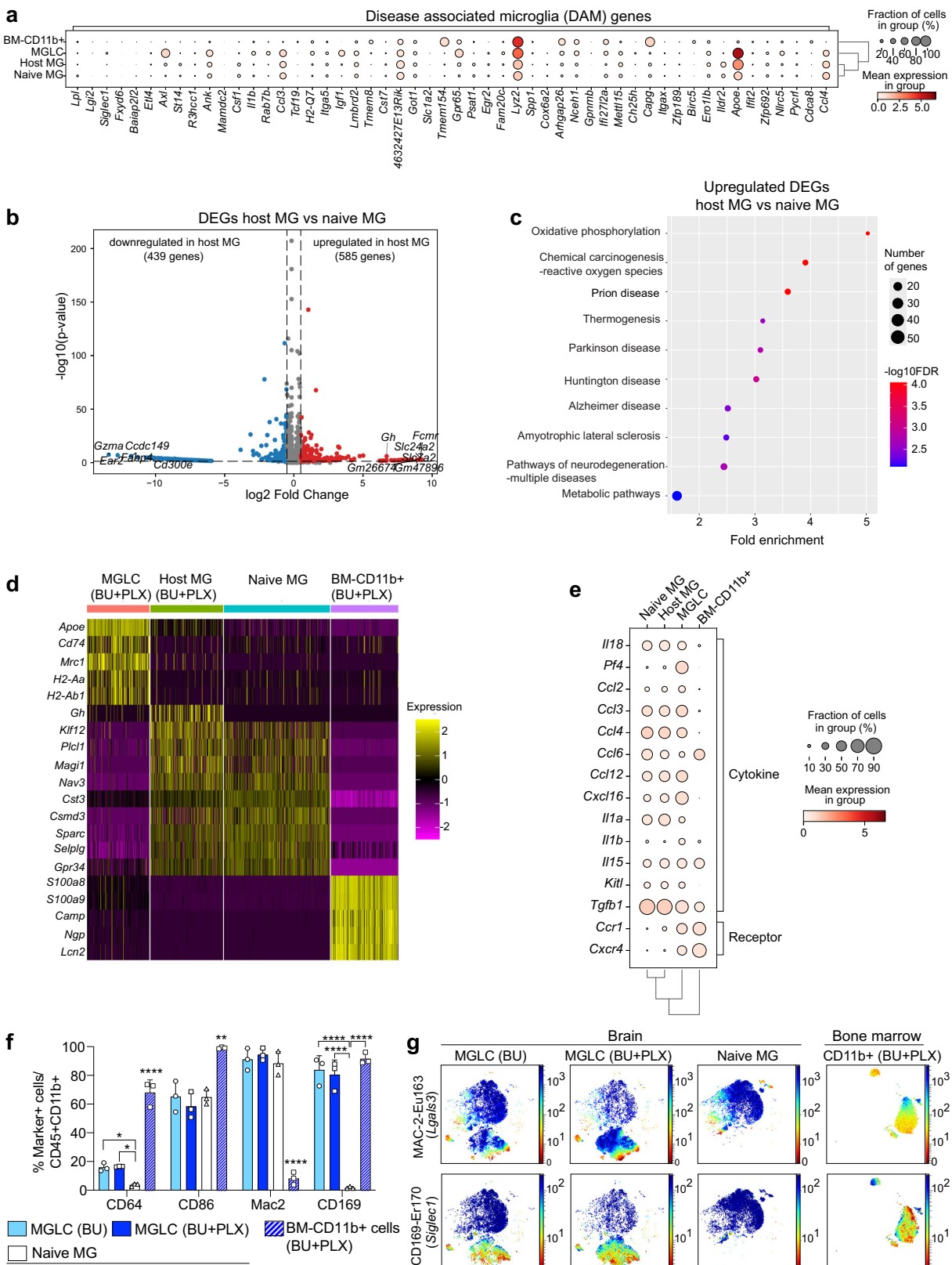

**Busulfan with PLX3397 improves lipid metabolism and proteostasis in a mouse model of CLN11/FTD**

We focused on the BU + PLX regimen to assess additional therapeutic outcomes due to the superior replacement of microglia, restoration of GRN, and clearance of lipofuscin achieved compared to BU alone. Consistent with observations in the brain, analyses of GRN

expression in the eye also demonstrated biochemical correction (23–32% of WT, Fig. 8a–c). Proteostasis defects such as altered maturation of lysosomal proteases and accumulation of ubiquitinated proteins have been reported in young and aged $Grn^{-/-}$ mice and in FTD-GRN individuals[54,109–111]. We confirmed an increased storage of ubiquitinated proteins and altered Cathepsin D (CTSD) maturation in

**Fig. 6 | Lack of DAM signature in MGLCs and long-term alterations in host microglia induced by busulfan. a–e** Single-cell RNA sequencing (scRNA-seq) analyses for samples depicted in Fig. 4a (day 270 after transplant, n = 3 mice/sample). **a** Dot plot showing the expression of disease-associated microglia (DAM) genes (≥ 3-fold upregulation in DAM vs naive microglia, Keren-Shaul, et al.[92]) in each sample. **b** Volcano plot showing the differentially expressed genes (DEGs) in host MG vs naive MG. The top 5 downregulated and upregulated gene names and *Growth hormone* (Gh) are indicated. **c** Pathway enrichment analyses (top ten) based on DEGs upregulated in host MG vs naive MG. Dot size: number of genes in each pathway; color scale: -log10 False Discovery Rate (FDR). FDR lower than 0.05 is considered statistically significant. **d** Heatmap showing top 5 DEGs between all samples. Scale = z-score. Each subcolumn depicts a single cell. **e** Dot plot showing the expression of genes encoding for cytokine and cytokine receptors in each sample. **f–g** Expression of brain-infiltrating monocyte-derived cells (MdCs) and brain-associate macrophages (BAMs) markers by high-dimensional CyTOF mass cytometry in cells isolated from brain and bone marrow (BM) of mice 7 months after bone marrow transplant. Groups: mice conditioned with either BU [MGLC (BU), n = 3] or BU + PLX3397 [MGLC (BU + PLX), n = 3 and BM-CD11b+ (BU + PLX), n = 3], untreated mice (naive MG, n = 3 mice). **f** Percentage of CD45 + CD11b+ cells positive for CD64, CD86, MAC-2, and CD169. **g** Optimized Stochastic Neighbor Embedding (Opt-SNE) plots showing the expression of MAC-2 and CD169 (gene names in brackets). **f** Data are reported as Mean ± SD. Source data are provided as a Source Data file. Statistical analyses: *p < 0.05, **p < 0.01, ***p < 0.0001, the exact p-values of all comparisons are reported in the Source Data file; **b** Wald with Benjamini–Hochberg post-hoc; **f** One-way ANOVA with Tukey post-hoc for each marker. Asterisks without comparison lines indicate a statistically significant difference between all groups/markers.

brain lysates of *Grn*$^{-/-}$ mice compared to WT (Fig. 8d–f). Both defects were normalized in *Grn*$^{-/-}$ mice treated with BU + PLX and WT BMT (Fig. 8d–f).

Loss of GRN results in decreased lysosomal Bis(Monoacylglycero) Phosphates (BMPs), regulatory lipids required for the activation of lysosomal hydrolytic enzymes involved in the catabolism of sphingolipids such as gangliosides[44,112]. This can lead to ganglioside accumulation in lysosomes, likely contributing to the neuroinflammation and neurodegeneration observed in CLN11 and FTD-GRN. To evaluate the correction of BMP deficiency, we measured them in the frontal cerebral cortex of *Grn*$^{-/-}$ mice treated with BU + PLX and WT BMT. Mass-spectrometry analyses confirmed a significant decrease in BMP levels in both sham and untreated *Grn*$^{-/-}$ mice compared to WT (Fig. 8g–h). Most notably, BMP lipid metabolism was normalized in *Grn*$^{-/-}$ mice receiving BU + PLX and WT BMT (Fig. 8g–h). These findings suggest that even partial reconstitution of GRN in the frontal cerebral cortex by MGLCs can restore lipid metabolism and has therapeutic benefits (Fig. 8i, Supplementary Fig. 15b–d).

To provide further evidence of phenotypic correction, we reviewed the literature to determine the deficits, timing, and magnitude of the behavioral phenotype in *Grn*$^{-/-}$ mice[103,105,108,110,113,114]. Since the reported neurobehavioral phenotypes *of Grn*$^{-/-}$ mice were variable, we performed serial neurobehavioral studies of *Grn*$^{-/-}$ and wild-type mice every two months for 12 months. Analyses of survival, spontaneous locomotion/exploratory behavior (Activity chamber), sociability (3-chamber sociability test), spatial memory (Y-maze spontaneous alternation), associative memory (passive avoidance test), cognitive deficit (nesting), and obsessive-compulsive behavior (grooming) showed no differences between wild-type and *Grn*$^{-/-}$ mice and could not be used as e read-outs of therapeutic efficacy (Supplementary Fig. 16).

In conclusion, our optimized BU + PLX conditioning regimen leads to CNS-wide engraftment of MGLCs, which can restore therapeutic levels of GRN in the brain and eyes of *Grn*$^{-/-}$ mice. Partial GRN restoration in the brain is sufficient to clear pathological lipofuscin, BMP deficiency, as well as ubiquitination and CTSD maturation defects in *Grn*$^{-/-}$ mice.

## Discussion

Correcting protein deficiencies in the CNS remains a significant obstacle in treating neurological diseases due to the BBB. HSCT offers a promising solution by delivering therapeutic proteins to the CNS through donor-derived hematopoietic cells that naturally home as MGLCs after conditioning the CNS. MGLC engraftment has also been documented in humans who underwent myeloablative conditioning and allogenic HSCT for cancer treatment[13,79] or Metachromatic leukodystrophy[66,115,116]. Interestingly, microglia replacement appears to be a normal physiological process that is commandeered after conditioning. Recent reports show that bone marrow-derived myeloid cells, resembling microglia, are protective in Alzheimer's disease patients[117]. However, despite the therapeutic implications for a pathway that allows the controlled entry of myeloid cells into the brain, little is known about the mechanisms involved in the migration and differentiation of hematopoietic-derived MGLCs in the CNS.

The low, variable, and slow-paced engraftment of MGLCs in the CNS following standard myeloablation[17–22,66] has been a key limitation of HSCT and has prompted the use of CSF1Ri to enhance microglia depletion and cell recruitment from the BM[18,23,24,32,33,59,118]. Here, we report a highly efficient conditioning regimen based on BU and the shortest post-transplant course of CSF1Ri reported to date that results in reproducibly high (≥90%) and stable replacement of microglia by MGLCs throughout the CNS, including the neuroretina and the spinal cord. This protocol has, in principle, clinical translatability. BU is the clinical agent for neurometabolic indications of HSCT, while PLX3397 is the only FDA-approved CS1R inhibitor with available safety data. The PLX3397 dose used in our study is well below what is prescribed for patients with TGCT (humans take ~400–800 mg/day orally for repeated cycles of 4 weeks[35], while mice take ~3 mg/day for six days). While there is no definitive evidence that PLX3397 administration depletes microglia in humans, several observations make it likely. It is well-established that it depletes CSF1R-expressing macrophages in humans and can enter the blood-brain barrier, as demonstrated in animal studies and clinical trials in patients with glioblastoma[35,119]. Additionally, individuals with genetic mutations that abolish or reduce CSF1R expression show either depleted or significantly reduced microglia[25–27]. Finally, since whole bone marrow is infrequently used as a cell source in the clinic and isolated CD34+ HSCPs are more prevalent, we also demonstrate that isolated, primitive HSPCs repopulate the CNS within a comparable timeframe as whole BM. This optimized conditioning protocol can potentially improve the efficacy of allogenic HSCT and autologous transplantation of genetically corrected HSPCs and could expand HSCT's application to more neurodegenerative disorders.

We also demonstrate that the timing of PLX3397 administration is key to maximizing MGLC chimerism. The highest level of microglia replacement is observed when PLX3397 is administered after busulfan and 15 days post-transplantation, suggesting that maximum replacement requires full conditioning of the brain, a process that combines the myeloablative effect of BU and microglia depletion of CSF1Ri. The necessity for the pre-conditioning effect that BU has in the CNS is demonstrated by the lack of MGLC engraftment with a regimen that combines PLX3397 with non-CNS penetrant treosulfan-based myeloablation[10]. How BU pre-conditions the CNS is poorly understood, but we and others showed that it does not cause overt BBB disruption[17,120]. At similar myeloablative doses, BU alters microglia morphology and induces the expression of senescence markers while reducing their regenerative capacity[33]. Consistent with this, the optimal timing of PLX3397 administration coincides with the peak of BU-induced microglia senescence reported by ref. 33. The data suggest that combining microglia depletion with a proliferative impairment of the microglia niche enables the competitive engraftment of

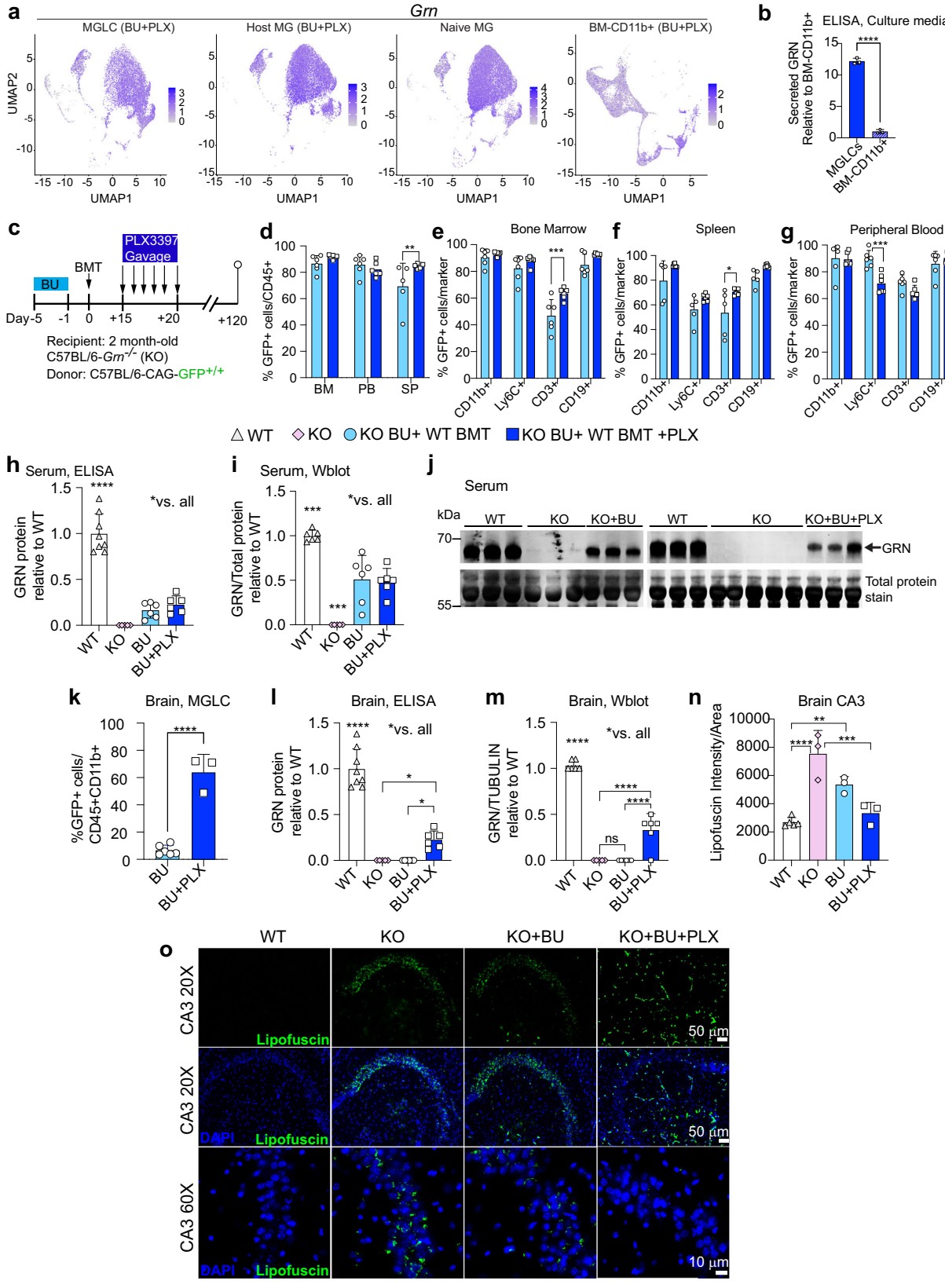

hematopoietic-derived cells into the brain[23,33]. Nevertheless, our findings show that administering PLX3397 before transplantation is superior to using BU alone, even though it is less effective. The pretransplant PLX3397 regimen could be preferable for diseases where lower than 80–90% microglia replacement is sufficient to achieve a therapeutic effect.

Another limitation of HSCT with standard conditioning is its slow repopulation kinetics and delayed therapeutic onset in humans and mice[17–22,66]. This is particularly problematic for fast-progressing neurological diseases like infantile-onset Krabbe and other sphingolipidosis[19,20,22]. Using standard conditioning protocols, it takes 45 days for MGLCs to be detected in the mouse brain after

**Fig. 7 | Busulfan with PLX3397 is more effective than busulfan alone in reconstituting progranulin and correcting lipofuscinosis in the brain of a mouse model of CLN11/FTD. a** Uniform Manifold Approximation and Projection (UMAP) showing *Grn* mRNA levels in MGLCs, host MG, naive MG, and BM-CD11b+ populations (scRNA-seq study in Fig. 4a, n = 3 mice/sample). **b** GRN secretion by ELISA in the media of MGLCs and BM-CD11b+ isolated from mice conditioned with BU + PLX and cultured side-by-side (study in Fig. 4a, n = 3 mice/sample). **c** Experimental timeline: C57BL/6-*Grn*[-/-] (KO) mice conditioned with busulfan (BU) or BU + PLX3397 (PLX) and transplanted with bone marrow from C57BL/6-CAG-GFP (WT BMT). Comparisons: wild-type C57BL/6 mice (WT, n = 8); disease controls, KO untreated (U, n = 6) or KO conditioned with BU and transplanted with KO BMT (Sham, n = 5); KO conditioned with BU and transplanted with WT BMT (BU, n = 6); KO conditioned with BU + PLX and transplanted with WT BMT (BU + PLX, n = 6). **d–g** Percentage of transplant-derived cells in hematopoietic compartments (BU and BU + PLX n = 6 mice/group). **h** GRN quantification by ELISA in serum [WT n = 8, KO n = 5, BU n = 6, BU + PLX n = 6]. **i–j** GRN quantification by Western blot in serum [WT n = 6, KO n = 5, BU n = 6, BU + PLX n = 6]. Uncropped blots are included in the Source Data file. **k** Percentage of transplant-derived MGLCs in the brain of KO mice (BU n = 6, BU + PLX n = 3). **l** GRN quantification by ELISA in brain [WT n = 8, KO n = 6, BU n = 6, BU + PLX n = 6. **m** GRN quantification by Western blot in brain [WT n = 6, KO n = 5, BU n = 6, BU + PLX n = 6]. Uncropped blots are in the Source Data file. **n** Quantification of lipofuscinosis in the CA3 region (WT n = 5, KO n = 3, BU n = 3, BU + PLX n = 3). **o** Representative images of lipofuscinosis in the CA3 region (representative of n = 5 WT, n = 3 KO, n = 3 BU, n = 3 BU + PLX). Scale bars are depicted. **b, d–i, k–n** Data are Mean ± SD. Source data are provided as a Source Data file. Statistical analysis: **p < 0.01, ***p < 0.001, ****p < 0.0001, the exact p-values of all comparisons are reported in the Source Data file; **b, k** Two-tailed unpaired t-test; **d–i, l–n** One-way ANOVA with Tukey post-hoc.

transplantation, and it can take months for them to replace a small fraction of the microglia niche[18]. Our studies show that adding CSF1Ri leads to much faster microglia replacement by MGLCs. In mice treated with the combination regimen, repopulation surpassed BU alone on day 21 (17 days post-HSCT) and achieved peak repopulation (~90%) by day 40 post-HSCT, which remained stable after nine months. This rapid microglia replacement allows for quicker delivery of therapeutic proteins and may prevent disease progression.

Post-transplant administration of PLX3397 at a 100 mg/kg/day dose for six days did not cause additional toxicities in mice conditioned with BU. These mice exhibited the same survival, spontaneous locomotion, exploratory behavior, and spatial and recognition memory as mice conditioned with BU alone. Despite reports on long-term effects of CSF1Ri on hematopoietic lineages (particularly for PLX5622)[34,72] and the reported lack of receptor selectivity of PLX3397[35], we did not observe significant adverse effects on hematopoietic reconstitution after a short course of PLX3397. Although PLX3397 inhibited KIT and FTL-3 receptors in vitro[121], our data suggest a lack of functional impact in vivo in the transplantation setting. As both KIT and FLT-3 are required for hematopoietic stem cell (HSC) survival and differentiation, and anti-KIT therapies are being used to deplete HSCs[122,123], the unaffected hematopoiesis and the lack of engraftment of donor cells in mice treated with PLX3397 alone suggest normal receptor function in HSCs. Ultimately, the inhibitor used, its receptor specificity, the total dose, the route of administration, and the treatment course may account for discrepancies in toxicity, highlighting the need to limit dose and exposure time.

In addition to MG replacement, there is significant therapeutic potential for replacing tissue macrophages (MF) and using these cells to deliver therapeutic proteins to tissues such as the heart, lungs, and liver. In theory, a conditioning protocol that includes CSF1Ri could deplete tissue MF and improve the replacement of these cells after transplantation. By investigating the reconstitution of tissue MFs in various organs, we found that while the total engraftment of donor-derived MF was similar in mice treated with BU-PLX and BU alone, the addition of PLX3397 accelerated their repopulation. Interestingly, PLX3397 speeded MF replacement, whether administered post-transplant or concurrently with myeloablation. This suggests that recruitment mechanisms differ between the CNS and periphery and tissue MF replacement protocols could be further optimized for reduced toxicity.

While transcriptome analyses have reported differences in gene expression between MGLCs and developmentally specified microglia[18,79,80], the single-cell heterogeneity of MGLCs following conditioning with BU + PLX3397 had not been appreciated. Consistent with previous studies, we found that BM-derived cells that engraft in the brain activated a transcriptional signature that was quite distinct from BM-CD11b+ cells and found no shared cell populations between cells in the BM and MGLCs, supporting the idea that environmental signals have a major role in dictating cell fate[28]. The MGLC population

can be categorized into six main subpopulations based on differential gene expression. Notably, most MGLCs could be categorized as homeostatic (31% of cells) and anti-inflammatory (51% of cells). We show that MGLCs acquire a hybrid homeostatic microglia/BAM/embryonic microglia signature. Most MGLC subpopulations express genes commonly enriched in BAMs. While MGLCs cannot be categorized into specific BAM subtypes[86], they shared more genes with CP-BAM and D-BAM than SD-BAM. Notably, although the BAM-characteristic MHCII genes were induced in all MGLC subpopulations, the expression of MHCII on MGLCs was higher following conditioning with BU alone. This finding may be attributed to the rapid turnover of hematopoietic-derived MHC[high] macrophages in the choroid plexus[84], where most BU-induced MGLCs localize. Another possibility is that MHCII[high] MGLCs are induced in response to the high fraction of senescent host microglia that persists in the brain following BU if not cleared by PLX3397[33]. All MGLC populations showed increased expression of *ApoE* and *Lyz2*, which are increased in both BAMs and embryonic E14.5 microglia[81,86].

As observed in other studies, MGLCs typically exhibit minimal or no expression of the ontogeny-specified transcriptional repressors *Sall1* and *Sall3*, which may contribute to their hybrid transcriptional state. However, we identified a small fraction of MGLCs expressing *Sall1* (3.6%). These cells are primarily found in MGLC cluster 0 and co-express genes typical of BAMs and homeostatic microglia. Cluster 0 also expressed genes enriched in CP[epi]-BAMs, which normally express *Sall1*. Whether *Sall1* + MGLCs localize to the apical surface of the CP epithelium like CP[epi]-BAMs remains to be investigated. We also report a specific induction of the *Irf8*, a master transcriptional regulator of microglia and macrophages in the brain, which, together with *Pu.1*, specifies the identity of microglia cells during embryonic development[86,88]. A fraction of MGLCs also showed increased expression of BAM master transcription factors *Runx3* and *Bcl3*, further supporting the mixed BAM-microglia gene signature. The hybrid transcriptional profile likely reflects differences in chromatin accessibility at these various loci established earlier during the development of microglia and the hemopoietic cells that repopulate the CNS postnatally. Although the precise functional consequences of this hybrid phenotype need investigation, our behavioral data, the long-term outcomes of patients who have undergone transplantation, and recent studies reporting the presence of bone-marrow-derived myeloid cells in the aging and AD brain[117], suggest MGLCs can functionally replace microglia.

The signaling pathways involved in the migration, recruitment, and differentiation of hematopoietic-derived MGLCs in the CNS were unknown. Here, we report several cytokines induced in the brain early after CSF1Ri. CSF1R ligands CSF1 and IL34 were transiently increased and may signal an empty microglia niche to induce the proliferation of the endogenous repopulating pool. CSF1 is primarily expressed in astrocytes, oligodendrocytes, and microglia, while IL34 in neurons[124,125]. Several myeloid chemokines (CCL2, CCL3,

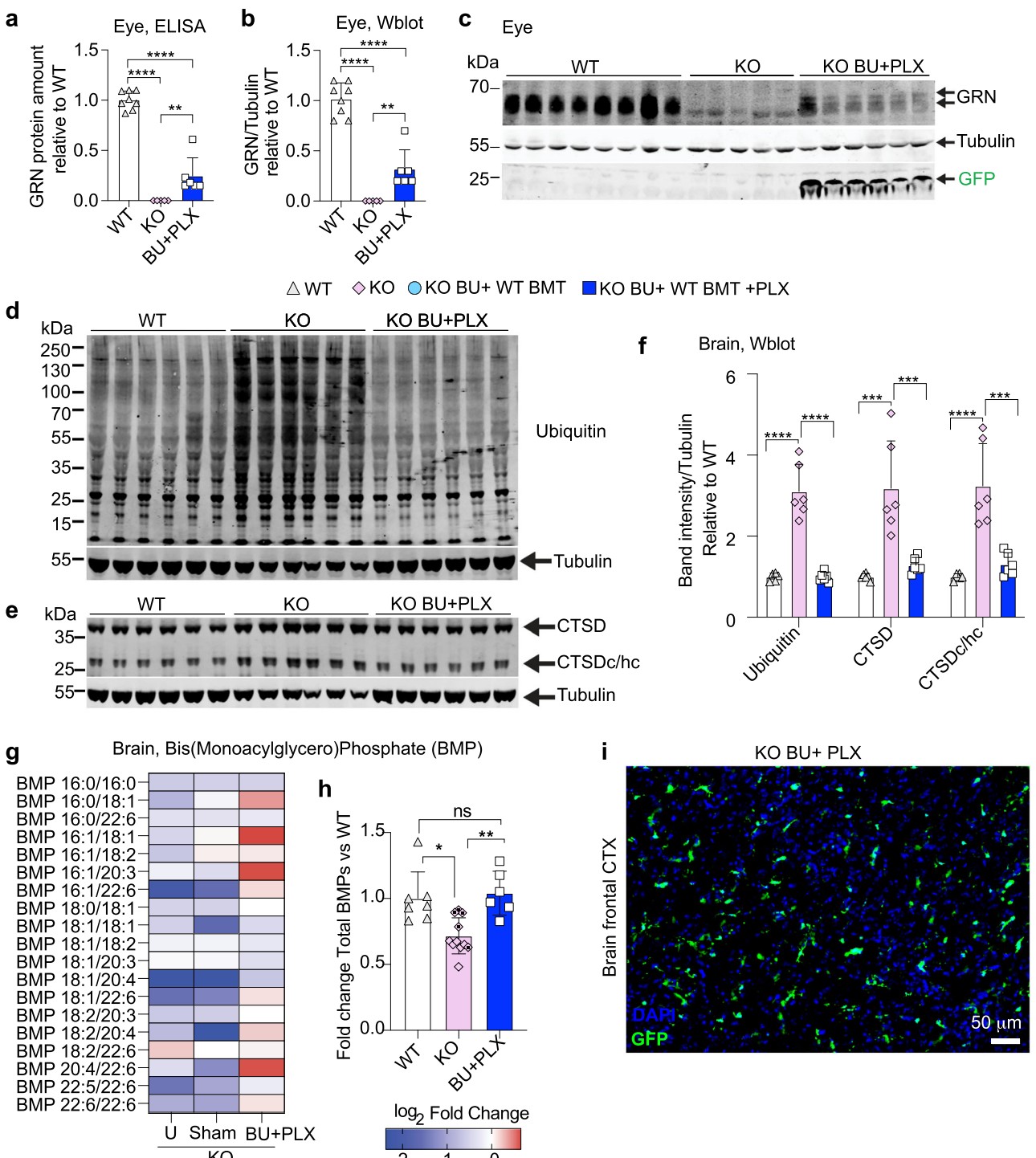

**Fig. 8 | Busulfan with PLX3397 reconstitutes progranulin in the eye and improves lipid metabolism and proteostasis in the brain of a mouse model of CLN11/FTD. a–i** Analysis of 6-month-old *Grn*$^{-/-}$ (KO) mice and wild-type (WT) controls. The study scheme and timeline are depicted in Fig. 7c. Analyses 4 months after bone marrow transplant (BMT). **a** GRN quantification by ELISA assay in eye lysates (WT n = 8, KO n = 5, BU + PLX n = 6). **b–c** GRN quantification by Western blot in eye lysates (WT n = 8, KO n = 5, BU + PLX n = 6). Tubulin was used as a loading control; the anti-GFP antibody detected the GFP protein expressed by transplant-derived cells. The uncropped blot is shown in **c**. **d–f** Analysis of proteostasis defects in KO mice. **d–e** Western blot analysis of ubiquitinated proteins, Cathepsin D (CTSD), and cleaved CTSD heavy chain (CTSDc/hc) in brain lysates (WT n = 6, KO n = 6, BU + PLX n = 6). Uncropped blots are shown in **d** and **e**. Tubulin was used as a loading control. **f** Quantification of ubiquitin and CTSD bands normalized by tubulin. **g–h** Targeted quantification of Bis(Monoacylglycero)Phosphate (BMP) in whole brain homogenates derived from the frontal brain. BMP levels were normalized by the total amount of POPC (1-palmitoyl-2-oleoyl-sn-glycero-3-phosphocholine). **g** Heatmap showing the normalized average amounts of BMP species relative to WT mice. Absolute amounts of BMP species are reported in the Source data file. **h** Histogram showing the normalized amounts of total BMP species in control KO mice [KO Untreated (U), n = 5, and KO Sham (Sham) n = 6] and KO mice conditioned with BU + PLX and transplanted with WT BM (BU + PLX n = 6). Dotted diamonds indicate Sham KO mice. **i** Representative image of transplant-derived GFP+ cells in the frontal cortex (fCTX) of KO mice conditioned with BU + PLX. The natural GFP fluorescence is depicted. Scale bar 50 μm. The image is representative of n = 3 mice. **a**, **b**, **f** Data are Mean ± SD. Source data are provided as a Source Data file. Statistical analysis: *$p < 0.05$, **$p < 0.01$, ***$p < 0.001$, ****$p < 0.0001$, the exact p-values of all comparisons are reported in the Source Data file; **a**, **b**, **f** One-way ANOVA with Tukey post-hoc; **h** Kruskal–Wallis with Dunn's post-hoc.

CCL4, CCL7, and CCL11) were also transiently induced after PLX3397 withdrawal and may represent signals for the recruitment, expansion, or maturation of MGLC progenitors in the brain. Interestingly, a transient increase in SDF-1/CXCL12, a CXCR4 ligand known to mobilize HPSCs[68], may also synergistically contribute to the overall CNS repopulation by transplant-derived cells. We found no induction of cytokines observed in immune-derived inflammatory processes[69,70]. Together, the data suggest a highly regulated effort to repopulate and replenish the depleted microglial niche by combining myeloid proliferative and chemoattractant signals. While it is probable that the recruitment effort is led by non-microglial cells such as neurons, astrocytes, and oligodendrocytes, further investigation is required to determine the specific contributions of each cell type.

After exposure to myeloablative doses of busulfan, residual host microglia show transcriptional differences from naive microglia that persist long-term. Being an alkylating drug, busulfan results in DNA damage and replicative senescence[33,126]. Consistent with this, our data shows alterations in genes regulating cellular metabolism, oxidative stress, DNA damage responses, and neurodegeneration pathways, with growth hormone (*Gh*) as a top differentially expressed gene likely linked to a busulfan-induced senescent phenotype[97]. These findings are particularly relevant for understanding the effects of busulfan conditioning and HSCT in the clinical setting.

As the rapid and robust repopulation of the CNS by transplant-derived cells could have important therapeutic implications, we also proved the therapeutic efficacy of the approach in a mouse model of GRN deficiency. In humans, GRN deficiency causes a spectrum of neurodegeneration ranging from childhood-onset (CLN11) to adult-onset disease (GRN-FTD), both of which lack effective treatments. Unlike BU alone, BU + PLX resulted in high engraftment of GFP+ MGLCs in the brain and retina, leading to GRN reconstitution in these tissues in a $Grn^{-/-}$ mouse. Although GRN was only partially reconstituted, it was sufficient to correct multiple disease biomarkers, including lipofuscinosis, protein ubiquitination, CTSD maturation defects, and anionic lipid deficiency. Transplantation also partially reconstituted GRN in the blood, suggesting therapeutic effects in peripheral organs. In the periphery, both conditioning regimens, BU and BU + PLX, resulted in similar levels of GRN reconstitution, but only BU + PLX impacted the CNS. These data demonstrate that CNS-wide microglia replacement by GRN-secreting MGLCs is necessary to provide therapeutic benefits in GRN-deficient mice following BMT. Interestingly, MGLC engraftment in the CNS of $Grn^{-/-}$ mice was lower than in wild-type recipient mice. This could reflect disease-specific differences in response to conditioning, implying optimization might be required for every indication. These results provide pre-clinical proof-of-concept for the efficacy of a microglia replacement therapy for CLN11/FTD-GRN. The future clinical translation of this approach could be enhanced by developing autologous transplantation methods. These approaches could mitigate risks while enabling the supraphysiological expression of GRN, thereby improving therapeutic effectiveness.

## Methods
### Mouse experimentation
Mice were housed in a 12-h dark/light cycle, temperature- (20–22 °C) and humidity (30–70%) -controlled environment with pressurized individually ventilated caging, sterile bedding, and unlimited access to sterile food and water in the animal facilities at Stanford University. All experiments were performed in accordance with the National Institutes of Health institutional guidelines and were approved by the Stanford University Administrative Panel on Laboratory Animal Care (IACUC 33365). Experiments included male and female mice based on availability from the vendor or colony. Data from both sexes were reported together, and no analyses were

performed based on sex, as we did not observe any sex-dependent impact on microglia replacement following conditioning with busulfan plus PLX3397 (Fig. 1e; the Source Data file includes sex-disaggregated data). At the end of each study, mice were deeply anesthetized using a mixture of Ketamine/Xylazine (80 mg/kg Ketamine/16 mg/kg Xylazine, intraperitoneal) and were perfused transcardially with 1X phosphate-buffered saline (PBS-1X, Fisher Scientific 10-010-023). Mice were euthanized by exsanguination while deeply anesthetized.

### Mouse conditioning
Adult (8–12-week-old) C57BL/6 J mice (Jax strain #000664) were conditioned with busulfan (Sigma-Aldrich 14843), total body irradiation (TBI), or treosulfan (MedChemExpress HY-16503). Busulfan was administered intraperitoneally at a dose of 25 mg/kg/day for either 4 days (total dose 100 mg/kg) or 5 days (total dose 125 mg/kg) before transplant, as detailed in each study. Treosulfan was administered intraperitoneally at a dose of 1 g/kg/day (500 mg/kg/dose, twice a day) for 5.5 days (total dose 5.5 g/kg) before transplant. Total body irradiation (TBI, 10 Gy, Kimtron IC-250) was performed 24 hours before the transplant. PLX3397 (Pexidartib, MedChemExpress HY-16749) was administered via gavage (100 mg/kg/day). The PLX3397 powder was resuspended in 100% DMSO and stored in aliquots at −80 °C; for administration the PLX3397 stock was diluted in a 1:1 mixture of Poly(ethylene glyco) Mn 400 (PEG400, Sigma-Aldrich 202398) and 1X Phosphate Buffered Saline pH 7.4 No Calcium/No Magnesium (abbreviated as PBS-1X, Fisher Scientific 10-010-023). The timing of administration and treatment duration is detailed in the figure legend. The optimal PLX3397 conditioning regimen for microglia replacement consisted of administering the drug 15 days after bone marrow transplant by oral gavage for 6 days (600 mg/kg, 100 mg/kg/day).

### Transplantation of total bone marrow
Bone marrow was harvested from adult (7–12-week-old) C57BL/6-Tg(CAG-EGFP)131Osb/LeySopJ mice (Jax strain #006567) and transplanted in sex-matched adult (7–12-week-old) C57BL/6 J mice (Jax strain #000664). In one study total bone marrow harvested from adult (15 week-old) hemizygous B6.Cg-Tg(CAG-mRFP1) mice (ubiquitously expressing RFP, Jax strain #005884) was transplanted in sex-matched adult (26-week-old) heterozygous B6.129P2(Cg)-Cx3cr1tm1Litt/J mice (expressing GFP from the mouse *Cx3r1* locus, Jax strain #005582). Total bone marrow cells were isolated by flushing femurs and tibiae with PBS-1X (Fisher Scientific 10-010-023) supplemented with 4 U/mL Heparin (Sigma-Aldrich H3149-500KU). After flushing the bone marrow cells were filtered using a 30 μm cell strainer, washed twice with PBS-1X and injected in total volume of 100 μL of PBS-1X ($1.5 \times 10^8$ cells/mL). Mice were transplanted 24 hours after conditioning with busulfan, treosulfan, or TBI with intravenous injection of total bone marrow cells in the retro-orbital sinus ($1.5 \times 10^7$ cells/mouse).

In one study eight-week-old B6(Cg)-Grntm1.1Aidi/J mice (Jax Strain #013175, abbreviated as $Grn^{-/-}$ or KO) were conditioned with busulfan (125 mg/kg, day −5 to 0; n = 17, 9 males and 8 females) and transplanted the day after with total bone marrow (BM) harvested either from age and sex-matched KO mice (Sham control mice n = 5, 3 males and 2 females) or from age and sex-matched adult C57BL/6-Tg(CAG-EGFP)131Osb/LeySopJ mice (Jax strain #006567; n = 12, 6 males and 6 females). Half of the $Grn^{-/-}$ mice conditioned with busulfan and transplanted with wild-type GFP+ bone marrow received PLX3397 by oral gavage 15 days after transplant (600 mg/kg, 100 kg/day; n = 6, 3 males and 3 females). Age-matched untreated $Grn^{-/-}$ mice (n = 6, 3 males and 3 females) and wild-type C57BL/6 J mice (Jax strain #000664, n = 10, 5 males and 5 females) were used as disease and healthy controls, respectively.

## Systemic transplantation of Lineage negative, KIT + SCA-1 + HSPCs expanded in culture

Lineage negative (Lin-) KIT + SCA-1+ (LKS) HSPCs were isolated from 12-week-old female C57BL/6-Tg(CAG-EGFP)131Osb/LeySopJ mice (Jax strain #006567) and cultured based on a published method[127]. Briefly, KIT+ cells were purified from total bone marrow (isolated as described above) using anti-KIT/CD117 microbeads and following the manufacturer's instructions (Miltenyi Biotec 130-097-146). Purified KIT+ cells were plated in cell bind plates (Costar 3337) at $5.5 \times 10^5$ cells/mL and cultivated for 14 days in F12 media (Gibco 11765-054) supplemented with 100 ng/mL mouse TPO (Peprotech 315-14), 10 ng/mL mouse SCF (Peprotech 250-03), 0.1% Polyvinyl alcohol (PVA, Sigma-Aldrich P8136), 1% HEPES (Gibco 15630-080), 1% ITS-X (Gibco 51500-056) and 1% Penicillin-Streptomycin-Glutamine (Gibco 10378-016). LKS HSPCs were maintained at 37 °C, 5% $CO_2$ and 5% $O_2$, half media changes were performed thrice a week. Bulk LKS HSPCs ($5.5 \times 10^5$ cells/mouse) were transplanted at culture day 14 into 8-week-old female C57BL/6 J mice (Jax strain #000664). Flow cytometry was used to evaluate the fraction of LKS HSPCs in culture at day 7, 11, and 14 using the following antibodies: anti-mouse CD45 PE-Cy7 (clone 30F11 Biolegend), anti-mouse SCA-1/Ly-6A/E PE (clone D7 Thermo Fisher Scientific), anti-mouse KIT/CD117 APC (clone 2B8 Thermo Fisher Scientific), anti-mouse CD3 PB (clone 17A2, BioLegend), anti-mouse TER-119 PB (clone TER-119 Thermo Fisher Scientific), anti-mouse Ly6C PacBlue (clone HK1.4 BioLegend), anti-mouse B220 PacBlue (clone RA3-6B2 Biolegend), eFluor™780 (Thermo Fisher Scientific 65-0865-18) was used as viability dye. Further details on primary antibodies used in this study are included in Supplementary Table 3.

## Intracerebroventricular transplantation of Lineage negative HSPCs

Lineage negative (Lin-) HSPCs were isolated from the whole bone marrow of 8-12-week-old C57BL/6-Tg(CAG-EGFP)131Osb/LeySopJ mice (Jax strain #006567) using the Mouse Lineage Cell Depletion Kit (130-090-858 Miltenyi Biotec) and transplanted in sex-matched 8–12-week-old C57BL/6 J mice (Jax strain #000664). Immediately after the purification, Lin- HSPCs were resuspended in PBS-1X at a concentration of $2 \times 10^7$ cells/mL and kept on ice while injected in the mouse lateral ventricle using a stereotactic apparatus. A total of 5 µL/ventricle were injected (1 µL/min) to have a total dose of $1 \times 10^5$ cells/ventricle.

## Flow cytometry analyses of cells isolated from mouse tissues

Mice were euthanized at various time points to analyze donor chimerism in tissues. Mice were deeply anesthetized using a mixture of Ketamine/Xylazine (80 mg/kg Ketamine/16 mg/kg Xylazine, intraperitoneal) and peripheral blood (PB) was collected using heparinized capillaries (Fisher Scientific, 22-260950) in PBS-1X supplemented with 10 mM EDTA pH 8.0 (Thermo Fisher Scientific 15-575-020), then, peritoneal cells were collected by washing the peritoneal cavity with 4 mL of PBS-1X. Femurs and tibiae, spleen, brain, liver, heart, lung, and eye were collected after the transcardial perfusion of the anesthetized mice with PBS-1X. Erythrocytes in PB were precipitated using 2% Dextran (Spectrum Chemical D1004) in PBS-1X and the supernatant was collected in 5 mL of FACS-BL buffer [PBS-1X supplemented with 5% Fetal Bovine Serum (FBS, Thermo Fisher Scientific 16000069, and 1% BSA fraction V (Sigma-Aldrich 10735078001)]. Peripheral blood mononuclear cells (PBMCs) were obtained by centrifugation at 400 g for 5 minutes. Bone marrow cells (BM, from femurs and tibiae) and splenocytes (SP) were extracted in RPMI (Thermo Fisher Scientific 61870127) supplemented with 10% FBS, 4 U/mL Heparin (Sigma Aldrich H3149-500KU) and 0.2 U/mL Deoxyribonuclease I (Worthington Biochemical Corporation LS002007) and, filtered using a 30 µm cell strainer.

Erythrocytes were lysed using the RBC lysis buffer (Thermo Fisher Scientific 00-4333-57). After RBC lysis the cells were washed and resuspended in FACS-BL buffer and kept on ice for the following procedures. For flow cytometry staining, cells were resuspended in 1% vol/vol Mouse BD Fc Block™ (clone 2.4G2 BD Biosciences) 10 minutes and stained in the dark for 30 min using the following antibodies: anti-mouse CD45 PE-Cy7 (clone 104 Thermo Fisher Scientific), anti-mouse CD45 PE-Cy7 (clone 30F11 Biolegend), anti-mouse/human CD11b-PE (clone M1/70 BioLegend), anti-mouse/human TER-119 PE-Cy5 (clone TER-119, Thermo Fisher Scientific), anti-mouse Ly6C BV605 (clone AL-21 BD Biosciences), anti-mouse CD3 APC (clone 17A2 Thermo Fisher Scientific), anti-mouse CD19 PB (clone 6D3 BioLegend); anti-mouse Ly6G PE (clone 1A8 Biolegend), anti-mouse CD41 PacBlue (clone MWReg30 BioLegend), anti-mouse CSF1R-APC (clone AFS98, eBioscience); eFluor™780 (Thermo Fisher Scientific 65-0865-18) was used as a viability dye. After staining, cells were washed and resuspended in FACS-BL buffer. For absolute cell quantification, 50 uL of absolute counting beads (Life Technologies, C36950) were added to the stained cells in a fixed ratio, following the manufacturer's guidelines. Stained cells were acquired using Cyto-FLEX and CytExpert Software (Beckman Coulter) and sorted using a MA900 Multi-Application Cell Sorter and MA900 Software (Sony Biotechnology). Flow cytometry data were analyzed using FlowJo software (FlowJo, LLC).

## Microglia and macrophage isolation

Isolation of brain microglia was performed using a modified version of a reported method[128]. The brain was minced in ice-cold HBSS 1X (Thermo Fisher Scientific 14185052), the pieces were collected by centrifugation at 400 g for 5 minutes and digested using the NTDK Neural Tissue Dissociation Kit (P) (Miltenyi Biotec 130-092-628) for 30 minutes following the manufacturer's guidelines. Digested samples were quenched using ice-cold HBSS 1X and filtered through a 70 µm cell strainer, resuspended in ice-cold 33% isotonic Percoll PLUS (GE Healthcare GE17-5445-01) and spun at 700 g for 15 minutes at 4 °C (brake 3) for myelin removal. The cell pellet was washed with ice-cold FACS-BL and kept on ice for downstream procedures.

To isolate macrophages from liver, heart and lung the organs were minced and digested in 2 mL of HBSS 1X buffer supplemented with 0.125 mg/mL Liberase TM (Sigma-Aldrich 05401119001), 0.5 mg/mL Collagenase type II (Sigma-Aldrich C6885) and 0.3 mg/mL Deoxyribonuclease I (Worthington Biochemical Corporation LS002007) at 37 °C for 30 minutes, then filtered through 100 µm cell strainer (liver and lung) and 40 µm cell strainer (heart), resuspended in ice-cold 33% isotonic isotonic Percoll PLUS (GE Healthcare GE17-5445-01) and spun at 700 g for 15 minutes at 4 °C (brake 3) for fat tissue removal. The cell pellet was then washed with ice-cold MACS buffer [PBS 1x, 0.5 % bovine serum albumin fraction V (Sigma-Aldrich 10735078001) and 0.002 mM EDTA pH 8.0 (Thermo Fisher Scientific 15-575-020)], spun at 400 g for 5 minutes at 4 °C and kept on ice for the following procedures.

For flow cytometry staining, microglia and macrophages cells were resuspended in 1% vol/vol Mouse BD Fc Block™ (clone 2.4G2 BD Biosciences) 10 minutes and stained in the dark for 30 min using the following antibodies: anti-mouse CD45 PE-Cy7 (clone 30F11 Biolegend), anti-mouse/human CD11b-PE (clone M1/70 BioLegend), anti-mouse/human TER-119 PE-Cy5 (clone TER-119, Thermo Fisher Scientific), anti-mouse Ly6C BV605 (clone AL-21 BD Biosciences), anti-mouse CD3 APC (clone 17A2 Thermo Fisher Scientific), anti-mouse CD19 PB (clone 6D3 BioLegend); eFluor™780 (Thermo Fisher Scientific 65-0865-18) was used as a viability dye. After staining, cells were washed and resuspended in FACS-BL buffer. Stained microglia and macrophages were acquired using CytoFLEX and CytExpert Software (Beckman Coulter). Microglia was sorted using a MA900 Multi-Application Cell Sorter (Sony Biotechnology). Flow cytometry data were analyzed using FlowJo software (FlowJo, LLC).

## Microglia-like cell and bone marrow CD11b+ cell cultures

After flow cytometry staining and FACS sorting, as described above, transplant-derived GFP + CD45 + CD11b+ cells isolated from either brain or bone marrow and were seeded in tissue-culture treated 48-well plates ($3 \times 10^5$ cell/well, Fischer Scientific FB012930) in complete culture media [DMEM/F12-Hepes (Thermo Fisher Scientific 11330032) supplemented with 10% fetal bovine serum, 40 ng/mL mouse GM-CSF (Peprotech 315-03), 100 mg/mL mouse CSF1 (Peprotech 315-02), 1% Penicillin-Streptomycin-Glutamine]. The cells were cultured at 37 °C, 5% $CO_2$ and 5% $O_2$, half media change media was performed thrice a week. After 11 days the complete culture media was replaced by serum-free media for 72 hours (DMEM/F12-Hepes supplemented with 40 ng/mL mouse GM-CSF, 100 mg/mL mouse CSF1, 1% ITS-X, 1% Penicillin-Streptomycin-Glutamine). Culture media collected on day 14 was used to measure mouse GRN by ELISA assay (progranulin mouse ELISA Kit, AdipoGen AG-45A-0019YEK-KI01).

## Histological analyses

Brain, liver, and eye were collected from mice following transcardial perfusion with cold PBS-1X followed by overnight fixation in paraformaldehyde solution 4% in PBS (Santa Cruz Biotechnology sc-281692). Fixed samples were washed once with PBS-1X and transferred to a sucrose solution 30% in PBS overnight for cryoprotection, embedded in Tissue-Tek optimal cutting temperature compound (OCT, Fisher Scientific 4585), and cut (20 µm sections) on a cryostat (Leica, Wetzlar, Germany, CM3050). Tissues and serial sections were stored at −20 °C until further use. To image transplant-derived GFP+ cells in the brain, spinal cord, retina, and liver the slides were washed once in in PBS-1X + 1 mM $CaCl_2$ + 0.5 mM $MgCl_2$ (abbreviated as PBS-1X + +) counterstained with Hoechst 3342 diluted 1 to 1000 in PBS-1X (Thermo Fisher Scientific PI62249), and mounted in Aqua Poly/Mount (Polysciences 18606-20) for fluorescent microscopy. The number of GFP+ cells in the retina was counted using 20X composite images of a whole retinal section (nasal-temporal). The slides were de-identified before and after image acquisition to ensure unbiased quantification. For immunofluorescence staining, slides were washed in PBS-1X + + to remove excess OCT. Sections were blocked and permeabilized in PBS-1X + + containing 0.2 % Triton X-100 (Sigma Aldrich T8787) and 10% normal goat serum or normal donkey serum (NGS or NDS, respectively; Sigma-Aldrich G9023 and Jackson ImmunoResearch 017-000-121) for 1 h at 25 °C. Primary antibodies were incubated overnight in PBS-1X + + with 5% NGS/NSDS at 4 °C. Secondary antibodies were incubated in PBS-1X + + with 5% NGS/NDS 90 minutes at 25 °C. Slides were then washed in PBS-1X + +, counterstained with Hoechst 3342 diluted 1 to 1000 in PBS-1X (PI62249 Thermo Fisher Scientific), and mounted in Aqua Poly/Mount (Polysciences 18606-20) for fluorescent microscopy. To stain microglia and myeloid cells, we used a rabbit anti-mouse IBA-1 antibody diluted 1 to 100 (AB 8395040 FUJIFILM Wako Pure Chemical Corporation) and a secondary Goat anti-rabbit Alexa Fluor 555 antibody diluted 1:500 (A21428 Thermo Fisher Scientific). To stain liver macrophages, we used a rat anti-mouse F4/80 antibody diluted 1 to 100 (clone BM8.1 Cell Signaling Technology) and a secondary Donkey anti-rat Cy3 antibody diluted 1:500 (P189CMI Fisher Scientific).

To measure lipofuscin storage (lipofuscinosis) brain slides were washed in PBS-1X, counterstained with Hoechst 3342 diluted 1 to 1000 in PBS-1X (PI62249 Thermo Fisher Scientific), and mounted in Aqua Poly/Mount (Polysciences 18606-20) for fluorescent microscopy. Further details on primary antibodies used in this study are included in Supplementary Table 3. Auto-fluorescent lipofuscin granules were imaged at 20X magnification in the CA3 region of the brain, where lipofuscinosis was evident in $Grn^{-/-}$ mice. Non-saturated auto-fluorescence images were taken at the same exposure from all brain sections for quantification. The lipofuscin autofluorescence was quantified with ImageJ and normalized by the selected area. The slides

were de-identified before and after image acquisition to ensure unbiased quantification. All images were visualized and acquired using an all-in-one Fluorescence Microscope BZ-X800 and BZ-X800 sofware (Keyence, Itasca).

## Cytokine measurement in whole brain lysates and plasma

Ten/thirteen-week-old C57BL/6 mice were conditioned with busulfan (100 mg/kg, n = 12; 25 mg/kg/day, from day −1 pre-transplant to day -4 pre-transplant) or left untreated (Untreated, n = 3). Fifteen days after bone marrow transplant (BMT) the busulfan-conditioned mice (n = 12) received PLX3397 (Pexidartib, MedChemExpress HY-16749) via oral gavage for six days (100 mg/kg/day). Mice were euthanized at 1, 4, 8 and 21 days after PLX3397 conditioning (n = 3 mice/day). Upon anesthesia (Ketamine 80 mg/kg/Xylazine 16 mg/kg) blood was collected in Safe-T-fill Dipotassium EDTA tubes (RAM Scientific 077053), and plasma was separated by centrifugation at 200 × g for 10 min. After blood collection, the mice were transcardially perfused with PBS-1X. Brains were dissected and one sagittal half/brain was snap-frozen in liquid nitrogen for cytokine analyses, while the other half was used for flow cytometry analyses. For cytokine analyses the frozen brains were lysed in RIPA Lysis buffer 1X (G Biosciences 786-489) supplemented with a protease inhibitor cocktail (Complete, Sigma-Aldrich 11836153001). Brain protein concentration was normalized using the BCA protein assay kit (Thermo Fisher Scientific 23223). Mouse cytokines were quantified using the Immune Monitoring 48-Plex Mouse ProcartaPlex Panel (EPX480-20834-901, Thermo Fisher Scientific). Luminex assay was performed at the Human Immune Monitoring Center (HIMC, Stanford University School of Medicine, Stanford, CA, USA). Plasma samples were diluted 3 folds with PBS-1X and run in duplicate. Whole brain lysates, normalized to 5 mg/mL were run undiluted, in duplicate. ELISA assay was used to measure IL34 (DY5195-05, Mouse IL-34 DuoSet ELISA, R&D Systems) and SDF-1/CXCL12 (DY460, Mouse CXCL12/SDF-1 DuoSet ELISA, R&D Systems) cytokines in brain lysates following the manufacturer's instructions.

## Behavior study in wild-type mice and $Grn^{-/-}$ mice

Mice were group housed under reversed light cycle (8:30 am Light OFF-8:30 pm Light ON) and behavior was tested during the animal dark cycle at the Stanford's Behavioral and Functional Neuroscience Laboratory (SBFN) by an experimenter who was blinded to the genotypes and treatments. In one study eight-week-old C57BL/6 J mice (Jax strain #000664) were either untreated (n = 10, 5 males and 5 females), conditioned with busulfan (n = 10, 5 males and 5 females) or busulfan and PLX3397 (n = 10, 5 males and 5 females). The conditioned mice were transplanted with total bone marrow from age and sex-matched C57BL/6-Tg(CAG-EGFP)131Osb/LeySopJ mice (Jax strain #006567), as detailed above. Behavioral analyses were performed between 17 and 19 weeks after bone marrow transplant, corresponding to 14-16 weeks following the end of the PLX3397 administration, at this time point the mice were 6-7 month-old. In another study untreated eight-week-old male B6(Cg)-Grntm1.1Aidi/J (Jax Strain #013175, abbreviated as $Grn^{-/-}$ or GRN KO, n = 12) and untreated age-matched male C57BL/6 J mice (Jax strain #000664, n = 15) underwent serial behavioral analyses from 2 up to 12 months of age.

**Activity chamber.** The locomotor assessment took place in an Open Field Activity Arena and Activity Monitor Software-811 (Med Associates Inc., St. Albans, VT. Model ENV-515) mounted with three planes of infrared detectors, within a specially designed sound attenuating chamber (Med Associates Inc., St. Albans, VT. MED-017M-027). The arena was 43 cm (L) x 43 cm (W) x 30 cm (H), and the sound attenuating chamber was 74 cm (L) x 60 cm (W) x 60 cm (H). The mice were placed in the corner of the testing arena and allowed to explore the arena for 10 minutes while being tracked by an automated tracking system. Parameters including distance moved, velocity, rearing, and

times spent in periphery and center of the arena were analyzed. The periphery was defined as the zone 5 cm away from the arena wall. The arena was cleaned with 1% Virkon solution at the end of each trial.

**Y-maze: spontaneous alternation.** The Y-Maze was made of plastic with 3 arms in a Y shape. The arms were labeled as Arm A, B, and C. Arm A was 20.32 cm (L) x 5 cm (W) 12.7 cm (H), and Arm B and C were 15.24 cm (L) x 5 cm (W), and 12.7 cm (H). The test is based on the willingness of rodents to explore a novel environment and designed to measure spontaneous alternations. A normal rodent would prefer to explore a different arm of the maze than the arm they previously visited. Animals were placed in the center of the maze facing the intersection between arms B and C. The first entry was excluded from data analysis due to potential influence from the experimenter. Using an overhead camera, the numbers of arm entries and alternations were recorded for 5 minutes. An arm entry was defined as when all four paws entered into a new arm of the maze. The apparatus was cleaned with 1% Virkon after each trial. Parameters including percent alternation rate and total number of entries into arms were analyzed.

**Novel place recognition (NPR) and novel-object recognition (NOR).** The Novel Place Recognition (NPR) task assesses the ability of a subject to detect that a familiar object has been moved to a new location. The Novel Object Recognition (NOR) task assesses the ability of a subject to detect that an object has been replaced with a different object. NPR/NOR testing was conducted in a plastic arena 52 cm (L) x 52 cm (W) x 40 cm (H). The walls of the plastic arena were black, and the floor was white. A white index card 12.7 cm (L) x 7.62 cm (W) was affixed to one wall as a visual cue. Two objects were used for this test: the green tower (18 cm H x 4 cm L x 4 cm W) and the white bottle (16 cm H x 4 cm L x 4 cm W). The green tower was made of dark green plastic and had a square base. The white bottle was made of white plastic and had a circular base. Assignment of one object as the NPR object and the other as the novel object introduced during NOR was pseudorandomized to ensure results were not influenced by an innate preference for either object. The object configuration was also pseudorandomized across subjects. The NPR/NOR tests were conducted over 3 days. On the first day, habituation, mice were each placed in the center of the empty arena and allowed to explore freely for the duration of a 10-minute trial. The NPR assessment was conducted on the second day. For NPR training, the arena had three identical objects placed in three different corners 10 cm from the wall. The mice were placed in the center of the arena and allowed to explore for 10 minutes. At the end of the NPR training, mice were returned to their home cage for 3–4 minutes prior to NPR testing. For NPR testing, one object was moved to the previously empty corner. Mice were then placed in the center of the arena and allowed to explore for 5 minutes. NOR assessment was conducted on the third day. One of the objects was removed and replaced with a different object. The mice were again placed in the center of the arena for a 5-minute trial. The trials were recorded, and mice were tracked with the automated tracking system Ethovision XT and Ethovision XT software 17.5 (Noldus Information Technology, Wageningen, Netherlands). An interaction was defined as when the nose of the mouse was within 2 cm an object. Arena and objects were cleaned with 1% Virkon between each trial.

**3-chambers social test.** The 3-Chambers apparatus was made of clear Plexiglas 60 cm (L) x 40.5 cm (W) x 22 cm (H) equally divided into three chambers by two pieces of clear dividers with a small hole 10 cm (L) x 5 cm (H) for mice to transition into all 3 chambers. The experiment consisted of three 10 minutes trials: Habituation, Sociability, and Social Discrimination. During the Habituation, the subject mouse was placed in the middle chamber with the wire cups placed inside the left and right chambers. The subject mouse was allowed to explore all three chambers for 10 min. After 10 min the subject mouse was guided into

the middle chamber without access to the left and right chamber. During Sociability trial, Stranger 1 (3-5 weeks old, same sex, novel juvenile mouse) was placed under the cup in the left chamber and a Novel Object (plastic cap) in the right chamber. The subject mouse was allowed to explore the three chambers for another 10 minutes. After Sociability test the subject mouse was placed back in the middle chamber. Social Discrimination begins by placing Stranger 2 under the cup in the left chamber and the Stranger 1 in the right chamber. The subject mouse is allowed to explore all three chambers again for another 10 minutes. The position of Stranger 1 and Stranger 2 in left and right chamber are pseudo randomized. The Stranger mice were only exposed to one subject mouse per day. The Stranger mice were habituated to the cups for 10 min x 3 days prior to testing to reduce stress and mobility under the cup. Each Stranger mouse was dedicated to single cup during the test and the cups were washed with 70% ethanol after the test. The subject mouse was monitored by Ethovision tracking system and the total time in chambers and interactions with Stranger mice/Novel Object were analyzed. Interaction was defined as sniffing or direct contact with the Stranger mice/Novel Object within 2 cm of the wire cup. Chambers Social Test was conducted in Grn$^{-/-}$ mice and wild type controls at 2, 4, 6, 9, and 12 months of age. New Stranger mice were used at each timepoint.

**Passive avoidance.** The Passive Avoidance Test (PAT) was conducted using GEMINI avoidance system and Gemini software (San Diego Instruments, San Diego, CA). This automated system contained two compartments which were separated by a guillotine door (gate). Both compartments had grid floors which delivered electric shocks, but one compartment was lit and another one was dark. The experiment consisted of 1 day of habituation, 1 day of training, and 1 day of testing. On habituation day, the mouse was placed in the lit compartment. After 30 sec acclimation, the gate was opened and the mouse was allowed to explore both compartments freely. The gate was programmed to close when the mouse entered the dark compartment preventing the mouse from returning to the lit compartment. The mouse was removed from the system 30 sec after entering the dark compartment. On the following day, Training Day, the mouse was placed in the lit compartment. After 30 sec of acclimation the gate was opened and the mouse was allowed to explore both compartments freely. The gate was closed after it entered the dark compartment and an electric shock (0.5 mA for 2 sec) was delivered 3 sec after the door was closed. The mouse remained in the dark compartment for an additional 30 sec before being removed and returned to the home cage. On the following day, Day 1 Testing Day, the mouse was placed in the lit compartment. After 5 sec of acclimation, the gate was opened. When the mouse entered the dark compartment, the gate was closed, and trial ended. The mouse was returned to the home cage. The maximum duration of each trial was 300 sec after the door opened. The time between the gate opening and the mouse passing through the gate was recorded as latency time. The compartments were cleaned with 1% Virkon between each animal. Grooming Behavior. Mice were singly housed in new home cages for 20 minutes, video recorded. The videos were hand-scored by the experimenter. The grooming behavior was defined as face-wiping, scratching/rubbing of the head and ears, and full-body grooming including licking or brushing any part of the body.

**Nesting behavior.** The Nesting behavior was conducted in a new mouse home cage with clean bedding (Innovive cage (37.3 cm(L) x 23.4 cm(W) x 14.0 cm(H)) made of clear plastic. Mice were singly housed during the experiment and a 3 g cotton nestlet (Ancare Corp.) approximately 5 cm (L) x 5 cm (W) x 0.5 cm (H) was introduced into the cage during 1st hour of the dark cycle. A nestlet score of 1–5 was recorded at 7 hours and 24 hours after introduction of the nestlet. Nesting Score: score of 1 = More than 90% of the nestlet untouched; score of 2 = 50-90% of the nestlet intact with nestlet partially shredded;

score of 3= Nestlet shredded 50-90% but no identifiable nest site. Less than 50% of the nestlet remains intact, but less than 90% is within a quarter of the cage floor area. Nestlet is not gathered into a nest but is spread around the cage. The material may sometimes be in a broadly defined nest area; score of 4= More than 90% of the nestlet is torn and the nestlet is gathered into an identifiable nest within a quarter of the cage floor area. However, the nest has flat wall which is defined as less than 50% of the circumference of mouse body height when curled up on its side. score of 5= More than 90% of the nestlet is torn and the nest is a crater, with walls higher than 50% of the circumference of the mouse body height. If the nesting score is ambiguous (e. g. a nest with a score of 5, but more than 10% of the nestlet un-shredded) a score of 4.5 was assigned.

### Single-cell RNA sequencing (scRNA-seq)

Brain and bone marrow cells were isolated, stained, and FACS-sorted as described above.

The scRNA-seq experiment comprised 12 samples [MGLCs, host MG, naive MG, and bone marrow (BM)-derived CD11b+ cells; n = 3 mice per sample]. Specifically, three adult C57BL/6 J mice (Jax strain #000664) conditioned with busulfan+PLX3397 and transplanted with Lin- KIT + SCA-1+ (LKS) HSPCs isolated from adult C57BL/6-Tg(CAG-EGFP)131Osb/LeySopJ mice (Jax strain #006567) were used to FACS sort GFP + CD45 + CD11b+ MGLCs, GFP- CD45 + CD11b+ host MG, and GFP + CD45 + CD11b+ BM-CD11b+ cells. Additionally, three age- and sex-matched untreated C57BL/6-Tg(CAG-EGFP)131Osb/LeySopJ mice were used to isolate GFP + CD45 + CD11b+ naive MG. Details regarding mouse conditioning and transplantation can be found in the sections above.

Cell Hashing with oligo-tagged antibodies was employed to label cells from distinct mice during scRNA-seq. The oligo-tagged antibodies were directed to mouse CD45 (clone 30-F11) and mouse MHCI (clone M1/42). The sequencing of each mouse-specific tag alongside the cellular transcriptome allowed us to pool cells from three different mice per sample and prevent sample-specific batch effects that occur during scRNA-seq. Each unique oligo-tagged antibody mix was added to the flow cytometry antibody mix described above (0.15 μL/1 × 10^6 cells). After staining, cells were washed and resuspended in ice-cold FACS-BL buffer. Stained cells were sorted using a MA900 Multi-Application Cell Sorter (Sony Biotechnology) in ice-cold FACS-BL buffer. The unique oligo-tagged antibodies were as follows: MGLC, host MG, BM-CD11b+ mouse 1 (TotalSeqTM-B 0301 anti-mouse Hashtag 1 Antibody, BioLegend 155831), mouse 2 (TotalSeqTM-B 0302 anti-mouse Hashtag 2 Antibody, BioLegend 155833), mouse 3 (TotalSeqTM-B 0303 anti-mouse Hashtag 3 Antibody, BioLegend 155835); naive MG mouse 1 (TotalSeqTM-B 0301 anti-mouse Hashtag 1 Antibody, BioLegend 155831), mouse 2 (TotalSeqTM-B 0302 anti-mouse Hashtag 2 Antibody, BioLegend 155833), mouse 3 (TotalSeqTM-B 0303 anti-mouse Hashtag 3 Antibody, BioLegend 155835).

A minimum number of 100,000 cells from each FACS sorted sample (n = 3 mice/sample) was provided to by MedGenome Inc. (Foster City, CA, USA) for downstream processing. The cells were used to generate gel emulsions (GEM) using the 10X Chromium Controller (10x Genomics). After encapsulating single cells in the GEMs, they were lysed, and cDNAs were generated. The cDNAs were quantified, and sequencing libraries for gene expression and proteins were prepared using the Dual Index Kit TT Set A (10x Genomics PN-3000431) & Dual Index Kit NT Set A (10x Genomics PN-3000483) respectively. Before sequencing, the quality of cDNA and libraries was assessed using a tapestation instrument (Agilent 4200). The libraries were then sequenced using the Illumina Novaseq 6000 sequencer and NovaSeq Control Software (Illumina). The Illumina raw BCL sequencing files were processed through the CellRanger software (10x Genomics) to generate FASTQ files and count matrices (https://www.10xgenomics.com/support/software/cell-ranger/latest/analysis/running-pipelines/cr-3p-multi). Feature-barcode matrices obtained from Cellranger

count for all the samples were processed using the 'Read10X()' function from Seurat package (v4.0)[129]. Subsequently, cell filtering was performed based on nfeatures (>200 & <6000) and the percentage of counts from mitochondrial genes (percent mt <5)[130]. Normalization and scaling were carried out, and the top 2000 genes with the highest standardized variance were used to identify significant principal components (PCs). Integration of all 12 samples (MGLC, host MG, naive MG, and BM-CD11b + , 4 groups, n = 3 mice per sample) was done using the Seurat package integration method RPCA, with the functions FindIntegrationAnchors() and IntegrateData(). Clustering was performed using a graph-based method with the FindClusters() function and the resolution parameter set to 0.2. The resulting clusters were visualized using Uniform Manifold Approximation and Projection for Dimension Reduction (UMAP). FindAllMarkers Seurat function was utilized to find markers for each cluster. Heatmaps showing the top 5 DEGs in each cluster were generated using the DoHeatmap function of SEURAT v4.0[129]. Cell annotation was generated by using the ScType R package with a pre-curated list of markers[131–133]. With the annotations from ScType, markers in each population were identified, and the top 5 DEGs in each population were visualized on a heatmap. For single-group analysis, all the steps were executed, excluding the integration process. The analysis of average gene expression was conducted using the Scanpy package (v1.9.3) in Python[134]. The mean log-normalized UMI counts for each gene of interest in different sample groups/clusters (MGLC, host MG, naive MG, and BM-CD11b+ or MGLC clusters) and the fraction of cells expressing a gene per group were computed and visualized using the DotPlot() function. The fraction of cells expressing a gene in each sample group/cluster was determined by dividing the number of cells expressing the gene by the total number of cells in the group/cluster. A gene was considered expressed in a cell if its normalized expression value was greater than zero. The log normalized UMI counts for each gene in the different sample groups ranged from 0 to 6. Pseudo-bulk analyses were performed using the decoupler-py (v1.5.0) and PyDeSeq2 (v0.4.4) Python packages[135,136]. Pseudo-bulk data were computed by summing counts mapped to each gene in each biological replicate (n = 3) per condition via the dc.get_pseudobulk() function in decoupler-py. Differentially expressed genes (DEGs) between host and naive microglia, and MGLC clusters were identified using the deseq2() function in PyDeSeq2. Benjamini−Hochberg (BH)-adjusted p-values less than 0.05 were considered significant. Enriched pathway analyses were conducted using ShinyGO (V0.8)[137] and plotted using ggplot2 package in R.

The scRNA-seq data of brain-border-associated macrophages (BAMs) generated by Van Hove et al. (GSE128855, Macrophage Aggregrate) was used to compare BAM gene expression to the scRNA-seq data generated in the present study. Raw read count matrix and cell annotation files were downloaded (https://www.brainimmuneatlas.org/download.php). The count matrix was processed following the above workflow using Seurat functions. The processed matrix was annotated with cell type information and analyzed for differential gene expression. After the processed matrix was annotated with cell type information, a differential gene expression analysis was carried out between all the different cell types of the GSE128855 Macrophage aggregate dataset. The top 5 expressed markers were tested on the subset MGLCs obtained after busulfan + PLX3397 conditioning. Heatmaps were generated for comparison to assess the qualitative expression of those markers in all MGLC clusters.

### CyTOF staining run and analyses

Extraction of brain and bone marrow cells was performed as described above. Immediately after isolation the cells were washed in ice-cold MACS buffer, spun at 400 g for 5 minutes at 4 °C, resuspended in 100 uL of MACS buffer and fixed using PROT1 (Smart Tube Inc proteomic Stablizer, Smart Tube, PROT1) using the manufacturer's instruction (MACS buffer: PROT1 ration 1:1.4)[138]. Upon 10 minutes incubation at

22 °C the cells were frozen in dry ice and stored at -80 °C. To process samples for CyTOF analysis, the samples were thawed for 10 minutes at 4 °C and washed twice with 1 mL of cell-staining medium (CSM; PBS with 0.5% BSA, 0.02% sodium azide and 2 mM EDTA pH 5) before proceeding with the staining using the surface and intracellular markers described in Supplementary Table 2. The metal-conjugated primary antibodies were either commercially available or conjugated using metal-specific Maxpar® Labeling Kits (https://fluidigm.my.site.com/Storefront/Cytometry/ConsumablesandReagentsCytometry/MaxparAntibodyLabelingKits?cclcl=en_US, Fluidigm/Standard Biotools; Supplementary Table 2). Following the two CSM washes the cells were resuspended in 1% vol/vol Mouse BD Fc Block™ (clone 2.4G2 BD Biosciences) in 50 μL of CSM and incubated for 10 minutes on ice. Master mixes of antibodies for surface (50 μL/sample) and intracellular (100 μL/sample) markers were prepared in CSM separately and filtered using a 0.1 μM filter. The staining was performed for 30 minutes at 22 °C on a shaker (300 rpm). After the surface staining the cells were washed with 4 mL of CSM and permeabilized with 1 mL of ice-cold methanol 100% for 15 minutes at 4 °C. Samples were then washed twice with 4 mL of CSM and then stained with the antibody mix for the intracellular markers. Cells were then washed with 4 mL of CSM and stained with 1:5000 191Ir/193Ir DNA intercalator (201192 A Fluidigm/Standard Biotools) in 1.6% PFA in PBS-1X overnight at 4 °C. The following day the cells were washed with 4 mL CSM and 2 ×4 mL ddH2O, filtered and resuspended in 139La/142Pr/59Tb/160Tm/175Lu normalization beads before being analyzed using a Helios mass cytometer and Helios instrument control software (Fluidigm) at 200 events/sec rate. IMD files were converted into FCS files and normalized with publicly available Matlab Normalizer v0.3 (https://github.com/nolanlab/bead-normalization)[139]. Raw count values were used to measure protein expression. FCS files were analyzed using OMIQ software from Dotmatics (https://www.omiq.ai/). Gated live cells (DNA + /cPARP-) were used for downstream analysis to calculate the percentage of positive cells or the expression of each marker. To identify cell subsets and to visualize single cell expression in 2D dimension, we run the optimized Stochastic Neighbor Embedding (Opt-SNE) on either live cells (DNA + / cPARP-) or CD45 + CD11b+ gated cells using all the markers for the clustering and the default settings for the run. Using the opt-SNE coordinates, we gated the GFP + CD45 + CD11b+ cells (MGLCs) and the GFP- CD45 + CD11b+ cells. Surface and intracellular markers were gated on GFP+ MGLCs and the GFP-negative microglia or live cells.

## Progranulin ELISA and Western blot

ELISA assays were used to measure GRN protein in the serum (DY2557-05, Mouse progranulin DuoSet ELISA, R&D Systems), brain, and eye lysates (AG-45A-0019YEK-KI01, progranulin mouse ELISA Kit, AdipoGen) following the manufacturer's instructions. Following transcardial perfusion, each brain was dissected sagittally and the frontal region, including the frontal cortex, was collected for BMP analyses. The remaining ipsilateral part of the brain was collected for Western blot and ELISA assays. The brain portions were snap-frozen in liquid nitrogen immediately after dissection. For the ELISA assay, the snap-frozen brains and eyes were lysed in ELISA buffer [20 mM Tris,150 mM NaCl, 0.1% Triton supplemented with protease inhibitors (cOmplete™ Protease Inhibitor Cocktail, 11836153001 Roche)] using a Tissue Lyser II (20 Hz for 2 minutes, Qiagen). Protein quantification in the lysates was performed by BCA (Pierce™ BCA Protein Assay Kits and Reagents 23223 Thermo Scientific).

For Western blot analyses, serum, brain, and eye lysates were denatured in Loading buffer-1X containing 1X Laemmli Sample Buffer (1610747 Thermo Scientific) and 1X Reducing Agent (novex NuPAGE Sample Reducing Agent NP0009 Invitrogen) at 95 °C for 5 minutes. Protein samples were loaded on 4–12% Invitrogen Nupage™ Bis-Tris Protein Gels (WG1403BOX Fisher Scientific) and run in MOPS SDS

Running Buffer-1X (NP0001 Invitrogen). Blotting was performed using iBlot2 Gel Transfer Device (Invitrogen). Membranes were blocked in Intercept TBS blocking buffer (927-60001 LI-COR Biosciences). Primary antibodies were: sheep anti-mouse GRN (AF2557, R&D systems), rabbit anti-GFP (AB 221569, Thermo Scientific), mouse anti-alpha Tubulin (clone DM1A, T6199-Sigma-Aldrich), rabbit anti-Ubiquitin (PA1-10023, Thermo Scientific), goat anti-Cathepsin D (AF1029, R&D Systems). Total protein stain was performed using the Revert™ 700 Total Protein Stain Kits for Western Blot Normalization of serum samples (926-11010 LI-COR Biosciences). Further details on primary antibodies used in this study are included in Supplementary Table 3. Fluorophore-conjugated secondary antibodies used were: Donkey Anti-Rabbit IgG Polyclonal Antibody (925-32213 IRDye® 800CW, LI-COR Biosciences; 1 to 10000 dilution), Donkey Anti-Mouse IgG Polyclonal Antibody (925-68072 IRDye® 680RD, LI-COR Biosciences; 1 to 10000 dilution), Donkey anti-goat IgG Polyclonal Antibody (925-32214 IRDye® 800CW, LI-COR Biosciences; 1 to 10000 dilution) and Donkey Anti-Sheep IgG Polyclonal Antibody (20062-1 CF™ 680, Biotium; 1 to 5000 dilution). Membranes were scanned using the Odyssey XF imaging system (LI-COR Biosciences).

## Analysis of Bis(Monoacylglycero)Phosphate (BMP) in the brain

Following transcardial perfusion, each brain was dissected sagittally and the frontal region, including the frontal cortex, was snap-frozen in liquid nitrogen and stored at −80 °C. Lipids were extracted from frozen brains following a published method[140]. Briefly, about 50 mg of brain tissue was homogenized in 150 µl KPBS with tissue disrupter and 25 µl of the homogenate were added to 1 mL lipid extraction buffer (Chloroform: Methanol, 2:1 v/v containing 750 ng mL$^{-1}$ of SPLASH LIPIDOMIX internal standard mix from Avanti, 330707 Sigma-Aldrich), followed by vortexing for 1 hour in a 4 °C cold room. Then, 200 µl of 0.9% (w/v) saline (VWR L7528) were then added to the mixture and vortexed for 10 minutes in a 4 °C cold room. The mixture was then centrifuged at $3000 \times g$ for 10 minutes at 4 °C to separate the methanol/saline (upper) and chloroform (lower) phases, the upper phase was discarded, and lipids-containing lower phase were collected to a new 1.5 mL tube and vacuum dried on SpeedVac. Lipids were reconstituted in 50 µL Acetonitrile:isopropanol:water 13:6:1 (v/v/v) for lipid quantitation as described below.

An Ascentis C18 column (5 Micron, 5 mum particle size, L x I.D. 5 cm×4.6 mm, 50530-U Sigma-Aldrich) with an Ascentis Express guard holder that is connected to a 1290 LC system was used to separate lipids. The LC system was coupled to the 6470 A triple quadrupole (QQQ) mass spectrometer to quantify targeted lipids as described[141].

The mass spectrometer was operated in Multiple Reaction Method (MRM) for targeted analysis of species of interest. Standard BMP 18:1/18:1 (Avanti) was optimized using a MassHunter Optimizer software. The precursor-product ion pairs (m/z) used for MRM of the compounds were as follows: BMP 16:0/16:0 MNH$_4^+$, 740.54-313.27; BMP 16:0/18:1 MNH$_4^+$, 766.56→338.9/313.27; BMP 16:0/22:6 MNH$_4^+$, 812.54→385.27/313.27; BMP 16:1/18:1 MNH$_4^+$, 764.54→338.9/311.26; BMP 16:1/18:2 MNH$_4^+$, 762.54→336.9/311.26; BMP 16:1/20:3 MNH$_4^+$, 788.6→362.96/311.26; BMP 16:1/22:6 MNH$_4^+$, 810.53→385.27; BMP 18:0/18:1 MNH$_4^+$, 794.6→340.9/338.9; BMP 18:1/18:1 MNH$_4^+$, 792.6→338.9; BMP 18:1/18:2 MNH$_4^+$, 790.56→338.9/336.9; BMP 18:1/20:3 MNH$_4^+$, 816.56→363.27/338.9; BMP 18:1/20:4 MNH$_4^+$, 814.56→361.27/338.9; BMP 18:1/22:6 MNH$_4^+$, 838.56→385.27/338.9; BMP 18:2/20:3 MNH$_4^+$, 814.56→363.27/336.9; BMP 18:2/20:4 MNH$_4^+$, 812.54→361.27; BMP 18:2/22:6 MNH$_4^+$, 836.54→385.27/336.9; BMP 20:4/22:6 MNH$_4^+$, 860.54→385.27/361.27; BMP 22:5/22:6 MNH$_4^+$, 886.56→387.31/385.27; BMP 22:6/22:6 MNH$_4^+$, 884.54→385.27 and POPC: 760.6 →124.9/60.2.

The annotation MNH4+ represents measurement in positive mode where M indicates the neutral mass while NH4+ is the ammonium adduct.

Lipids were annotated and quantified using a high-throughput analysis software called MassHunter for qualitative analysis and QQQ quantitative analysis software known as Quant-My-Way. The identification of BMP species in the figures was confirmed by manually verifying the peak alignment, matching retention times, and comparing MS/MS spectra to characteristic fragmentation patterns of standard compound. To ensure accurate identification and quantification, two transitions for each compound were analyzed, and their relative responses were compared. The MRM method and retention time were used for quantification of all lipid species with the quantification software. The raw abundances of all species were then exported to Microsoft Excel for further analysis. The abundances of BMP species were normalized to the abundance of endogenous control lipid in the same sample.

## Statistical analyses

All the data shown in the present manuscript are reported either as Mean ± standard deviation of the mean (SD) or Mean ± standard error mean (SE, behavioral analyses, n > 10). The number of sampled units, n, upon which we reported statistics is the single mouse for the in vivo experiments (one mouse is n = 1) or the number of independent biological replicates for the in culture experiment (one independent biological replicate n = 1). Mice that underwent neurobehavioral studies were serially evaluated over time. GraphPad Prism 7 software (GraphPad Software) was used for statistical analyses. Parametric tests were used with data having a normal distribution (evaluated using the Shapiro–Wilk test). The statistical tests used were: two-tailed unpaired t-test for two-group comparisons, one-way ANOVA with Tukey post hoc or Kruskal–Wallis test with Dunn's correction for comparisons with more than two groups, two-way ANOVA with Tukey or Sidak post-hoc for multiple variable comparisons and Wald with Benjamini–Hochberg post-hoc correction for pseudobulk DEG analyses as detailed in the figure legends. For all the data sets analyzed by parametric tests, alpha = 0.05. All statistical tests were performed two-sided. $p$-values < 0.05 were considered significant. The statistical analysis performed for each data set is indicated in the figure legends. For all figures *$p$ < 0.05, **$p$ < 0.01, ***$p$ < 0.001, ****$p$ < 0.0001. The exact $p$-values of all comparisons are reported in the Source Data file.

## Reporting summary

Further information on research design is available in the Nature Portfolio Reporting Summary linked to this article.

# Data availability

The data generated in this study are provided in the Source Data file. Single-cell transcriptomic data generated in this study are publicly available in the NCBI's Gene Expression Omnibus (GEO) database, accession number GSE261246 and GSE241877. Data generated in previous studies and used here as a comparison (GSE128855) are publicly available in GEO. Source data are provided with this paper.

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

## Acknowledgements

This work was supported by grant number 22022-310753 from the Chan Zuckerberg Initiative DAF, an advised fund of the Silicon Valley Community Foundation (N.G.-O.), MPSI Pilot Grant from the Orphan Disease Center University of Pennsylvania (N.G.-O.), Stanford's Innovative Medicines Accelerator (N.G.-O.) and Stanford's Maternal and Child Health Institute (N.G.-O., K.L.D., and M.A.-R.). K.N. is supported by the Sarafan ChEM-H Chemistry/Biology Interface Program as a Kolluri Fellow and the Bio-X Stanford Interdisciplinary Graduate Fellowship affiliated with the Wu Tsai Neurosciences Institute (Bio-X SIGF: Mark and Mary Steven's Interdisciplinary Graduate Fellow). M.A.-R. is a Stanford Terman Fellow and a Pew-Stewart Scholar for Cancer Research, supported by the Pew Charitable Trusts and the Alexander and Margaret Stewart Trust. K.L.D. is the Anne T. and Robert M. Bass Endowed Faculty Scholar in Pediatric Cancer and Blood Diseases and the Harriet and Mary Zelencik Endowed Faculty in Children's Cancer and Blood Diseases. We thank the Stanford Human Immune Monitoring Center (HIMC) for running the Luminex assay. Additionally, we acknowledge the support provided by Nay L. Saw, Gaku Ogawa, Rachel Lam, and Mehrdad Shamloo from Stanford's Behavioral and Functional Neuroscience Laboratory (SBFNL) for their assistance with Neurobehavioral analyses and intracerebroventricular injections. Lastly, we thank the Metabolomics Knowledge Center (MKC) at Sarafan ChEM-H and its director, Yuqin Dai, for their assistance.

## Author contributions

P.C. conceived and conducted the study, designed and performed the experiments, developed the methodology, carried out the analyses and the interpretation of results, created figures, and wrote and revised the

manuscript and figures. R.S. performed the bioinformatic analyses of the scRNA-seq data and contributed to the preparation of the related figures. M.V.S.-N. contributed to the isolation of cells for chimerism analyses and to mouse conditioning. K.N. performed and analyzed the lipid mass spectrometry data. J.X. performed lipid extraction for mass spectrometry analyses. J.S. performed CyTOF mass cytometry runs and analyses. L.N.P.-V. and J.A.B. contributed to the isolation of cells for chimerism analyses. A.L. contributed to the bioinformatic analyses of the sc-RNAseq data. M.C. performed the bioinformatic analyses of the scRNA-seq data and contributed to the preparation of the related figures. K.L.D. supervised, financed, analyzed, and interpreted the CyTOF mass cytometry data. M.A.-R. supervised, financed, analyzed, and contributed to the mass spectrometry analyses. N.G.-O. conceived and directed the study, provided funding, assisted with experimental design, and wrote and revised the manuscript and figures.

## Competing interests

The authors declare no competing interests.
