## [Peer Review File · Nature Communications]

CNS-wide repopulation by hematopoietic-derived microglia-like cells corrects progranulin deficiency in miceREVIEWER COMMENTS

Reviewer #1 (Remarks to the Author):

This well written, rigorous manuscript from Pasqualina Colella, et al. has three major contributions. First they describe the efficacy of a novel conditioning regimen for hematopoietic stem cell transplantation (HSCT) that results in robust repopulation of brain microglia-like cells. Second, they present a detailed scRNAseq analysis of the transcriptomic profile of the results microglia-like cells in comparison to microglia, BM cells, and border-associated macrophages. Finally, they use the new HSCT protocol to treat progranulin deficiency related neurodegeneration.

HSCT triggers bone marrow-derived cells to migrate to the CNS where they become microglia-like cells (MGLCs) following myeloablative conditioning of the recipient. CNS-penetrating DNA alkylating drugs, like Busulfan (BU), are used for myeloablation in the clinic. Despite these advances, even at the highest tolerated BU dose in conjunction with HSCT, results in low and variable engraftment of MGLCs and limits the therapeutic efficacy of HSCT. To overcome this, the role of microglia depletion by reducing colony-stimulating factor 1 receptor (CSF1R) through CSF1R inhibitors (CSF1Ri; PLX5622 and PLX3397) has been examined to overcome the limited engraftment of bone marrow derived MGLCs in the CNS. Combination therapeutics using CSF1Ri along with myeloablative total body irradiation or BU has led to near-complete replacement of microglia with bone-marrow derived MGLCs. However, these combination therapeutics need significant optimization before use in the clinic. PLX3397 is an FDA-approved CSF1Ri for tenosynovial giant cell tumor. The authors defined 1. Efficacy with fewest dose and duration of treatment, (Fig 1a description) 2. Potential toxicities through examining hematopoiesis and neurobehavior (Fig 2i-o), and 3. Kinetics, signaling molecules, and characteristics of the cells associated with brain repopulation following CSF1Ri (Fig 1i, fig 3, fig 4 and fig 5). Importantly, they identify several surface markers of MGLC that can be used to differentiate them from host MG. They then apply their BU and PLX337 regimen with transplantation of wild-type cells to a mouse model with progranulin deficiency (Grn^{-/-}) and show a restoration of progranulin protein levels which, in turn, corrects brain lipid metabolism.

Overall, the questions are interesting and important, the experimental design is rigorous, the data are clear and convincing, and the writing is excellent. I would like to see a bit more description of the effects of HSCT on the progranulin mice. They focus on the BMP as a primary outcome, and this is an interesting and important outcome measure in these mice. They don't see any of the previously described behavioral phenotypes (Ext Fig 10), which is unfortunate but not a major concern. However, they don't show any histopathology, and it would be typical with these models to show effects of their treatment on microgliosis and lipofuscinosis, which are robust abnormalities in GrnKO mice. Reduced survival is also seen in most cohorts, if the N is large enough (it may not be here); was this examined?

Minor concerns:

1. In figure 1h, images are not from the same area of the olfactory bulb, hippocampus, or meninges when looking at BU compared to BU+PLX. This is an issue because, for example, different areas of the hippocampus have more microglia compared to others. Would like to see images that are clearly from parallel brain regions for the two conditions. As an alternative, a 300um image of BU alone would be nice in Fig 1g. Could consider replacing the 100um images with this. The effect is obvious, and the zoom doesn't add much to the story.
2. Figure 1 is out of order, you could make flow graph "d" and move it up, then make the quant from flow "e" and move "e" and "f" down.
3. Why is the data for BU alone not shown in figure 1j? The legend says the stats are against untreated mice but on the figure it says vs. BU, which is correct? Why is the blue color scheme for BU+PLX not used here, with black instead?
4. On line 160-161: "BU administration alone did not significantly reduce the number of microglia (86 ± 12%, D14 Fig. 1k)" I think this is mislabeled and supposed to correspond to figure 1j, although as above 1j doesn't show the BU alone.
5. Fig 3e/g, what are the units for the mean expression (0-4)?

6. Line 316, what disease?

7. Since there are BMP data from wild-type mice, include on the heatmap in figure 6i and show as absolute, instead of relative levels. It is hard to interpret if treatment + BMT restores BMP to wild-type levels, and this presentation obscures the typically large differences in abundance of different BMP species.

8. What brain region is the western and ELISA data from in Fig 6b-e? If not from the same area as the BMP data, do we know that treatment restores progranulin levels significantly in the region where BMP data were collected?

Reviewer #2 (Remarks to the Author):

Reviewer #3 (Remarks to the Author):

This manuscript by Colella et al. is focused on microglia replacement via hematopoietic stem cell transplantation (HST). The authors show that combining busulfan conditioning with a PLX CSF1R inhibitor results in enhanced BM engraftment in the brain. Next, they profile the BM-derived microglia via scRNA-seq and report that they exhibit a mixed microglia/BAM-like phenotype. Finally, they show that their transplantation approach increases the amounts of Bis(Monoacylglycerol)Phosphates (BMP) in the brain. While the rationale of microglia replacement to treat neurodegenerative diseases is very relevant and timely, the current manuscript does not offer any new conceptual advance in the field and there are clear concerns. The only aspect that would have been innovative, is a detailed exploration of using microglia replacement for treating progranulin deficiency. However, the results on this front were very limited. It often feels as if the manuscript was rushed, and in its current form is not suitable for publication in a journal as Nature Communications.

Main comments with regard to insufficient conceptual advance and incompleteness:

1) It is already known since 2018 that PLX treatment following (HST) strongly increases BM-microglia engraftment and many articles have come out since then using this approach, with most, but not all, of which referenced by the authors. The authors claim that the innovation lies in the fact that they have now optimized the PLX treatment protocol for highest translational potential. However, this claim is very unconvincing. Firstly, it is strange and problematic that in Figure 1B there are only 2 datapoints for the BU+PLX condition, which is the most important condition. These experiments were also not repeated. Furthermore, the same 2 datapoints seem to be re-used in Extended figure 1d. In Extended Figure 1d the authors show that placing BU conditioned mice on PLX chow for 4 weeks (diet1x, diet2x) does not significantly increase BM-microglia engraftment. This is in full contradiction with work by others. For example, PMID: 35190726 has shown that BU + 4 weeks of 290 ppm PLX3397 results in > 90% BM-microglia engraftment. And PLX3397 chow at around 580 ppm is known to deplete the vast majority of microglia within 1 week PMID: 30370589, the same timing as with the author's oral gavage. It is thus completely unclear why the diet approach did not work for the authors. Did the authors actually check whether their PLX chow was depleting microglia?

Finally, the rationale that a shorter-term treatment with high doses of PLX3397 would a priori be more clinically relevant than a longer treatment with a lower dose is false. High doses may be more toxic than lower doses. Thus, a lower dose over a longer period may give the same level of engraftment, but be accompanied with less side-effects. In conclusion, the authors did not really optimize anything in terms of clinical translation, they just provide one oral gavage regimen that works (they did not titrate dosages), compared this to PLX diet data that are in contradiction with

the literature, and did not assess overall toxicity.

The results of Figure 2a-j show that PLX3397 also affects peripheral myeloid cells (24h post treatment) and that macrophages in peripheral organs are efficiently replaced following HST. These are known and reported facts (eg. PMID: 32900927). Additionally, it is strange that the authors use single markers to denote immune populations. For example, a Ly6C⁺ cell is not always a monocyte, Ly6C is also expressed on neutrophils and various lymphocytes. The authors need to use a gating strategy.

2) The authors performed transcriptional profiling of BM-microglia. This did not provide novel findings, as this has been already performed by several other groups, both at bulk and single-cell level (pmid: 29643186, pmid: 35294256, pmid: 32783928, pmid: 30523248). The genes and signatures that the authors describe are well known to be expressed in BM-derived microglia. The authors provide a very long description of the single-cell data, spread over 3 paragraphs and 3 figures. This feels as a filler.

3) Microglia replacement was performed in Grn^{-/-} mice, as a model for FTD. Regretfully, this section, which could have been innovative, was brief and inconclusive. First, the authors only incorporated the BU + PLX condition. GRN is known to be produced by hematopoietic cells in the periphery, which the authors also report. Therefore, the observed GRN protein levels in the brain of BU+PLX mice and the rescue of BMP may in part be due to peripheral GRN production. Since PLX is not needed for peripheral hematopoietic reconstitution, it is important to include the BU only condition for all read-outs. This will show to what extent efficient microglia replacement is actually necessary for rescue or that peripheral GRN is sufficient. Furthermore, currently there is no proof that the measured GRN is derived from microglia. The GRN could have originated from BAMs or other immune cells present in the border regions of the brain, or could have reached the brain via the periphery.

Secondly, the authors only assessed rescue of BMP levels. It is known that Grn^{-/-} mice develop pathology, although this only becomes substantial at late stages (> 18 months of age). It is known that 12 months is too early to observe robust deficits in Grn^{-/-} mice.

In conclusion, the strong statement made in the title "CNS Repopulation by Hematopoietic-Derived Microglia-Like Cells Corrects Progranulin deficiency", is not supported at all by the currently provided data.

Reviewer #4 (Remarks to the Author):

The study of microglia repopulation by bone marrow-derived stem cells is of high interest because of the therapeutic implications of this phenomenon that has been observed in pre-clinical and clinical data for neurological diseases. However, the mechanisms involved in the migration and differentiation of hematopoietic-derived MGLCs in the CNS are still under investigation. This comprehensive study not only provides an optimized protocol to achieve almost full replacement of microglia by microglia-like cells (MGLCs) in a very timely manner using a combination of CNS-penetrant myeloablation regimen Bu or TBI, and the CSF1R inhibitor PLX3397, but Colella and colleagues also studied the transcriptional and proteomic identity of the MGLCs. Indeed, they performed single cell RNAseq and high-dimensional CyTOF mass cytometry showing that MGLCs have a hybrid microglia/brain border-associated macrophage (BAM) transcriptional and translational identity. Importantly, they have identified several surface markers specific of MGLCs so that can be used to differentiate them from host MGs. The authors also identified chemokines that increased in the brain prior to the MGLC engraftment and may represent signals for the recruitment, expansion, or maturation of MGLC progenitors in the brain. Finally, they validated their conditioning strategy in a mouse model of GRN deficiency transplanted with wild-type bone marrow cells.

Below my minor and major comments:

Figure 1b, it looks like there are only 2 mice in the group BU+PLX. No statistical analysis can be performed with 2 mice, and if this is the case, additional mice have to be added in this treatment group at the 3-month timepoint. Also, the mice numbers are difficult to distinguish in some graphs, adding the n number in the legend would be clearer.

Figure 1d, there are 2 distinct groups in the GFP+ engraftment cells in the BU group. Is there a sex difference? Busulfan impact can be influenced by sex, so the authors should carefully analyze and show whether sex had no impact in the results by adding more information on the sex of the mice used (males in blue dots and female in red for instance).

The discrimination index should be provided for the NORT test.

In figures 4 and 5, the authors introduce the disease MG as coming from a neuropathic LSD. While references are present, more information should be provided in the manuscript on this disease and the rationale for using this model as control group. This said, this group do not add much information in an already very comprehensive manuscript and may be removed.

In figure 4, MHCII is shown to be highly expressed in CD45+ CD11b+ MGLCs in mice receiving BU alone but not in the MGLCs in mice receiving BU+PLX. The authors should discuss this very intriguing data.

Fig 5f is referenced in line 374, but there is no figure 5f panel.

Figure 6a shows that MGLCs expresses high level of Grn, but not that they secrete GRN. The sentence in lines 386-387 should be revised accordingly.

To validate that BU+PLX was superior to BU alone for the treatment of GRN-deficiency using bone marrow cell transplant, both strategies should have been used simultaneously and the outcomes compared. Indeed, partial reconstitution of GRN led to normalized BMP lipid metabolism in Grn-/- mice receiving BU + PLX and WT BMT. Similar result may have been achieved with BU alone.

Define MLD (line 438).

Line 478: the authors probably mean to say "insufficient".

Reviewer #5 (Remarks to the Author):

Overall, this is an interesting study with a lot of data and appropriate controls, however some points should be clarified as follow:

- 1) In Fig 1 I-o: there is no discussion of what cells are secreting the cytokines. There is no mention of where they could come from, could neuron/astrocytes be involved in attracting new microglia-like cells?
- 2) The single-cell transcriptomics analysis is interesting and there is a lot of effort to characterize the signature of these infiltrating cells in general. However, since these cells are not entirely microglial cells, it would be interesting to look at hematopoietic cell or monocyte markers
- 3) The UMAP of host microglia GFP- is confusing in Fig. 3 and 4. UMAP for host microglia GFP- looks stretched.
- 4) One population, cluster 8, is the only one that expresses Sall1, a transcriptional factor essential for microglial identity, and does not express BAM markers. There is not a clear discussion around cluster 8, while it seems to be present across all experimental groups, including in the GFP+ cells. Discussing the heterogeneity of the response to infiltrated cells would be interesting. In addition,

some staining to locate this population in BU+PLX+BM transplant mice would be interesting to see if it is regionally localized or spread out across the tissue.

5) The host microglia and the naive microglia are transcriptomically different. It would be nice to discuss this and to compare them, to see if the treatment inherently changes their transcriptome. It is something that it would be interesting to know for the prospect of the treatment becoming a potential therapy.

6) The number of samples (N = 1) for the scRNA sequencing is low. Having replicates would confirm the solidity of the data.

7) How many cells from each mouse were obtained for scRNA sequencing? The number must be stated.

8) Please clarify how the GFP- host MG cells were obtained. Were they from a different mouse than the GFP+ cells, and if yes, were the mice conditioned with BU+PLX prior?

9) BAMs signature is shown to be expressed with primarily antigen-expressing markers. In the Van Hove et al., 2019 paper that is cited, several subtypes of BAM are characterized with distinct signatures. Is the BAM signature observed in the GFP+ cells related to a BAM subtype in particular, and if not what does it mean?

10) There is a claim that there is a proliferative signal from cytokines and signaling molecules, however in Fig 3e, none of the proliferative markers are high in MGLCs. Please, clarify?

Since the goal of the paper is to emphasize how such a regimen could be an alternative for human microglia transplant, it would be nice to emphasize how this could be translated to humans in the discussion.

1) no behavioral changes and only 20% alteration in GRN specifically.

2) No known evidence / not cited that PLX can also deplete microglia in humans.

3) Great proof of concept but not "immediately translatable".

Pitfall of study that the authors must address:

The use of enzymes for digestion and preparation for sorting alters microglia/macrophages reactivity, thus the interpretability of the identity (specifically the cytokines signature) is difficult.

Minor comment:

Typo in line 90, PLX 3337 instead of PLX 3397.

Natalia Gomez-Ospina, M.D., Ph.D.
Taube Pediatric Neurodegenerative Disease Scholar
Assistant Professor, Department of Pediatrics
Divisions of Medical Genetics, and Stem Cell Transplantation
Lokey Stem Cell Biology Building, Stanford, CA 94305
[*gomezosp@stanford.edu*](mailto:gomezosp@stanford.edu)

March 26th, 2024,

Point-by-point Response to Reviewer's Comments

Reviewer 1

This well written, rigorous manuscript from Pasqualina Colella, et al. has three major contributions. First they describe the efficacy of a novel conditioning regimen for hematopoietic stem cell transplantation (HSCT) that results in robust repopulation of brain microglia-like cells. Second, they present a detailed scRNAseq analysis of the transcriptomic profile of the results microglia-like cells in comparison to microglia, BM cells, and border-associated macrophages. Finally, they use the new HSCT protocol to treat progranulin deficiency related neurodegeneration.

HSCT triggers bone marrow-derived cells to migrate to the CNS where they become microglia-like cells (MGLCs) following myeloablative conditioning of the recipient. CNS-penetrating DNA alkylating drugs, like Busulfan (BU), are used for myeloablation in the clinic. Despite these advances, even at the highest tolerated BU dose in conjunction with HSCT, results in low and variable engraftment of MGLCs and limits the therapeutic efficacy of HSCT. To overcome this, the role of microglia depletion by reducing colony-stimulating factor 1 receptor (CSF1R) through CSF1R inhibitors (CDF1Ri; PLX5622 and PLX3397) has been examined to overcome the limited engraftment of bone marrow derived MGLCs in the CNS. Combination therapeutics using CSF1Ri along with myeloablative total body irradiation or BU has led to near-complete replacement of microglia with bone-marrow derived MGLCs. However, these combination therapeutics need significant optimization before use in the clinic. PLX3397 is an FDA-approved CSF1Ri for tenosynovial giant cell tumor. The authors defined 1. Efficacy with fewest dose and duration of treatment, (Fig 1a description) 2. Potential toxicities through examining hematopoiesis and neurobehavior (Fig 2i-o), and 3. Kinetics, signaling molecules, and characteristics of the cells associated with brain repopulation following CSF1Ri (Fig 1i, fig 3, fig 4 and fig 5). Importantly, they identify several surface markers of MGLC that can be used to differentiate them from host MG. They then apply their BU and PLX337 regimen with transplantation of wild-type cells to a mouse model with progranulin deficiency (Grn^{-/-}) and show a restoration of progranulin protein levels which, in turn, corrects brain lipid metabolism.

Overall, the questions are interesting and important, the experimental design is rigorous, the data are clear and convincing, and the writing is excellent. I would like to see a bit more description of the effects of HSCT on the progranulin mice. They focus on the BMP as a primary outcome, and this is an interesting and important outcome measure in these mice. They don't see any of the previously described behavioral phenotypes (Ext Fig 10), which is unfortunate but not a major concern. However, they don't show any histopathology, and it would be typical with these models to show effects of their treatment on microgliosis and lipofuscinosis, which are robust abnormalities in in

GrnKO mice. Reduced survival is also seen in most cohorts, if the N is large enough (it may not be here); was this examined?

Responses to Comments by Reviewer 1

We sincerely thank Reviewer 1 for the positive comments and for acknowledging the clarity, rigor, novelty, relevance, and contributions of the data presented in this manuscript. We are also thankful for the helpful suggestions.

Major Comment 1: I would like to see a bit more description of the effects of HSCT on the progranulin mice. They focus on the BMP as a primary outcome, and this is an interesting and important outcome measure in these mice. They don't see any of the previously described behavioral phenotypes (Ext Fig 10), which is unfortunate but not a major concern. However, they don't show any histopathology, and it would be typical with these models to show effects of their treatment on microgliosis and lipofuscinosis, which are robust abnormalities in in Grn KO mice.

Response: We sought additional evidence of disease correction by examining robust disease markers in GRN-deficient mice at around six months of age, which aligns with our study's time point of analysis (2-month-old mice analyzed four months after bone marrow transplant). While most brain histopathological defects, including microgliosis, have been consistently reported in older GRN-deficient mice (≥ 9 -12 months), lipofuscin storage, as well as proteostasis defects (e.g., maturation of lysosomal proteases and protein ubiquitination), have been reproducibly observed to begin in young GRN-deficient mice and progress with age.

We examined the accumulation of lipofuscin in the CA3 region of the hippocampus and observed a complete correction in Grn^{-/-} mice treated with BU+PLX and WT BMT (see New Fig. 6n-o). We also found that abnormalities in protein ubiquitination and CATHEPSIN D (CTSD) maturation were improved in Grn^{-/-} mice following treatment with BU+PLX and WT BMT (see New Fig. 7d-f).

All these data are included in the Revised version of the manuscript lines, as described below:

Line 442: "Studies on the brain histopathology of GRN-deficient mice have shown a late-onset appearance of several disease markers, such as microgliosis and astrogliosis (≥ 9 -12 months of age)^{45,57,103-106}. However, lipofuscin storage in the CA3 region of the hippocampus, a well-known disease biomarker, has been detected in both young and aged GRN-deficient mice¹⁰⁶⁻¹⁰⁹. We confirmed a significant increase in CA3 lipofuscin deposits in Grn^{-/-} compared to WT mice at six months of age. Notably, lipofuscin storage was cleared in Grn^{-/-} treated with BU+PLX and WT BMT, while no significant decrease was observed in Grn^{-/-} mice after BU conditioning and WT BMT (Fig. 6n-o)."

And

Line 455: "Proteostasis defects such as altered maturation of lysosomal proteases and accumulation of ubiquitinated proteins have been reported in young and aged Grn^{-/-} mice and in FTD-GRN individuals^{55,110-112}. We confirmed an increased storage of ubiquitinated proteins and altered CATHEPSIN D (CTSD) maturation in brain lysates of Grn^{-/-} mice compared to WT. Both defects were normalized in Grn^{-/-} mice treated with BU+PLX and WT BMT (Fig. 7d-f)."

Major Comment 2. Reduced survival is also seen in most cohorts, if the N is large enough (it may not be here); was this examined?

Response: We did not observe a decrease in the survival of Grn^{-/-} mice (n=12) compared to WT mice (n=15) over a 13-month observation period (neurobehavioral study depicted in Supplementary Fig. 16). In this experiment, one WT and one KO mice died at 12 months of age of unknown cause. In the bone marrow transplant experiments depicted in Figs 6 and 7, Grn^{-/-} mice were treated at two months of age and euthanized four months later, at six months of age. No fatalities were observed in the KO mice (sham + untreated n=11, treated by WT bone marrow transplant n=12). These survival data are now mentioned in line 480, and a graph is included in Supplementary Fig 16.

Minor comments

1. In figure 1h, images are not from the same area of the olfactory bulb, hippocampus, or meninges when looking at BU compared to BU+PLX. This is an issue because, for example, different areas of the hippocampus have more microglia compared to others. Would like to see images that are clearly from parallel brain regions for the two conditions. As an alternative, a 300um image of BU alone would be nice in Fig 1g. Could consider replacing the 100um images with this. The effect is obvious, and the zoom doesn't add much to the story.

Response: We updated the images to compare the same brain regions for the two conditions (BU alone vs. BU+PLX3397, New Figure 1h).

2. Figure 1 is out of order, you could make flow graph "d" and move it up, then make the quant from flow "e" and move "e" and "f" down.

Response: The panels have been modified accordingly.

3. Why is the data for BU alone not shown in figure 1j? The legend says the stats are against untreated mice but on the figure, it says vs. BU, which is correct? Why is the blue color scheme for BU+PLX not used here, with black instead?

Response: We have substantially modified this graph to show the depletion and re-appearance of CD45+ CD11b+ cells in BU and BU+PLX-treated mice compared to untreated mice. **Both conditions are now included in the revised Fig 1j.** Notably, earlier time points (days 21, 24, 28) in the BU-conditioned mice were not collected because, without PLX, we did not expect significant microglia depletion. The results section has been revised as follows:

Line 157: "To examine the kinetics of microglia depletion and brain repopulation, we looked at freshly isolated microglia preparations at 1, 4, 8, 20, 70 and 190 days after PLX withdrawal (corresponding to 21, 24, 28, 40, 90 and 210 post-BMT, respectively, **Fig. 1i**). Flow cytometry analyses showed acute and marked microglia depletion in the BU+PLX group (~90% at D21 post-BMT, **Fig. 1j**). BU alone depleted the MG niche partially and at a much slower rate (~19% at D40 and 50% at D90 and D210, **Fig. 1j**)."

4. On line 160-161: “BU administration alone did not significantly reduce the number of microglia ($86 \pm 12\%$, D14 Fig. 1k)” I think this is mislabeled and supposed to correspond to figure 1j, although as above 1j doesn’t show the BU alone.

Response: It was mislabeled. As discussed in the previous answer, these data have been replotted and re-written to clarify and address the reviewer’s concerns.

5. Fig 3e/g, what are the units for the mean expression (0-4)?

Response: In the revised version of the manuscript, we have defined the units for the mean expression in the figure legends and the methods section.

See Legends for Figure 3-5. “The dot size indicates the percentage of cells expressing the gene in each sample/cluster, while the color scale represents the mean gene expression calculated as the mean log-normalized UMI counts for each gene of interest. Dendrograms at the right show the clustering of the samples based on the expression profiles of the depicted genes. “

6. Line 316, what disease?

Response: In response to a suggestion by Reviewer 4, we have removed the data from this disease model. The disease microglia in the first submission were derived from a mouse model of Mucopolysaccharidosis type 1, another lysosomal storage disorder.

7. Since there are BMP data from wild-type mice, include on the heatmap in figure 6i and show as absolute, instead of relative levels. It is hard to interpret if treatment + BMT restores BMP to wild-type levels, and this presentation obscures the typically large differences in abundance of different BMP species.

Response: The analysis of individual BMP species is reported in the Source data. The reviewer is correct in that there are significant differences in the abundance of various BMP species. Indeed, analysis of individual BMP species reported in the Source data reveals that some species are more effectively corrected than others.

Due to the high abundance of BMP 16:0/16:0 and 16:0/18:1 compared to other BMPs (20 to 1000-fold higher), compiling the data in a heatmap does not effectively visualize differences among treatment groups. After numerous attempts at plotting the data, we found that representing BMP values relative to the wild type allows us to summarize it without flattening the differences among treatment groups (Fig 7g of the revised manuscript). To simplify and as suggested by our expert collaborators, who conducted and analyzed the BMP mass spectrometry data, we reported the quantification of Total BMP species (Fig. 6j, now Fig. 6h).

8. What brain region is the western and ELISA data from in Fig 6b-e? If not from the same area as the BMP data, do we know that treatment restores progranulin levels significantly in the region where BMP data were collected?

Response: The brains were dissected on the sagittal plane, and the brain frontal region, including the frontal cortex, was used for BMP measurements, while the remainder of the ipsilateral part of the brain was used for Western blot and ELISA assays. To confirm that the treatment restores progranulin levels in the exact region used for BMP analysis, we quantified GRN in the homogenates used for the BMP analyses by ELISA and Western blot. Importantly, these analyses showed that GRN was restored in the BMP homogenates in similar amounts to the brain lysates quantified in the original version of the manuscript (new Supplementary Fig. 15b-d). This data is consistent with the efficient repopulation of the brain frontal cortex by MGLCs on Grn^{-/-} mice (new Fig. 7i).

Reviewer 2

Reviewer 3

This manuscript by Colella et al. is focused on microglia replacement via hematopoietic stem cell transplantation (HST). The authors show that combining busulfan conditioning with a PLX CSF1R inhibitor results in enhanced BM engraftment in the brain. Next, they profile the BM-derived microglia via scRNA-seq and report that they exhibit a mixed microglia/BAM-like phenotype. Finally, they show that their transplantation approach increases the amounts of Bis(Monoacylglycero)Phosphates (BMP) in the brain. While the rationale of microglia replacement to treat neurodegenerative diseases is very relevant and timely, the current manuscript does not offer any new conceptual advance in the field and there are clear concerns. The only aspect that would have been innovative, is a detailed exploration of using microglia replacement for treating progranulin deficiency. However, the results on this front were very limited. It often feels as if the manuscript was rushed and, in its current form, is not suitable for publication in a journal such as Nature Communications.

Responses to comments from Reviewer 2/3

We thank the peer-in-training reviewer pair for their in-depth review of our manuscript. While the comments relating to innovation and conceptual advances are not all addressable through additional experimentation, we have included additional data and substantially modified the manuscript considering them. The reviewer's comments have been broken down to address a single point at a time.

Major comment 1a. It is already known since 2018 that PLX treatment following (HST) strongly increases BM-microglia engraftment. Many articles have come out using this approach, with most, but not all, of which referenced by the authors.

Response: We apologize for missing some crucial references. To ensure that any work that combines 1) conditioning (busulfan or radiation) with 2) pharmacological inhibition of CSF1R (PLX3397 or PLX5622) and 3) transplantation is cited, we reviewed the published literature and added additional citations. Notably, we found two articles published in high-profile journals since we prepared the initial manuscript, both using Busulfan and PLX5622 complexed with chow for Alzheimer's (Cell Stem Cell) and autoimmune encephalomyelitis (Nature Neuroscience), which we think **validate the great interest in the application of similar protocols to treat neurological diseases despite the concept per se not being novel**. We hope that the opening sentence, "Current protocols for replacing microglia with bone marrow-derived cells in mouse models, differ in the type of CSF1R inhibitor used, its formulation, the time of initiation, which can range from 14 days to several months, and the duration of administration, which typically lasts 2 to 4 weeks^{18,23,24,33,58-61}" acknowledges the prior work. We will add any other references the reviewer thinks should be included.

Major comment 1b. The authors claim that the innovation lies in the fact that they have now optimized the PLX treatment protocol for highest translational potential. However, this claim is very unconvincing. Firstly, it is strange and problematic that in Figure 1B there are only 2 datapoints for the BU+PLX condition, which is the most important condition. These experiments were also not repeated. Furthermore, the same 2 datapoints seem to be re-used in Extended figure 1d.

Response: We foremost want to make clear that the experiment combining BU+PLX has been repeated in at least five independent transplantation studies throughout the paper. Most notably, the comparison with Busulfan was repeated as an independent experiment using n=10 mice/cohort (BU n=10 mice, BU+PLX n=10 mice). The results of this experiment are presented in Fig. 1d-e and Fig. 2.

Nevertheless, for the short-term transplantation study described in Fig 1b, we increased the number of mice treated with Busulfan+PLX3397 from n=2 to n=4 (new Fig. 1b). The two mice initially reported in the manuscript were included in the Extended Data Fig. 1d as an internal/positive control for the various PLX3397 regimens tested. In the revised version of the manuscript, the four mice are only reported in the main Fig. 1b. Importantly, we re-demonstrate minimal variability among the mice treated with the optimized 6-day PLX3397 gavage regimen (90 ± 3.2 , n=4, new Fig. 1b).

Major comment 1c. In Extended Figure 1d the authors show that placing BU conditioned mice on PLX chow for 4 weeks (diet1x, diet2x) does not significantly increase BM-microglia engraftment. This is in full contradiction with work by others. For example, PMID: 35190726 has shown that BU + 4 weeks of 290 ppm PLX3397 results in > 90% BM-microglia engraftment. And PLX3397 chow at around 580 ppm is known to deplete the vast majority of microglia within 1 week PMID: 30370589, the same timing as with the author's oral gavage. It is thus completely unclear why the diet approach did not work for the authors. Did the authors actually check whether their PLX chow was depleting microglia?

Response: We agree with the reviewer that this data is confusing and incomplete. Our goal is not to show the superiority of a gavage-based protocol with the PLX-complexed chow, as this would merit an entirely different study with many more conditions. Accordingly, we excluded this data from the new version of the manuscript.

For the reviewer's curiosity, we did assess the effectiveness of PLX3397 diets in depleting microglia. In pilot experiments, we observed a dose-dependent reduction of brain CSF1R in C57BL/6 mice conditioned for two weeks with PLX3397-complexed chow at 290 and 580 ppm (data depicted in panels a-b below). However, when we co-administered the identical diet batches with Busulfan, microglia replacement remained low (maximum of 22%, data depicted in panels d-e below). Furthermore, the 580 ppm diet combined with Total Body Irradiation (TBI, 10Gy) resulted in $81 \pm 6.4\%$ microglia replacement replicating previous studies (data depicted in panels f-g below, not included in the manuscript). Importantly, we achieved high bone marrow chimerism ($90 \pm 3\%$, data depicted in panels d-e below), confirming effective Busulfan conditioning.

Major comment 1d. Finally, the rationale that a shorter-term treatment with high doses of PLX3397 would a priori be more clinically relevant than a longer treatment with a lower dose is false. High doses may be more toxic than lower doses. Thus, a lower dose over a longer period may give the same level of engraftment but be accompanied by less side-effects. In conclusion, the authors did not really optimize anything in terms of clinical translation, they just provide one oral gavage regimen that works (they did not titrate dosages), compared this to PLX diet data that are in contradiction with the literature, and did not assess overall toxicity.

Response:

First, we want to emphasize that we are not using high doses of PLX3397. The food intake of an adult mouse is 4-6 g of chow/day [Bachvanov et al., Behav Genet. 2002 Nov;32(6):435-43; PMID 12467341]. Considering that complete microglia depletion is achieved with a 580-600 ppm PLX3397-complexed diet [Najafi et al., Glia. 2018 Nov;66(11):2385-2396], the chow-based PLX3397 dose/mouse corresponds to 145 ± 29 mg/kg/day (if the mouse weighs 0.020 kg), 116 ± 23 mg/kg/day (if the mouse weighs 0.025 kg), and 97 ± 19 mg/kg/day (if the mouse weighs 0.030 kg). Therefore, the 100 mg/kg/day dose we used does not represent a higher dose than the diet-based regimens typically applied for longer durations (1-4 weeks). Instead, it represents a more effective and controlled microglia depletion regimen. In addition, conditioning regimens based on PLX5622, described as a more potent derivative of PLX3397, are commonly administered to mice via drug-complex chow at 1200 ppm for 2-4 weeks [e.g., Shibuya, Y., et al. Sci Transl Med 14, eab19945 (2022)], therefore, at much higher doses and for more extended periods. To clarify this point, we included the following sentences in the Results and Discussion:

Line 100. "To establish a highly efficient and reproducible protocol with high translational potential, we optimized the route and duration of drug administration. We administered PLX3397 (PLX) to adult C57BL/6 mice by oral gavage at 100 mg/kg/day, a dose that was chosen based on the effectiveness of the 580-600 ppm complexed chow⁶² and the daily mouse food intake of 4-5 g/day⁶⁴."

Line 509. “The PLX3397 dose used in our study is well below what is prescribed for patients with TGCT (humans take ~400-800 mg/day orally for repeated cycles of 4 weeks³⁶, while mice take ~3 mg/day for six days).”

Second, it is hard to dispute that an oral gavage administration is more like dosing in humans than a drug complexed in the chow. For example, Turalio, PLX3397, is prescribed 250 mg orally twice daily. Although the doses are similar, the pharmacokinetics and pharmacodynamics are different. Bioavailability is likely greater with gavage, resulting in higher peak plasma levels and lower troughs, although the area under the curve might remain unchanged.

Third, we have added more information on the optimization process, particularly the duration, in Supplementary Figure 1.

Line 104: “We observed maximal depletion of CD45+ CD11b+ microglia cells by flow cytometry after a 6-day regimen of PLX administered by oral gavage ($95 \pm 2\%$ depletion vs. untreated mice, **Supplementary Fig. 1a-c**)”

Major comment 1e. The results of Figure 2a-j show that PLX3397 also affects peripheral myeloid cells (24h post treatment) and that macrophages in peripheral organs are efficiently replaced following HST. These are known and reported facts (eg. PMID: 32900927). Additionally, it is strange that the authors use single markers to denote immune populations. For example, a Ly6C+ cell is not always a monocyte, Ly6C is also expressed on neutrophils and various lymphocytes. The authors need to use a gating strategy.

Response: The rationale for the analyses presented in Fig. 2a-j is to evaluate the effects of PLX3397 conditioning on the frequency of major hematopoietic lineages for our specific regimen, considering likely differences in pharmacokinetics and dynamics, as discussed in a previous comment.

The study cited by the reviewer (PMID:32900927, Lei et al., Proc Natl Acad Sci U S A. 2020 Sep 22;117(38):23336-2333) did not use PLX3397 but instead employed a 1200 ppm PLX5622 complexed chow for 3 weeks, which is known to have the highest side effects on hematopoiesis.

We acknowledge the Reviewer's comment regarding the specificity of the Ly6C marker and have revised the text to refer only to Ly6C+ cells (the gating strategy is shown in Supplementary Fig. 6a). As suggested by the Reviewer, to better define the Ly6C+ population, we reanalyzed the data with a gating strategy that distinguishes lymphocytes, monocytes, and granulocytes in mouse peripheral blood (PB) based on their relative size and granularity [Liu et al., STAR Protoc 1, 100029 (2020); Proserpio et al., Methods Mol Biol 2386, 27-41 (2022)]. This gating strategy showed a significant decrease in circulating Ly6c+ CD11b+ granulocytes, Ly6C+ CD11b+ and CD11b- monocytes, and the less abundant Ly6C+ CD11b+ Lymphocytes 24 hours after PLX3397 withdrawal (new Supplementary Fig. 6 b-d). The updated results are described in lines 209-216 of the manuscript.

The observed decrease in Ly6C+ cells in peripheral blood is consistent with previous findings, acknowledged in the original, which reported a decrease in Ly6C+ cells specifically in the peripheral blood of mice treated with a 290 ppm PLX3397-complexed chow for 3 weeks (Szalay et al., Nat Commun 7, 11499 (2016)). The fact that a previous study using a different PLX3397 regimen reported similar findings does not diminish the relevance of our conclusions but rather supports the robustness of the effects we observed using a different PLX3397 regimen.

Major comment 2. The authors performed transcriptional profiling of BM-microglia. This did not provide novel findings, as this has been already performed by several other groups, both at bulk and single-cell level (pmid: 29643186, pmid: 35294256, pmid: 32783928, pmid: 30523248). The genes and signatures that the authors describe are well known to be expressed in BM-derived microglia. The authors provide a very long description of the single-cell data, spread over 3 paragraphs and 3 figures. This feels as a filler.

Response: Reviewer 2/3 holds a different opinion on the novelty and significance of the sc-RNA sequencing than the other three reviewers. Other reviewers have put great interest in this data and asked for further analysis, indicating that they do not consider it filler.

Our work reports for the first time the single-cell transcriptional analysis of HSPC-derived microglia-like cells (MGLC) engrafted in the brain after Busulfan myeloablation and PLX3397. Furthermore, we are the first to use high-dimensional CyTOF analysis to analyze these cells.

The articles mentioned by the Reviewer, all cited in our original manuscript, differ from our work since they use either: 1) bulk RNA sequencing, making it impossible to evaluate the single-cell heterogeneity of the MGLCs [PMID: 29643186, Cronk et al., J Exp Med 2018 Jun 4;215(6):1627-1647; PMID 35294256 Shibuya et al. Sci Transl Med. 2022 Mar 16;14(636):eabl9945]; or 2) Total body irradiation (TBI), not Busulfan, to condition the mice pre-transplant [PMID: 32783928, Xu et al., Cell Rep 2020 Aug 11;32(6):108041]. TBI is non-clinically relevant, and it is known to induce significantly higher brain inflammation than Busulfan, affecting the transcriptional signature of MGLC differently.

One of the articles cited by the Reviewer didn't use CSF1R inhibition [PMID: 30523248, Shemer et al., Nat Commun 2018 Dec 6;9(1):5206].

A list of key differences between our work and the articles cited by the Reviewer are listed below:

1. PMID: 29643186 [Cronk et al., J Exp Med 2018 Jun 4;215(6):1627-1647] reports on MGLC bulk RNASeq after TBI plus PLX5622.
2. PMID 35294256 [Shibuya et al. Sci Transl Med. 2022 Mar 16;14(636):eabl9945] reports on MGLC bulk RNASeq after Busulfan plus PLX5622.
3. PMID: 32783928 [Xu et al., Cell Rep 2020 Aug 11;32(6):108041] reports on MGLC scRNASeq after TBI plus PLX5622 and the transplant of CX3CR1^{+/GFP} bone marrow.
4. PMID: 30523248 [Shemer et al., Nat Commun 2018 Dec 6;9(1):5206] reports on MGLC bulk RNASeq after TBI without any CSF1R inhibition.

Beyond these key methodological differences between our work and the articles mentioned by the Reviewer, our analysis of the transcriptional signature of the MGLCs at the single-cell level reveals many novel findings about the phenotype and heterogeneity of MGLCs engrafted in the brain as compared to the existing literature. The scRNAseq data analysis is summarized in three main figures (Fig. 3-5), and further detailed in six supplementary figures.

Major comment 3. Microglia replacement was performed in Grn^{-/-} mice, as a model for FTD. Regretfully, this section, which could have been innovative, was brief and inconclusive. First, the authors only incorporated the BU + PLX condition. GRN is known to be produced by hematopoietic cells in the periphery, which the authors also report. Therefore, the observed GRN protein levels in the brain of BU+PLX mice and the rescue of BMP may in part be due to peripheral GRN production.

Since PLX is not needed for peripheral hematopoietic reconstitution, it is important to include the BU only condition for all read-outs. This will show to what extent efficient microglia replacement is actually necessary for rescue or that peripheral GRN is sufficient. Furthermore, currently there is no proof that the measured GRN is derived from microglia. The GRN could have originated from BAMs or other immune cells present in the border regions of the brain, or could have reached the brain via the periphery.

Secondly, the authors only assessed rescue of BMP levels. It is known that *Grn*^{-/-} mice develop pathology, although this only becomes substantial at late stages (> 18 months of age). It is known that 12 months is too early to observe robust deficits in *Grn*^{-/-} mice.

In conclusion, the strong statement made in the title “CNS Repopulation by Hematopoietic-Derived Microglia-Like Cells Corrects Progranulin deficiency”, is not supported at all by the currently provided data.

Response: We addressed the Reviewer’s comments by adding new data to the revised version of the manuscript, including:

1. We included the BU-only condition and compared its efficiency and efficacy to BU+PLX3397. The results showed lower microglia replacement and undetectable brain GRN with BU-only despite similar bone marrow chimerism. See updated Fig 6.

Line 422; “To evaluate the effectiveness of our optimized BU + PLX conditioning in comparison with BU alone, we transplanted 2-month-old *Grn*^{-/-} mice¹⁰³ with wild-type BM from CAG-GFP mice (WT BMT, **Fig. 6c**). Conditioned *Grn*^{-/-} mice transplanted with BM from *Grn*^{-/-} mice (KO) were used as sham controls. Four months after transplant, the donor chimerism in BM and PB was similarly high after BU and BU+PLX conditioning (>80%, **Fig. 6d**) but differed slightly in the spleen (85% vs. 70%, respectively, **Fig. 6d**). The myeloid (CD11b+) and B-cell compartments (CD19+) were mostly donor-derived while T (CD3+) and Ly6C+ cells had a higher contribution of host-derived cells (**Fig. 6e-g**). Compared to BU alone, PLX improved T cell chimerism in the BM and spleen while it decreased Ly6C+ chimerism in PB (**Fig. 6e-g**). Following BU or BU+PLX treatment and WT BMT, serum GRN levels ranged from 16% to 51% of wild-type levels, as measured by ELISA and Western blot, and there was no significant difference between the two conditions (**Fig. 6h-j**).

Consistent with our observation in wild-type mice, brain chimerism was significantly higher in *Grn*^{-/-} mice conditioned with BU+PLX than in those conditioned with BU alone ($64 \pm 13\%$ vs. $7 \pm 3\%$, respectively, **Fig. 6k**). Notably, despite similar amounts of GRN being measured in the circulation, GRN was only detected in the brain of *Grn*^{-/-} mice conditioned with BU+PLX by both ELISA and Western blot (**Fig. 6l-m** and **Supplementary Fig. 15a**). Brain GRN was reconstituted between 23% to 33% of wild-type levels on average when using BU+PLX and WT BMT (**Fig. 6l-m**). ”

2. We tested disease markers reproducibly reported to show abnormalities in GRN-deficient mice at the ages at which we performed our analysis. To summarize, BU+PLX and WT BMT treatment corrected lipofuscin storage, ubiquitinated protein storage, and defects in CATHEPSIN D maturation. Importantly, BMT with BU alone did not improve lipofuscinosis despite similar reconstitution of GRN in the periphery. See updated Figures 6 and 7.

Line 442.”Studies on the brain histopathology of GRN-deficient mice have shown a late-onset appearance of several disease markers, such as microgliosis and astrogliosis (≥ 9 -12 months of age)^{45,57,103-106}.

However, lipofuscin storage in the CA3 region of the hippocampus, a well-known disease biomarker, has been detected in both young and aged GRN-deficient mice¹⁰⁶⁻¹⁰⁹. We confirmed a significant increase in CA3 lipofuscin deposits in *Grn*^{-/-} compared to WT mice at six months of age. Notably, lipofuscin storage was cleared in *Grn*^{-/-} treated with BU+PLX and WT BMT, while no significant decrease was observed in *Grn*^{-/-} mice after BU conditioning and WT BMT (Fig. 6n-o).

Line 455. “Proteostasis defects such as altered maturation of lysosomal proteases and accumulation of ubiquitinated proteins have been reported in young and aged *Grn*^{-/-} mice and in FTD-GRN individuals^{55,110-112}. We confirmed an increased storage of ubiquitinated proteins and altered CATHEPSIN D (CTSD) maturation in brain lysates of *Grn*^{-/-} mice compared to WT. Both defects were normalized in *Grn*^{-/-} mice treated with BU+PLX and WT BMT (Fig. 7d-f).”

3. To further confirm GRN secretion from MGLCs, we sorted and cultivated them in parallel with BM-CD11b+ cells (sorting of both populations was performed as depicted in Fig. 3a). GRN ELISA assay on culture media confirmed GRN secretion from MGLCs at higher levels compared to BM-CD11b+ cells (new Fig. 6b).

Reviewer 4

The study of microglia repopulation by bone marrow-derived stem cells is of high interest because of the therapeutic implications of this phenomenon that has been observed in pre-clinical and clinical data for neurological diseases. However, the mechanisms involved in the migration and differentiation of hematopoietic-derived MGLCs in the CNS are still under investigation. This comprehensive study not only provides an optimized protocol to achieve almost full replacement of microglia by microglia-like cells (MGLCs) in a very timely manner using a combination of CNS-penetrant myeloablation regimen Bu or TBI, and the CSF1R inhibitor PLX3397, but Colella and colleagues also studied the transcriptional and proteomic identity of the MGLCs. Indeed, they performed single cell RNAseq and high-dimensional CyTOF mass cytometry showing that MGLCs have a hybrid microglia/brain border-associated macrophage (BAM) transcriptional and translational identity. Importantly, they have identified several surface markers specific of MGLCs so that can be used to differentiate them from host MGs. The authors also identified chemokines that increased in the brain prior to the MGLC engraftment and may represent signals for the recruitment, expansion, or maturation of MGLC progenitors in the brain. Finally, they validated their conditioning strategy in a mouse model of GRN deficiency transplanted with wild-type bone marrow cells.

Below my minor and major comments:

Responses to comments from Reviewer 4

We thank the reviewer for emphasizing our work's multiple contributions to the field and for the helpful comments.

Major comment 1. Figure 1b, it looks like there are only 2 mice in the group BU+PLX. No statistical analysis can be performed with 2 mice, and if this is the case, additional mice have to be added in this treatment group at the 3-month timepoint. Also, the mice numbers are difficult to distinguish in

some graphs, adding the n number in the legend would be clearer.

Response: We increased the number of mice treated with Busulfan+PLX3397 from n=2 to n=4 (new Fig. 1b). The number of mice/cohorts is detailed in each figure legend. The n's have been added to all the figure legends.

Major comment 2. Figure 1d, there are 2 distinct groups in the GFP+ engraftment cells in the BU group. Is there a sex difference? Busulfan impact can be influenced by sex, so the authors should carefully analyze and show whether sex had no impact in the results by adding more information on the sex of the mice used (males in blue dots and female in red for instance).

Response: The reviewer is correct in that Busulfan conditioning has been reported to be influenced by sex. However, the variability we observe in the engraftment of MGLCs in the brains of mice treated with BU alone (original Fig. 1d, now Fig. 1e) is not sex-dependent, as depicted in the graph below. For the analysis, we separated females (F) and males (M) and performed a statistical analysis using the two-tailed unpaired t-test, as the data showed a normal distribution.

Minor comments

1. The discrimination index should be provided for the NORT test.

Response: We provided this data in new Fig. 2o. We observed no statistically significant differences among treatment cohorts.

2. In figures 4 and 5, the authors introduce the disease MG as coming from a neuropathic LSD. While references are present, more information should be provided in the manuscript on this disease and the rationale for using this model as control group. This said, this group do not add much information in an already very comprehensive manuscript and may be removed.

Response: We agree with the reviewer that this comparison does contribute much to our main conclusions. We have removed the data concerning the disease MG from the revised version of the manuscript. Of note, the disease microglia were from mice with Mucopolysaccharidosis type 1.

3. In figure 4, MHCII is shown to be highly expressed in CD45+ CD11b+ MGLCs in mice receiving BU alone but not in the MGLCs in mice receiving BU+PLX. The authors should discuss this very intriguing data.

Response: We discussed this finding in the Results and Discussion sections as reported below:

Results line 338. “Compared to naïve MG, the fraction of MHCII+ MGLC (GFP+) was highest in mice conditioned with BU alone (naïve $2.5 \pm 0.5\%$ vs. BU $25 \pm 6.5\%$, Fig. 4f-g) and significantly higher than in MGLCs of mice treated with BU + PLX-treated mice ($6 \pm 0.7\%$). This difference may be attributed to the

localization of most MGLCs engrafted in mice conditioned with BU alone to the choroid plexus (**Fig. 1h**), where MHCII^{high} macrophages are replenished by hematopoiesis-derived cells with a fast turnover⁸⁷.”

Discussion line 585. “Notably, although the BAM-characteristic MHCII genes were induced in all MGLC subpopulations, the expression of MHCII on MGLCs was higher following conditioning with BU alone compared to BU + PLX3397. This finding may be attributed to the rapid turnover of hematopoietic-derived MHC^{high} macrophages in the choroid plexus⁸⁵, where most BU-induced MGLCs localize. Another possibility is that MHCII^{high} MGLCs are induced in response to the high fraction of senescent host microglia that persists in the brain following BU if not cleared by PLX3397³³. All MGLC populations showed increased expression of *ApoE* and *Lyz2*, which are increased in both BAMs and embryonic E14.5 microglia^{82,87}.”

4. Fig 5f is referenced in line 374, but there is no figure 5f panel.

Response: We apologize for the typo. In the revised version of the manuscript Figure 5 has been re-arranged.

5. Figure 6a shows that MGLCs expresses high level of Grn, but not that they secrete GRN. The sentence in lines 386-387 should be revised accordingly.

Response: To confirm GRN secretion from MGLCs, we sorted and cultivated them in parallel with BM-CD11b+ cells (sorting of both populations was performed as depicted in Fig. 3a). GRN ELISA assay on culture media confirmed GRN secretion from MGLCs at higher levels compared to BM-CD11b+ cells (new Fig. 6b).

6. To validate that BU+PLX was superior to BU alone for the treatment of GRN-deficiency using bone marrow cell transplant, both strategies should have been used simultaneously and the outcomes compared. Indeed, partial reconstitution of GRN led to normalized BMP lipid metabolism in *Grn*^{-/-} mice receiving BU + PLX and WT BMT. Similar result may have been achieved with BU alone.

Response: We addressed the Reviewer’s comments by adding new data to the revised version of the manuscript, including:

1. We included the BU-only condition and compared its efficiency and efficacy to BU+PLX3397. The results showed lower microglia replacement and undetectable brain GRN with BU-only despite similar bone marrow chimerism. See updated Fig 6.

Line 422; “To evaluate the effectiveness of our optimized BU + PLX conditioning in comparison with BU alone, we transplanted 2-month-old *Grn*^{-/-} mice¹⁰³ with wild-type BM from CAG-GFP mice (WT BMT, **Fig. 6c**). Conditioned *Grn*^{-/-} mice transplanted with BM from *Grn*^{-/-} mice (KO) were used as sham controls. Four months after transplant, the donor chimerism in BM and PB was similarly high after BU and BU+PLX conditioning (>80%, **Fig. 6d**) but differed slightly in the spleen (85% vs. 70%, respectively, **Fig. 6d**). The myeloid (CD11b+) and B-cell compartments (CD19+) were mostly donor-derived while T (CD3+) and Ly6C+ cells had a higher contribution of host-derived cells (**Fig. 6e-g**). Compared to BU alone, PLX improved T cell chimerism in the BM and spleen while it decreased Ly6C+ chimerism in PB (**Fig. 6e-g**). Following BU or BU+PLX treatment and WT BMT, serum GRN levels ranged from 16% to 51% of wild-type levels, as measured by ELISA and Western blot, and there was no significant difference between the two conditions (**Fig. 6h-j**).

Consistent with our observation in wild-type mice, brain chimerism was significantly higher in *Grn*^{-/-} mice conditioned with BU+PLX than in those conditioned with BU alone (64 ± 13% vs. 7 ± 3%, respectively, **Fig.**

6k). Notably, despite similar amounts of GRN being measured in the circulation, GRN was only detected in the brain of *Grn*^{-/-} mice conditioned with BU+PLX by both ELISA and Western blot (**Fig. 6l-m and Supplementary Fig. 15a**). Brain GRN was reconstituted between 23% to 33% of wild-type levels on average when using BU+PLX and WT BMT (**Fig. 6l-m**). ”

2. We tested disease markers reproducibly reported to show abnormalities in GRN-deficient mice at the ages at which we performed our analysis. To summarize, BU+PLX and WT BMT treatment corrected lipofuscin storage, ubiquitinated protein storage, and defects in CATHEPSIN D maturation. Importantly, BMT with BU alone did not improve lipofuscinosis despite similar reconstitution of GRN in the periphery. See updated Figures 6 and 7.

Line 442. ”Studies on the brain histopathology of GRN-deficient mice have shown a late-onset appearance of several disease markers, such as microgliosis and astrogliosis (≥ 9 -12 months of age)^{45,57,103-106}. However, lipofuscin storage in the CA3 region of the hippocampus, a well-known disease biomarker, has been detected in both young and aged GRN-deficient mice¹⁰⁶⁻¹⁰⁹. We confirmed a significant increase in CA3 lipofuscin deposits in *Grn*^{-/-} compared to WT mice at six months of age. Notably, lipofuscin storage was cleared in *Grn*^{-/-} treated with BU+PLX and WT BMT, while no significant decrease was observed in *Grn*^{-/-} mice after BU conditioning and WT BMT (**Fig. 6n-o**).

Line 455. ”Proteostasis defects such as altered maturation of lysosomal proteases and accumulation of ubiquitinated proteins have been reported in young and aged *Grn*^{-/-} mice and in FTD-GRN individuals^{55,110-112}. We confirmed an increased storage of ubiquitinated proteins and altered CATHEPSIN D (CTSD) maturation in brain lysates of *Grn*^{-/-} mice compared to WT. Both defects were normalized in *Grn*^{-/-} mice treated with BU+PLX and WT BMT (**Fig. 7d-f**). ”

3. To further confirm GRN secretion from MGLCs, we sorted and cultivated them in parallel with BM-CD11b+ cells (sorting of both populations was performed as depicted in Fig. 3a). GRN ELISA assay on culture media confirmed GRN secretion from MGLCs at higher levels compared to BM-CD11b+ cells (new Fig. 6b).

Together, these data demonstrate that CNS-wide microglia replacement by GRN-secreting MGLCs is necessary to provide therapeutic benefits in *Grn*^{-/-} mice.

7. Define MLD (line 438).

Response: We defined MLD as Metachromatic leukodystrophy.

8. Line 478: the authors probably mean to say “insufficient”.

Response: We reworded the sentence to clarify its meaning:

Discussion line 536 : “Nevertheless, our findings show that administering PLX3397 before transplantation is superior to using BU alone, even though it replaces less microglia than post-transplant PLX3397. The pre-transplant PLX3397 regimen could be preferable for diseases where lower than 80-90% microglia replacement is sufficient to achieve a therapeutic effect.”

Reviewer 5

Overall, this is an interesting study with a lot of data and appropriate controls, however, some points should be clarified as follow:

Responses to comments from Reviewer 5

We appreciate Reviewer 5 for the thorough review of our manuscript and valuable insights, especially regarding the scRNA-seq data.

Major comments:

1. In Fig 1 I-o: there is no discussion of what cells are secreting the cytokines. There is no mention of where they could come from, could neurons/astrocytes be involved in attracting new microglia-like cells?

Response: We agree with the reviewer this is an interesting question and want to warrant further investigation in future work. We now include the following discussion:

Line 618. “Here, we report several cytokines induced in the brain early after CSF1Ri. CSF1R ligands CSF1 and IL34 were transiently increased and may signal an “empty” niche to induce the proliferation of the endogenous repopulating pool. CSF1 is primarily expressed in astrocytes, oligodendrocytes, and microglia, while IL34 in neurons^{125,126}.”

Line 627. “Together, the data suggests this is a highly regulated effort to repopulate and replenish the depleted microglial niche by combining myeloid proliferative and chemoattractant signals. While it is probable that the recruitment effort is led by non-microglial cells such as neurons, astrocytes, and oligodendrocytes, further investigation is required to determine the specific contributions of each cell type.”

2. The single-cell transcriptomics analysis is interesting and there is a lot of effort to characterize the signature of these infiltrating cells in general. However, since these cells are not entirely microglial cells, it would be interesting to look at hematopoietic cell or monocyte markers.

Response: Based on the reviewer’s suggestion, we also looked at hematopoietic, monocyte markers, and markers of brain-infiltrating monocyte-derived cells (MdCs) in our scRNA-seq and CyTOF data. These results are described in lines Fig 5f-g and Supplementary Figures 14.

Line 393. “Genes primarily expressed in monocyte and dendritic cells^{87,96} were either not expressed or expressed in a small fraction of MGLCs (**Supplementary Fig. 14a-b**). Compared to naïve MG, only *Irgal* was expressed in a higher fraction of MGLCs ($17 \pm 5\%$ vs. $2.3 \pm 1\%$ respectively; t-test p-value 0.006) and was primarily found in cluster 7 ($77 \pm 3.4\%$), the most divergent MGLC subpopulation (**Supplementary Fig. 14a-b**). Analyses of markers upregulated in brain monocyte-derived cells (MdCs)¹⁰⁰, such as CD64¹⁰⁰ and CD86¹⁰⁰, using high-dimensional CyTOF mass cytometry, showed that their expression is highest in the BM-CD11b+ cells. A higher percentage of MGLCs expressed CD64 compared to naive MG (16% vs. 3.7%, respectively, **Fig. 5f-g**), while no significant differences were found in the expression of CD86 (**Fig. 5f-g**).

We also examined CD169 (aka Siglec1) and MAC-2/LGALS3 (aka GALECTIN 3) as they are expressed in BAM and MDCs in the adult brain¹⁰⁰ but not in healthy microglia^{82,93-95,101}. While the fraction of MAC-2+ CD45+ CD11b+ cells in the brain did not differ, CD169 stained most MGLCs and BM-CD11b+ cells but not naive MG, thereby constituting another MGLC-specific surface marker (Fig. 5f-g). Analysis of genes expressed in HSPCs showed mostly absent HSPC markers in MGLCs, host, and naïve MG except for *Cd48* and *Cd34* (Supplementary Fig. 14c-d).”

3. The UMAP of host microglia GFP- is confusing in Fig. 3 and 4. UMAP for host microglia GFP- looks stretched.

Response: UMAPs of host microglia have been replaced in all figures (new Fig 3b,f, new Fig. 4c; new Fig. 6a; new Supplementary Fig. 8a; new Supplementary Fig. 9b).

4. One population, cluster 8, is the only one that expresses *Sall1*, a transcriptional factor essential for microglial identity, and does not express BAM markers. There is not a clear discussion around cluster 8, while it seems to be present across all experimental groups, including in the GFP+ cells. Discussing the heterogeneity of the response to infiltrated cells would be interesting. In addition, some staining to locate this population in BU+PLX+BM transplant mice would be interesting to see if it is regionally localized or spread out across the tissue.

Response: The new single-cell RNA-seq data in the revised manuscript was generated using three distinct mice per group. In contrast to the original data, this dataset differs in that the cells were in the mice for a longer period (9 vs. 4 months). Although the main points have not changed, the cluster numbering has changed.

The new single-cell RNA-seq data confirmed the expression of *Sall1* in a small fraction of MGLCs ($3.6 \pm 0.3\%$ *Sall1*+ MGLCs, new Fig. 3e) mainly distributed in cluster 0. Notably, MGLCs in cluster 0 co-expressed BAM and homeostatic microglia markers (new Fig. 3e-g and new Fig. 4 a-b).

We tried to localize these *Sall1*+ MGLCs in a brain histological section by immunostaining analyses with an anti-SALL1 antibody (clone NRNSTNX), but our efforts were unsuccessful.

We discuss all our findings concerning *Sall1*+ cells in cluster 0 in the revised version of the manuscript as reported below:

Line 599. “As observed in other studies, MGLCs typically exhibit minimal or no expression of the ontogeny-specified transcriptional repressors *Sall1* and *Sall3*, which may contribute to their hybrid transcriptional state. However, we identified a small fraction of MGLCs expressing *Sall1* (3.6%). These cells are primarily found in MGLC cluster 0 and co-express genes typical of BAMs and homeostatic microglia. Cluster 0 also expressed genes enriched in CP^{epi}-BAMs, which normally express *Sall1*. Whether *Sall1*+ MGLCs localize to the apical surface of the CP epithelium like CP^{epi}-BAMs remains to be investigated.”

5. The host microglia and the naive microglia are transcriptomically different. It would be nice to discuss this and to compare them, to see if the treatment inherently changes their transcriptome. It is something that it would be interesting to know for the prospect of the treatment becoming a potential therapy.

Response: We performed additional analyses to compare the gene expression of host and naïve microglia. These new data are included in the revised version of the manuscript as reported below:

Results Line 371. “*ApoE* upregulation was also observed in host MG (2-fold increase vs. naive MG, t-test $p < 0.001$, **Fig. 5a**). To evaluate the state of the residual host MG, we performed differential gene expression analysis compared to naive MG. The analysis showed 1033 DEGs (**Fig. 5b**). Pathway analysis on the upregulated and downregulated DEGs showed significantly enriched pathways related to cellular energetic metabolism (e.g., oxidative phosphorylation), chemotherapy-induced reactive oxygen species, DNA damage, and neurodegeneration (**Fig. 5b-c and Supplementary Fig. 12d-f**). Differential gene expression analysis between all samples revealed the gene encoding for Growth hormone (*Gh*) as the top DEG in conditioned host MG (**Fig. 5d**), likely reflecting a Busulfan-induced senescent phenotype⁹⁸.”

Discussion line 633. “After exposure to myeloablative doses of Busulfan, residual host microglia show transcriptional differences from naive microglia that persist long-term. Being an alkylating drug, Busulfan results in DNA damage and replicative senescence^{33,127}. Consistent with this, our data shows alterations in genes regulating cellular metabolism, oxidative stress, DNA damage responses, and neurodegeneration pathways, with growth hormone (*Gh*) as a top differentially expressed gene likely linked to a Busulfan-induced senescent phenotype⁹⁸. These findings are particularly relevant for understanding the effects of Busulfan conditioning and HSCT in the clinical setting.”

6. The number of samples (N = 1) for the scRNA sequencing is low. Having replicates would confirm the solidity of the data.

Response: We repeated the single-cell RNA sequencing (RNA-Seq) experiment with an independent set of 12 samples, including MGLCs (n=3 mice), host microglia (MG) (n=3 mice), naive MG (n=3 mice), and bone marrow (BM)-derived CD11b+ cells (n=3 mice). Cell Hashing with oligo-tagged antibodies was employed to uniquely label cells from distinct mice, mitigating batch effects during scRNA-seq. Differential gene expression analyses revealed similar results when comparing the original manuscript's data (n=1 run, cells pooling from 2 mice/group) with the revised manuscript's data (n=3 mice/group; please see the analysis below).

The revised manuscript only includes data from the novel experiment with n=3 samples per group due to constrained and methodological differences between the n=1 and n=3 experiments, including: 1. the donor cell type used (total bone marrow in n=1 vs. LKS HSPCs in n=3), and 2. the time of analysis after transplant (4 months in n=1 vs. 9 months in n=3).

7. How many cells from each mouse were obtained for scRNA sequencing? The number must be stated.

Response: We reported this information in Supplementary Table 2, now Supplementary Table 1. We also include the number of cells in the scRNA-seq analyses in the new Supplementary Fig. 9a (MGLC 12062 cells, host MG 14036 cells, Naive MG 20470 cells, BM-CD11b+ 12974 cells).

8. Please clarify how the GFP- host MG cells were obtained. Were they from a mouse different from the GFP+ cells, and if yes, were the mice conditioned with BU+PLX prior?

Response: We obtained GFP- host MG cells from the same mice as the GFP+ cells and were indeed conditioned with BU+PLX prior. A more detailed version of how the new scRNAs-seq data was generated is reported in the methods section as follows:

Methods line 996. “ The scRNAseq experiment comprised 12 samples (MGLCs, host MG, naïve MG, and bone marrow (BM)-derived CD11b+ cells; n=3 mice per sample). Specifically, three adult C57BL/6J mice (Jax strain #000664) conditioned with Busulfan+PLX3397 and transplanted with Lin- KIT+SCA-1+ (LKS) HSPCs isolated from C57BL/6-Tg(CAG-EGFP)131Osb/LeySopJ mice (Jax strain #006567) were used to FACS sort GFP+ CD45+ CD11b+ MGLCs, GFP- CD45+ CD11b+ host MG, and GFP+ CD45+ CD11b+ BM-CD11b+ cells. Additionally, three age- and sex-matched untreated C57BL/6-Tg(CAG-EGFP)131Osb/LeySopJ mice were used to isolate GFP+ CD45+ CD11b+ naïve MG. Details regarding mouse conditioning and transplantation can be found in the sections above.”

9. BAMs signature is shown to be expressed with primarily antigen-expressing markers. In the Van Hove et al., 2019 paper that is cited, several subtypes of BAM are characterized with distinct signatures. Is the BAM signature observed in the GFP+ cells related to a BAM subtype in particular, and if not what does it mean?

Response: Based on the reviewer’s question, we included a comparison between MGLCs and naïve MG with the different BAM subclasses identified by Van Hove et al. In summary, the MGLCs do not express a signature of a specific BAM subtype. For full details see excerpts of the Result and Discussion sections.

Results line 311. “The transcriptional signature of naïve BAM subpopulations isolated from the dura (D-BAM), subdural meninges (SD-BAM), and choroid plexus (CP-BAM) has been characterized by Van Hove et al. with scRNAseq analyses⁸⁷. Most BAM-enriched genes reported by Van Hove et al.⁸⁷ were expressed at higher levels and in a larger fraction of MGLCs compared to naïve MG as a whole (**Fig. 4d**). Apart from *ApoE* and *Lyz2*, which were highly expressed in all MGLC subpopulations, BAM-enriched genes were found to be expressed at varied levels and combinations (**Supplementary Fig. 10a**). To evaluate any similarities between the described BAM and MGLCs subpopulations, we compared their gene expression. The results showed that although several CP-BAM genes (such as *Hspa1a* and *Hspa1b*), and to a lesser extent, D-BAM genes were enriched in MGLCs, we did not identify a signature of a specific BAM subtype (**Fig. 4e and Supplementary Fig. 10b-d**). Notably, CP-BAMs include a small population of *Sall1*-expressing cells residing on the apical surface of the CP epithelium (CP^{epi}-BAMs)⁸⁷. MGLC cluster 0, which contains the highest fraction of *Sall1*+ cells did not exhibit a CP^{epi}-BAM signature (**Supplementary Fig. 10d**). The expression of BAM master transcription factors in a higher fraction of MGLCs than naïve MG (e.g. *Runx3* 30 ± 2% vs. 2.7 ± 2% of cells, respectively, t-test p-value < 0.001) together with the reduced expression of *Sall1* in most MGLCs likely contributes to their hybrid transcriptional signature (**Supplementary Fig. 10e-f**). “

Discussion line 586. “We show that MGLCs acquire a hybrid homeostatic microglia/BAM/embryonic microglia signature. Most MGLC subpopulations express genes commonly enriched in BAMs. While MGLCs cannot be categorized into specific BAM subtypes⁸⁷, they shared more genes with CP-BAM and D-BAM than SD-BAM.”

10. There is a claim that there is a proliferative signal from cytokines and signaling molecules, however in Fig 3e, none of the proliferative markers are high in MGLCs. Please, clarify?

Response: In the few days immediately after PLX removal, we observed an increase in the expression of the CSF1 and IL34 both of which are proliferative cytokines for microglia, monocytes and macrophages (see Fig 1l).

At the time of the single-cell RNA sequencing (scRNA-seq) analysis (9 months after HSCT), MGLCs did not express genes involved in cycling or proliferation (2±0.1% of MGLCs were Mki67+, as shown in Fig. 3e),

suggesting that cell proliferation occurs earlier, likely around the time of PLX withdrawal when these signals are released in the brain.

11. Since the paper aims to emphasize how such a regimen could be an alternative for human microglia transplant, it would be nice to emphasize how this could be translated to humans in the discussion.

- 1) no behavioral changes and only 20% alteration in GRN specifically.
- 2) No known evidence / not cited that PLX can also deplete human microglia.
- 3) Great proof of concept but not “immediately translatable

Response: The reviewer raises a crucial question we (and others) are trying to address. We think of the translatability of this approach as two separate questions. The first one concerns the translatability of combining a similar regimen for childhood-onset neurodegenerative diseases for which HSCT is the standard of care in patients already undergoing busulfan conditioning. The second one concerns the translatability of BU + PLX for CLN11/FTD. Studies with larger animals, such as NHP, could provide additional safety, pharmacology, and biodistribution data to support this approach for any indication, including FTD/GRN, and we are currently seeking partners to do this. Of note, through a personal communication, we know of a group already seeking IRB approval at a US institution with the highest volume and most experience in transplanting LSDs.

To emphasize how this regimen could be applied to humans and specifically address the 3 points highlighted by the reviewer, we have included the following discussion:

Points 1 and 2. Discussion Line 506. “This protocol has, in principle, clinical translatability. BU is the clinical agent for neurometabolic indications of HSCT, while PLX3397 is the only FDA-approved CS1R inhibitor with available safety data. The PLX3397 dose used in our study is well below what is prescribed for patients with TGCT (humans take ~400-800 mg/day orally for repeated cycles of 4 weeks³⁶, while mice take ~3 mg/day for six days). While there is no definitive evidence that PLX3397 administration depletes microglia in humans, several observations make it likely. It is well-established that it depletes CSF1R-expressing macrophages in humans and can enter the blood-brain barrier, as demonstrated in animal studies and clinical trials in patients with glioblastoma^{36,120}. Additionally, individuals with genetic mutations that abolish or reduce CSF1R expression show either depleted or significantly reduced microglia²⁵⁻²⁷.”

2. Discussion Line 656. “These results provide crucial pre-clinical proof-of-concept for the efficacy of a microglia replacement for CLN11/FTD-GRN. The future clinical translation of this approach could be enhanced by developing autologous transplantation methods. These approaches could mitigate risks while enabling the supraphysiological expression of GRN, thereby improving therapeutic effectiveness.”

12. Pitfall of study that the authors must address:

The use of enzymes for digestion and preparation for sorting alters microglia/macrophages reactivity, thus the interpretability of the identity (specifically the cytokines signature) is difficult.

Response: The reviewer makes an excellent point. While we cannot exclude the possibility that some of the expressed cytokine genes detected by scRNAseq are induced by the enzymatic brain dissociation, we are confident that the differences in gene expression observed for a given cytokine among MGCLs, host

MG, and naive MG are interpretable. MGCLs, host, and naive MG were isolated side-by-side and underwent a standardized dissociation procedure. GFP+ MGLCs and GFP- host MG were dissociated within the same tube for each distinct mouse. Furthermore, analysis of immediate-early genes (IEGs), known to be strongly induced in microglia/macrophages following enzymatic brain dissociation, showed similar expression across the three brain cell populations (new Supplementary Fig. 13c). This point is now included in the **Results section line 386**. “While we cannot exclude the possibility that the enzymatic brain dissociation process activated some cytokine genes, we observed a similar expression of immediate-early genes in all samples (IEGs, **Supplementary Fig. 13c**)^{84,96}, supporting the validity of this comparison.”

We would also like to clarify that the cytokine quantification depicted in Figure 1l-o and Supplementary Fig. 5a is not affected by tissue dissociation, as it was performed on whole tissue lysates derived from one sagittal half of the brain.

Minor comment:

Typo in line 90, PLX 3337 instead of PLX 3397.

Response: We corrected the typo.

Natalia Gomez-Ospina, M.D., Ph.D.
Assistant Professor

Pasqualina Colella, Ph.D.
Senior Research Scientist

REVIEWERS' COMMENTS

Reviewer #1 (Remarks to the Author):

The authors have addressed all of my concerns and improved the manuscript. One minor comment regarding Line 81-82 "GRN is a highly secreted lysosomal protein that is constitutively expressed but primarily enriched in microglia in the CNS 50,51." This probably overstates things as neurons also express high levels of progranulin, up to ~half of brain progranulin in multiple studies.

Reviewer #3 (Remarks to the Author):

The authors have sufficiently addressed the comments that were raised.

Reviewer #4 (Remarks to the Author):

The authors have adequately addressed the reviewer's comments.

Reviewer #5 (Remarks to the Author):

The authors have addressed my concerns. The manuscript in its current state is significantly improved.

Natalia Gomez-Ospina, M.D., Ph.D.
Taube Pediatric Neurodegenerative Disease Scholar
Assistant Professor, Department of Pediatrics
Divisions of Medical Genetics, and Stem Cell Transplantation
Lokey Stem Cell Biology Building, Stanford, CA 94305
[*gomezosp@stanford.edu*](mailto:gomezosp@stanford.edu)

May 13th, 2024,

Point-by-point Response to Reviewer's Comments

REVIEWERS' COMMENTS

Reviewer #1 (Remarks to the Author):

The authors have addressed all of my concerns and improved the manuscript. One minor comment regarding Line 81-82 "GRN is a highly secreted lysosomal protein that is constitutively expressed but primarily enriched in microglia in the CNS 50,51." This probably overstates things as neurons also express high levels of progranulin, up to ~half of brain progranulin in multiple studies.

Response: We agree with Reviewer #1 and modified the text as reported below:

Lines 78-79: GRN is a highly secreted, ubiquitous lysosomal protein expressed in the CNS in neurons, microglia, and other glial cells^{50,51}.

Reviewer #3 (Remarks to the Author):

The authors have sufficiently addressed the comments that were raised.

Reviewer #4 (Remarks to the Author):

The authors have adequately addressed the reviewer's comments.

Reviewer #5 (Remarks to the Author):

The authors have addressed my concerns. The manuscript in its current state is significantly improved.

We sincerely thank the reviewers for their remarks which improved our manuscript.

Natalia Gomez-Ospina, M.D., Ph.D.
Assistant Professor

Pasqualina Colella, Ph.D.
Senior Research Scientist